# Ancient gene linkages support ctenophores as sister to other animals

Darrin T. Schultz[1,2,3 ✉], Steven H. D. Haddock[2,4], Jessen V. Bredeson[5], Richard E. Green[3], Oleg Simakov[1 ✉] & Daniel S. Rokhsar[5,6,7 ✉]

A central question in evolutionary biology is whether sponges or ctenophores (comb jellies) are the sister group to all other animals. These alternative phylogenetic hypotheses imply different scenarios for the evolution of complex neural systems and other animal-specific traits[1–6]. Conventional phylogenetic approaches based on morphological characters and increasingly extensive gene sequence collections have not been able to definitively answer this question[7–11]. Here we develop chromosome-scale gene linkage, also known as synteny, as a phylogenetic character for resolving this question[12]. We report new chromosome-scale genomes for a ctenophore and two marine sponges, and for three unicellular relatives of animals (a choanoflagellate, a filasterean amoeba and an ichthyosporean) that serve as outgroups for phylogenetic analysis. We find ancient syntenies that are conserved between animals and their close unicellular relatives. Ctenophores and unicellular eukaryotes share ancestral metazoan patterns, whereas sponges, bilaterians, and cnidarians share derived chromosomal rearrangements. Conserved syntenic characters unite sponges with bilaterians, cnidarians, and placozoans in a monophyletic clade to the exclusion of ctenophores, placing ctenophores as the sister group to all other animals. The patterns of synteny shared by sponges, bilaterians, and cnidarians are the result of rare and irreversible chromosome fusion-and-mixing events that provide robust and unambiguous phylogenetic support for the ctenophore-sister hypothesis. These findings provide a new framework for resolving deep, recalcitrant phylogenetic problems and have implications for our understanding of animal evolution.

Five major lineages arose early in animal evolution and survive to the present day: sponges (poriferans), ctenophores (comb jellies), placozoans (microscopic flat animals), cnidarians (such as anemones, jellyfishes and hydra) and bilaterians (such as chordates, molluscs, arthropods and diverse worms)[1,8,10,13,14]. Although morphological and phylogenomic studies consistently unite bilaterians, cnidarians, and placozoans into a monophyletic clade (Parahoxozoa) that excludes sponges and ctenophores[8,10,14] the relationship between sponges, ctenophores and Parahoxozoa remains controversial. There are two competing scenarios—the sponge-sister hypothesis[7,8] and the ctenophore-sister hypothesis[9,10]—reflecting which lineage diverged first among animals (Fig. 1a).

As sponges and ctenophores are such disparate animals[13], the nature of the first diverging animal lineage has implications for the evolution of fundamental animal characteristics. Adult sponges are generally sessile filter-feeding organisms with body plans organized into reticulated water-filtration channels, structures built out of silica or calcium carbonate, and specialized cell types and tissues used for feeding, reproduction and self-defence, but they lack neuronal and muscle cells[15]. By contrast, ctenophores are gelatinous marine predators that move using eight longitudinal 'comb rows' of ciliary bundles[16,17]; they are superficially similar but unrelated to cnidarian medusae[13,18] and possess multiple nerve nets[19]. Thus, whereas the sponge-sister scenario suggests a single origin of neurons on the ctenophore–parahoxozoan stem, the ctenophore-sister scenario implies either that either ancestral metazoan neurons were lost in the sponge lineage, or that there was convergent evolution of neurons in the ctenophore and parahoxozoan lineages[3,6]. Similar considerations apply to other metazoan cell types[18], gene regulatory networks, animal development[13,18] and other uniquely metazoan features.

Despite its importance for understanding animal evolution, the relative branching order of sponges, ctenophores and other animals has proven to be difficult to resolve[2]. The fossil record is largely silent on this issue as verified Precambrian sponge fossils are extremely rare[20] and putative fossils of the soft-bodied ctenophores are difficult to interpret[21]. Morphological characters of living groups (for example, choanocytes of sponges) are not sufficient to resolve the question because true homology is difficult to assign, and such characters are easily lost or can arise convergently[13,22]. The ctenophore-sister hypothesis is supported by a pair of gene duplications shared by sponges,

[1]Department of Neuroscience and Developmental Biology, University of Vienna, Vienna, Austria. [2]Monterey Bay Aquarium Research Institute, Moss Landing, CA, USA. [3]Department of Biomolecular Engineering and Bioinformatics, University of California, Santa Cruz, CA, USA. [4]Department of Ecology and Evolutionary Biology, University of California, Santa Cruz, CA, USA. [5]Department of Molecular and Cell Biology, University of California, Berkeley, CA, USA. [6]Molecular Genetics Unit, Okinawa Institute of Science and Technology Graduate University, Onna, Japan. [7]Chan Zuckerberg Biohub, San Francisco, CA, USA. ✉e-mail: darrin.schultz@univie.ac.at; oleg.simakov@univie.ac.at; DSRokhsar@gmail.com

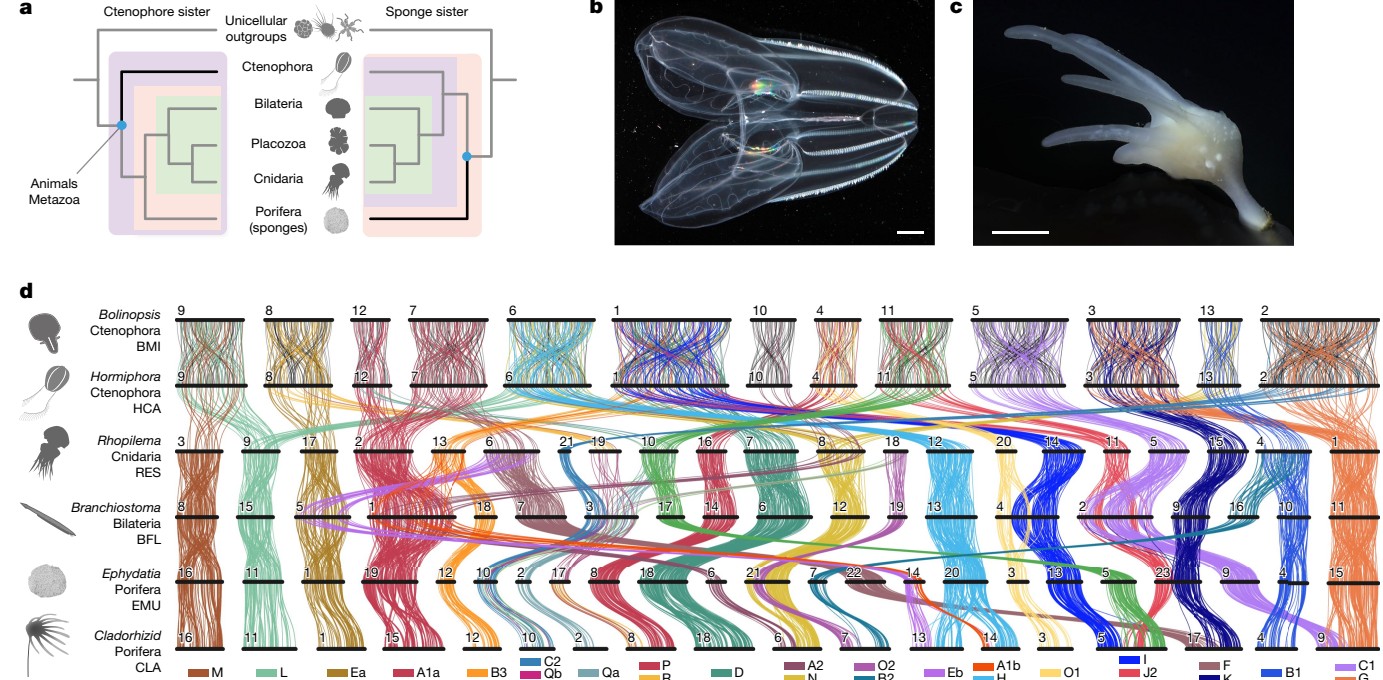

**Fig. 1 | Conserved synteny and the phylogenetic position of ctenophores and sponges. a**, Two alternative metazoan phylogenetic hypotheses, with either ctenophores (left) or sponges (right) as sister to all other animals. **b,c**, Specimens of species of which the genomes are reported here. Scale bars, 1 cm. **b**, The lobate ctenophore *B. microptera* from the Monterey Bay, California. **c**, Undescribed cladorhizid demosponge collected offshore of Big Sur, California at a depth of 3,975 m. **d**, Ribbon diagram showing conserved syntenies among animals (α ≤ 0.05, permutation test one-sided false-discovery rate), including (from top to bottom) two ctenophores (*B. microptera* (BMI) and *H. californensis*); the jellyfish *R. esculentum*; the bilaterian amphioxus *B. floridae*; and two demosponges (*E. muelleri* and the cladorhizid demosponge). Each horizontal black bar represents a chromosome. The vertical lines between species represent orthologous genes, coloured according to the BCnS synteny groups[12]. Only groups of genes that have significantly conserved chromosome-scale linkage (synteny) between metazoan species are shown. There is extensive 1:1 conserved chromosomal synteny between the two ctenophores, consistent with the conserved ctenophore karyotype. Orthologous gene pairs in the two ctenophores that do not participate in conserved syntenies with BCnS are shown in grey. Photography credits: Shannon Johnson © 2019 MBARI (**b**), © 2021 MBARI (**c**).

bilaterians, placozoans and cnidarians but not ctenophores[23]. Although sophisticated methods for sequence-based phylogenomics have been developed and applied to increasingly large molecular datasets, there is still considerable debate about the relative position of sponges and ctenophores as results are sensitive to how sequence evolution is modelled[11], which taxa or sites are included[24,25], and the effects of long-branch artifacts and nucleotide compositional variation[26]. New approaches are needed.

We reasoned that patterns of synteny, classically defined as chromosomal gene linkage without regard to gene order[27], could provide a powerful tool for resolving the ctenophore-sister versus sponge-sister debate. Chromosomal patterns of gene linkage evolve slowly in many lineages[12,28–30], probably because it is improbable for interchromosomal translocations to be fixed in populations with large effective population sizes[28,31,32]. Notably, some changes in synteny are effectively irreversible. For example, when two distinct ancestral synteny groups are combined onto a single chromosome by translocation, and subsequent intrachromosomal rearrangements mix these two groups of genes, it is very unlikely that the ancestral separated pattern will be restored by further rearrangement and fission, in the same sense that spontaneous reduction in entropy is improbable[12]. Such rare and irreversible changes are particularly useful for resolving challenging phylogenetic questions as they give rise to shared derived features that unambiguously unite all descendant lineages[33–35]. Deeply conserved syntenies observed between animals and their closest unicellular relatives[12] suggest that outgroup comparisons could be used to infer ancestral metazoan states and polarize changes within animals to address the sponge-sister versus ctenophore-sister debate. Yet, chromosome-scale genome sequences of the unicellular

or colonial eukaryotic outgroups closest to animals (choanoflagellates, filastereans and ichthyosporeans) have not been reported.

Here we show that conserved syntenies between animals and their closest unicellular relatives support ctenophores as the sister group to all other animals. Specifically, we find seven sets of genes for which (1) ctenophores share ancestral metazoan gene linkages with one or more unicellular eukaryotes; and (2) bilaterians, cnidarians, placozoans and sponges are united (to the exclusion of ctenophores) by shared derived patterns of synteny that arose by ancient interchromosomal translocations. In four of these cases, irreversible mixing after chromosome fusion evidently occurred on the bilaterian–cnidarian–sponge (BCnS) stem lineage, providing unambiguous support for the ctenophore-sister scenario. The alternative sponge-sister hypothesis is not supported by any synteny-based characters, and would require reversal of four sets of fusion-with-mixing events and/or extensive convergent fusion in both sponges and on the bilaterian/cnidarian stem to account for the observed patterns of synteny. To enable these analyses, we generated chromosome-scale genome sequences for three animal species (two sponges and a ctenophore), and three non-animal species (a filasterean, ichthyosporean and choanoflagellate) to serve as outgroups. Our analyses further reveal ancient syntenies conserved between animals and their closest unicellular relatives (animal plesiomorphies) as well as metazoan syntenies shared by all animals but not present in unicellular organisms (animal synapomorphies). These findings establish a phylogenetic framework for understanding the early evolution of metazoan genomes and characters.

To examine conserved syntenies across animals, we traced the chromosomal distribution of orthologous genes among diverse metazoan

lineages using previously and newly sequenced genomes (Fig. 1bc, Methods, Supplementary Information 1–3 and Supplementary Data 1). Figure 1d highlights conserved metazoan synteny groups, that is, groups of genes of which orthologues are linked on the same chromosome across multiple lineages, regardless of gene order. Syntenic groups shown in Fig. 1d are statistically significant (Methods). In Fig. 1d, lines connecting orthologous genes are coloured according to the previously identified BCnS ancestral linkage groups (ALGs)[12,28–30]. For example, the group on the far left represents the BCnS ALG_M (comprising genes found on jellyfish *Rhopilema esculentum* chromosome 2 (RES2), amphioxus *Branchiostoma floridae* chromosome 8 (BFL8), and sponge chromosomes CLA16 (of a cladorhizid demosponge) and EMU16 (of *Ephydatia muelleri*)). Note that, by our definition, two different conserved synteny groups can coexist on the same chromosome in some species. For example, amphioxus chromosome BFL5 is seen to be a combination of BCnS ALGs Ea and Eb, which are found on distinct chromosomes in other species.

Our results extend previous findings[12] of BCnS ALGs by incorporating a new chromosome-scale genome sequence of a recently discovered bioluminescent deep-sea cladorhizid demosponge[36] (Fig. 1c, Supplementary Information 2 and Extended Data Fig. 1c–i) complementing the spongillid demosponge *E. muelleri*[37]. Although the cladorhizid and spongillid lineages diverged approximately 450 million years ago[38], chromosomes of the two demosponges correspond simply with each other (Fig. 1d and Extended Data Fig. 2f–h) and with bilaterian and cnidarian chromosomes (Fig. 1d and Extended Data Fig. 2i–k), consistent with the previously described genome tectonic schema[12]. Further comparisons with other recently released chromosome-scale demosponge genome sequences[39] confirm the high degree of conserved synteny in this group, but show that one of the rearrangements that we found in the cladorhizid genome is the result of a fission in that lineage (ALG_H; Extended Data Fig. 2f–k). We also sequenced the genome of a previously undescribed hexactinellid (glass) sponge (Extended Data Fig. 1j–n and Supplementary Information 2), but found it to be considerably rearranged. Despite many lineage-specific genomic changes in glass sponges, relicts of 10 out of 29 BCnS ALGs are detectable (Extended Data Fig. 3). Owing to the high degree of rearrangement, we do not consider hexactinellid genomes further.

However, in contrast to demosponges, genomic comparisons between the cydippid ctenophore *Hormiphora californensis*[40] and other metazoans reveal patterns of both conserved and altered synteny (Fig. 1d and Extended Data Fig. 2b–e). For example, whereas the BCnS group ALG_Ea is localized to a single ctenophore chromosome (*H. californensis* chromosome 8 (HCA8)), the BCnS synteny group ALG_A1a (comprising genes found on amphioxus chromosome BFL1, jellyfish chromosome RES2, and sponge chromosomes EMU1 and CLA15) is partitioned across two ctenophore chromosomes (HCA12 and HCA7). To test whether the observed patterns of ctenophore synteny are unique to the *H. californensis* lineage or common across ctenophores, we assembled and analysed the genome of the recently redescribed[41] lobate comb jelly *Bolinopsis microptera* (Fig 1b; the assembly is reported in the Methods, Supplementary Information 1, Extended Data Fig. 1a,b and Supplementary Table 1.1–1.4). Despite the 160–260-million-year divergence between lobate and cydippid ctenophores[10] their *n* = 13 chromosomes show one-to-one correspondence (without gene order conservation) (Fig. 1d and Extended Data Fig. 2a). This finding implies that a common *n* = 13 karyotype is ancestral for the *Hormiphora*–*Bolinopsis* crown group, and that cross-metazoan patterns of synteny shown in Fig. 1d are general.

Interpreting the differences in synteny between ctenophores and other animals depends on the ancestral metazoan state (Fig. 2c–g, Extended Data Fig. 4 and Supplementary Information 4). If BCnS syntenies are ancestral to all metazoans, then the partitioned syntenies observed in ctenophores would have arisen by rearrangements that split the ancestral chromosomes in the ctenophore lineage (syntenic autapomorphies;

Fig. 2e) and would therefore be uninformative for discriminating between the ctenophore-sister and sponge-sister hypotheses. Alternatively, if the patterns of synteny found in ctenophores are ancestral to animals, the derived syntenies shared by BCnS to the exclusion of ctenophores could have arisen by fusion on the BCnS stem lineage, which would represent syntenic synapomorphies (Fig. 2f,g). In this case, ctenophores would be excluded from the BCnS clade and established as the sister clade of all other extant metazoans. Note that the extensive conservation of synteny between sponges, bilaterians, and cnidarians[12] confirmed here makes it improbable that ctenophores could share syntenies with cnidarians and bilaterians to the exclusion of sponges and, indeed, we did not find any such cases in analyses described below.

To provide outgroups for inferring ancestral metazoan syntenies, we assembled chromosome-scale sequences of representatives of three unicellular lineages closest to animals (collectively, outgroups): the choanoflagellate *Salpingoeca rosetta* (chromosome number, *n* = 36), the filasterean amoeba *Capsaspora owczarzaki* (chromosome number, *n* = 16) and the ichthyosporean *Creolimax fragrantissima* (chromosome number, *n* = 26). Chromosome-scale sequences and karyotypes were obtained by integrating previously reported subchromosomal draft sequences[42–44] with new chromatin conformation data (Methods, Supplementary Information 3 and Extended Data Fig. 5).

Chromosomal comparisons across animal and non-animal outgroup genomes revealed conserved ancestral metazoan synteny groups ranging in size from 5 to 29 genes, totalling 291 genes (out of 2,474 outgroup-metazoan orthology groups; Methods, Figs. 2 and 3, Extended Data Tables 1 and 2, Extended Data Figs. 5–8, Supplementary Information 5–9 and Supplementary Data 2). This finding extends previous observations based on subchromosomal assemblies of non-animal species[12]. Each such ancestral metazoan synteny group is a collection of genes of which the orthologues are consistently linked on single chromosomes in diverse metazoans and at least one outgroup (Fig. 2h–i and Extended Data Table 2). In contrast to the readily detected conserved syntenies among sponges, cnidarians and bilaterians, conserved syntenies involving ctenophores and non-animal outgroups are not visually evident in pairwise comparisons with other animals (Extended Data Figs. 2 and 6) but are statistically supported in multispecies comparisons (Methods and Supplementary Information 4 and 11). On the basis of permutation tests, the false-discovery rate of a conserved group of five linked genes in a four-species comparison is α ≤ 0.0003, and groups of eight or more linked genes never occurred in ten million permutations (Supplementary Information 8 and Extended Data Table 2). To maximize coverage of lineages relevant for the branching order of sponges and ctenophores, we considered orthologous genes across quartets of the form {outgroup, sponge, ctenophore, cnidarian/bilaterian}, which does not presuppose either the ctenophore-sister or sponge-sister hypothesis (Fig. 3 and Supplementary Information 4, 8 and 9). The extensive conservation of synteny across BCnS and within ctenophores makes our analysis insensitive to which genomes are used to represent these major metazoan clades (Supplementary Information 8). Here we used the scallop *Pecten maximus*[45], the fire jellyfish *R. esculentum*[46], the freshwater sponge *E. muelleri*[37] and the ctenophore *H. californensis* to represent the bilaterian, cnidarian, sponge and ctenophore genomes (Methods), although our findings do not depend on these choices (Figs. 2 and 3). We used two different methods for identifying orthologues—a simple mutual-best-hits method and an alternative orthologue-clustering approach (OrthoFinder[47]; Supplementary Information 10 and 11), and obtained comparable results using both approaches (Extended Data Fig. 9).

Although choanoflagellates are considered to be the closest living relatives of animals[48,49], we found that the more distantly related filasterean *Capsaspora* shares 29 conserved synteny groups with metazoans, compared to 20 between the choanoflagellate *Salpingoeca* and metazoans (Extended Data Figs. 6 and 7), perhaps indicating more rapid interchromosomal rearrangement in the *Salpingoeca* lineage. The even

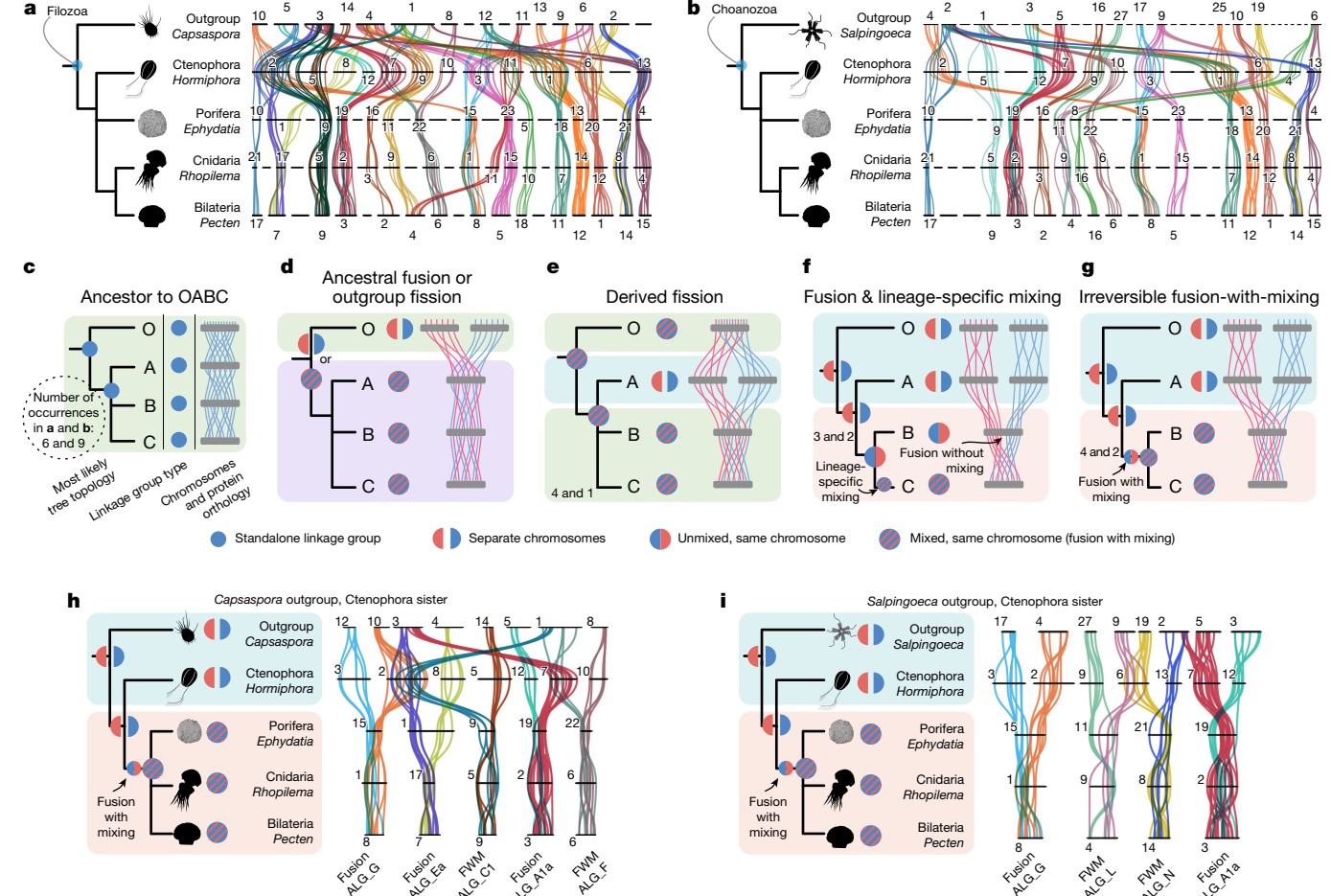

**Fig. 2 | Patterns of conserved synteny between animals and outgroups, and their implications. a,b**, Conserved linkages between the chromosomes of animals and two non-animal outgroups (α ≤ 0.05, permutation test one-sided false-discovery rate). **a**, The filasterean amoeba *C. owczarzaki*. **b**, The choanoflagellate *S. rosetta*. Each synteny group (conserved between metazoans and an outgroup) was assigned a distinct colour (different from Fig. 1). **c–g**, Schematics showing phylogenetic information (ancestor to OABC (**c**); ancestral fusion or outgroup fission (**d**); derived fusion (**e**); fusion and lineage-specific mixing (**f**); and irreversible fusion with mixing (**g**)) conveyed by patterns of conserved synteny based on a quartet analysis. Node O designates the outgroup, and nodes A, B and C are ingroups of which the phylogenetic branching is to be determined. The thin red and blue lines in **d–g** represent genes of distinct synteny groups on different chromosomes in at least one species. Changes in syntenic characters are indicated schematically on parsimonious phylogenies on the left of each diagram. **d,e**, Single-species differences are phylogenetically uninformative. **f,g**, Shared chromosomal distributions between outgroup O and one of the ingroups (labelled taxon A) imply that the other two ingroups (taxa B and C) are related by fusion of ancestral synteny groups. **f** and **g** differ in whether all fusions have subsequently mixed. Fusion with mixing (**g**) is the strongest phylogenetic character because it represents an irreversible change, as discussed in the main text. **h,i**, Subsets of the synteny groups shown in **a** and **b** (*Capsaspora* (**h**) and *Salpingoeca* (**i**)) that match the phylogenetically informative patterns indicated in **f** and **g**. In all such cases, ctenophore syntenies match the outgroup and sponges share fusions with bilaterians and cnidarians. Note that groups A1a and G are found in both outgroups. We did not observe any cases in which sponge syntenies match the outgroup to the exclusion of ctenophores, bilaterians and cnidarians.

more distantly related ichthyosporean *Creolimax* still retains eight conserved synteny groups with metazoans. Although we considered each outgroup-plus-metazoan comparison separately, we found widespread overlap between the ancient synteny groups defined independently by comparison with *Capsaspora* and *Salpingoeca*. In total, our analysis defined 31 ancestral metazoan synteny groups that are traceable to the last common ancestor of Metazoa and shared by one or both of *Capsaspora* and *Salpingoeca* (Extended Data Table 1). The extensive conservation of synteny within BCnS implies that the ancestral metazoan synteny groups correspond to subsets of the BCnS groups, and we name them using the BCnS notation with the suffixes _x and _y. If we relax the condition that an outgroup gene must be present, more metazoan genes can be added to these ancestral metazoan syntenic units (Extended Data Fig. 8).

Conservation of synteny between animals and their unicellular relatives may at first seem surprising, as these lineages diverged more than 800 million years ago[50]. Within animals, it has been estimated that ongoing small-scale translocations between chromosomes typically transfer 1% of genes to a different chromosome every ~40 million years[12]. The limited residual conservation of synteny between animals and close unicellular relatives suggests that small-scale translocations have occurred at similarly low rates along both choanoflagellate and filasterean lineages. The more extensive conservation observed between animals and *Capsaspora* versus *Salpingoeca* may be due to variations in this rate or differences in other chromosomal rearrangements over deep time. The *Capsaspora* karyotype is predominantly metacentric and, notably, we find that 11 of the 29 ancient synteny groups found in *Capsaspora* are concentrated on single chromosome arms, rather than dispersed across whole *Capsaspora* chromosomes, based on estimates of centromere position using chromatin conformation contacts. This raises the possibility that *Capsaspora* chromosome arms preserve ancient filozoan units and suggests further attention to the chromosome biology of non-metazoan relatives. We found no significant functional

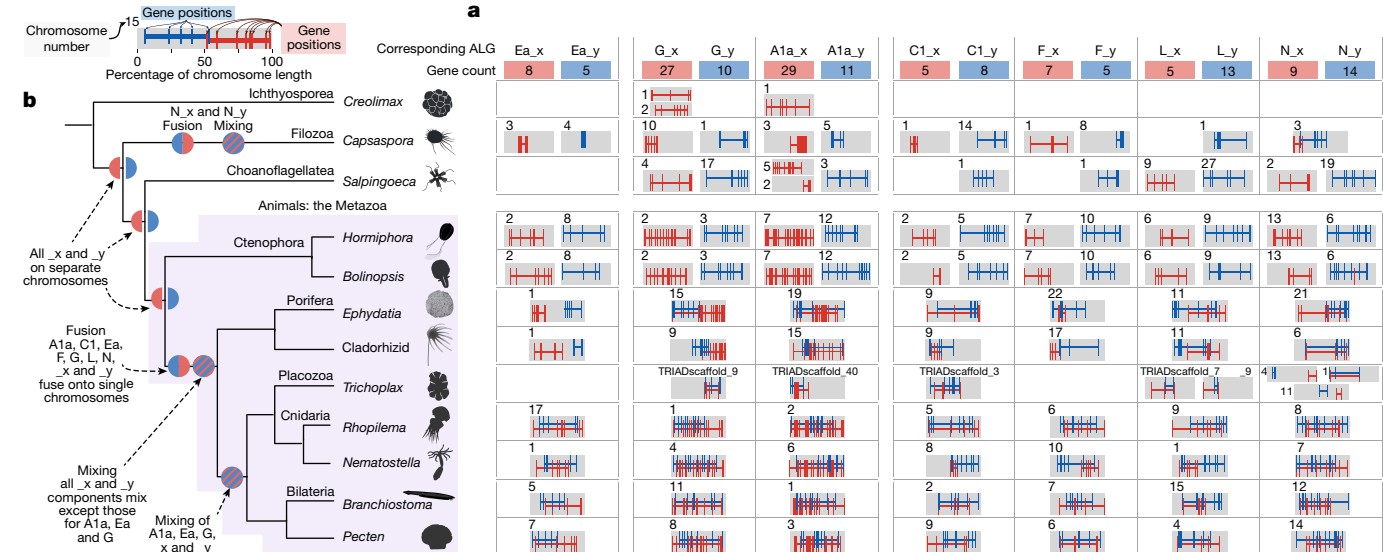

**Fig. 3 | Phylogenetically informative syntenies support ctenophores as the sister clade to other animals. a**, The rows represent the species considered in our analyses. The top three rows are non-metazoan outgroups. Columns show pairs of phylogenetically informative and significantly large metazoan syntenies ($\alpha \leq 0.05$, permutation test one-sided false-discovery rate), labelled according to their BCnS names (A1a, C1 and so on) with the suffix _x or _y denoting subgroups shared across metazoans. The number of genes participating in each metazoan synteny group is indicated in red and blue rectangles at the top of each column. Only genes with defined orthologues in outgroups are shown here. Extended Data Fig. 8 shows a larger set of genes requiring only metazoan orthologues. Inset: the convention for representing gene distributions on chromosomes (top left). The grey rectangles represent chromosomes (or large scaffolds in the placozoan *Trichoplax*). The chromosome number or scaffold name is located above or to the left of the grey rectangle. Red and blue vertical hashes represent the relative position of genes participating in phylogenetically informative pairs of metazoan synteny groups. **b**, The most parsimonious phylogeny according to the logic of Fig. 2c–g (Extended Data Fig. 4 and Supplementary Information 4), the results of ref. 12 and the accepted monophyly of demosponges.

associations of anciently linked groups of genes (Supplementary Information 12 and Supplementary Data 3), consistent with a general slow rate of synteny loss due to the infertility of translocation heterozygotes[28,31], which allows only small-scale interchromosomal translocation[32].

With conserved ancestral metazoan syntenies in hand, we tested the ctenophore-sister versus sponge-sister hypotheses by identifying shared, derived syntenic characters using standard phylogenetic methods. As noted above, two or more metazoan synteny groups can co-occur on the same chromosome in one or more genomes, corresponding to ancient fusions (that is, translocations[51]) (Fig. 2h,i). As only shared derived characters are phylogenetically informative, changes that are unique to a single lineage can be disregarded (Fig. 2e and Extended Data Fig. 4b,d,e,g). There are two different types of chromosomal fusions between two ancestrally linked groups of genes: without mixing (Fig. 2f) or with subsequent intermixing (Fig. 2g and Supplementary Data 4 and 5). Fusion-without-mixing is potentially reversible, as observed in Robertsonian fusions and fissions involving whole chromosome arms[51]. However, in the fusion-with-mixing case, reversion is extremely unlikely, comparable to the spontaneous reduction of entropy after mixing of two fluids[12].

We encoded the state of each potential fusion into a phylogenetic character matrix as 0 (no fusion, that is, ALGs found on separate chromosomes), 1 (fused but unmixed) or 2 (fused and mixed). The mixed/unmixed status of a fusion was determined on the basis of the likelihood of the observed gene arrangement under a model of random rearrangement (that is, entropy of mixing of the two fused groups) (Methods, Supplementary Information 13 and Supplementary Data 4 and 5). The same fusion character states were obtained using orthology defined by mutual-best-hits or OrthoFinder. We then applied the machinery of Bayesian phylogenetics[52] to this character matrix, using asymmetric transition probabilities to reflect the highly improbable unmixing transition (Methods and Supplementary Data 6).

Bayesian phylogenetic analyses of the fusion character matrix strongly support the ctenophore-sister topology (Fig. 4 and Supplementary Information 14). The same conclusion is clear from direct examination of the fusions identified in our data. Specifically, there are seven derived fusions shared by bilaterians, cnidarians and sponges to the exclusion of ctenophores (Extended Data Table 1). Of these seven derived fusions, four are accompanied by mixing of genes from two different ancestral chromosomes—a process that is essentially irreversible (Figs. 2e and 4b–d); the other three are mixed only in bilaterians and cnidarians (Fig. 3).

We reject the alternative sponge-sister hypothesis as it would require either (1) multiple convergent fusions (that is, involving the same groups of genes) in both the sponge and bilaterian-cnidarian lineages (Fig. 4c and Supplementary Information 15) or (2) the precise reversal of multiple fusions-with-mixings in the ctenophore lineage to match the original patterns found in the ancestral metazoan lineage (Fig. 4d). The extreme unlikeliness of recovering the observed syntenic patterns by chance is shown by simulations in which we permuted the configuration of the genes in each of the *C. owczarzaki*, *S. rosetta*, *H. californensis*, *E. muelleri* and *R. esculentum* genomes (Supplementary Information 15 and Extended Data Fig. 10). Across one hundred million randomized *Hormiphora* genomes, we never found syntenic signals comparable to those observed with the actual genome, indicating that syntenic support for ctenophore-sister is unlikely to have arisen by chance (Fig. 4e). We also note the complete absence of syntenic synapomorphies of a hypothetical ctenophore–bilaterian–cnidarian clade that excludes sponges, both in the actual data and in genome-shuffling simulations (Fig. 4f, Extended Data Fig. 10 and Extended Data Tables 1 and 2). This lack of homoplasy allows for a simple interpretation of the results (Fig. 4).

## Conclusions

### Support for the ctenophore-sister hypothesis

Our findings provide strong support for the ctenophore-sister scenario and reject the sponge-sister hypothesis. Although we encoded syntenic states as a character matrix and analysed it using a Bayesian

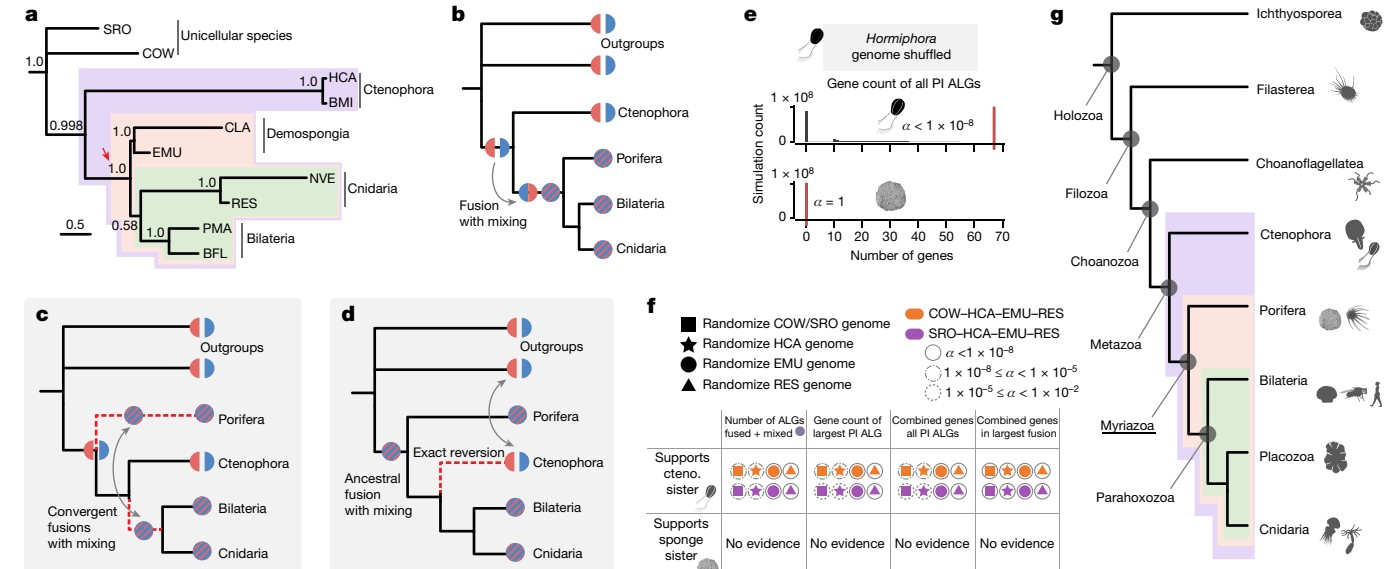

**Fig. 4 | Phylogenetic analysis of patterns of conserved synteny and alternative interpretations. a**, Bayesian phylogenetic analysis of conserved syntenies supports monophyly of the group comprising demosponges, Cnidaria and Bilateria, to the exclusion of Ctenophora, with high posterior probability (red arrow: 1.0, 100,000 generations with 25% burn-in). Bayesian analysis was run on both constrained (per-phylum constrained tree shown) and unconstrained tree topologies (Supplementary Information 14 and Supplementary Data 6). This panel corresponds to Supplementary Fig. 14.1c. **b**, Character transitions involving ALGs C1, F, L and N in Fig. 3 are most parsimoniously interpreted as fusion with mixing on the myriazoan stem after divergence from ctenophores, which retain the ancestral metazoan state as inferred from outgroup comparisons. **c,d**, To interpret the observed patterns under the alternative sponge-sister hypothesis would require unlikely convergent chromosomal changes (either convergent fusions (**c**) or exact unmixing and fissions to the ancestral state (**d**)) that were not seen in our genomes. **e**, The number of genes in the genome-shuffling simulations

($n = 1 \times 10^8$) that support the ctenophore-sister (upper) or sponge-sister (lower) hypothesis. For the ctenophore *Hormiphora*, the number of fusion-with-mixing events is significantly higher in the observed genomes (vertical red bars) than in the *Hormiphora* genome-shuffling simulations (vertical grey histogram bars). Significance is shown as the one-sided false-discovery rate, $\alpha$, of a genome-shuffling permutation test. There were no groups of genes that supported the sponge-sister hypothesis in the real genomes, and none occurred in the genome-shuffling simulations. **f**, Additional statistical measures also support only the ctenophore (cteno.)-sister hypothesis in genome-shuffling simulations of *Hormiphora*, *Capsaspora* (COW), *Salpingoeca* (SRO), *Ephydatia* (EMU) and *Rhopilema*. PI ALG, phylogenetically informative linkage groups. The shape indicates the treatment; the colour indicates the outgroup. The full figure is shown in Extended Data Fig. 10. **g**, Summary of phylogenetic relationships among animals and close outgroups including syntenic characters. Myriazoa (underlined) is the name proposed for the clade containing extant animals, except Ctenophora. Outgroup topology follows ref. 49.

phylogenetic framework (Fig. 4a and Supplementary Fig. 14.1), the cladistic logic supporting our conclusions is easily appreciated, as emphasized above (Fig. 2, Extended Data Fig. 4 and Supplementary Information 4). Previous phylogenetic analyses of sequence-based characters have not resolved the sponge-sister versus ctenophore-sister hypotheses because the phylogenetic signal is weak and distributed across thousands of individual amino acid positions that are often saturated or subject to confounding evolutionary forces[11]. By contrast, the synteny-based characters that support ctenophores as sister to other animals in our analysis are clear: sponges, bilaterians, and cnidarians share multiple irreversible changes in synteny to the exclusion of ctenophores (BCnS syntenic synapomorphies) (Figs. 2 and 3). Support for the ctenophore-sister hypothesis is directly testable by future genome sequencing, as it is a strong prediction of our model that all bilaterian, placozoan, cnidarian or sponge genomes should share the four fusion-with-mixing syntenic synapomorphies shown in Figs. 2 and 3 and, to a lesser extent, the three fusion-without-mixing events (pending considerations of sponge monophyly; Supplementary Information 7.2.6). The placement of ctenophores as sister to other animals also rejects the old notion of a Coelenterata clade that would unite ctenophores with cnidarians[53].

## Myriazoans

The clade containing all sponges, bilaterians, cnidarians and placozoans is diverse, accounting for all living animals other than ctenophores. In recognition of this morphological diversity, we propose that this

clade be called Myriazoa, from the Greek myria (extremely great in number) and zoa (animals) (Fig. 4g). While Myriazoa is supported by shared derived chromosomal fusions, there are currently no obvious morphological characters that unite them. The name Benthozoa was proposed for this clade[23] on the basis of the inference of a pelagic ancestral metazoan and a derived benthic adult ancestor of the clade sister to ctenophores, but a benthic life history stage may not be a shared derived feature of this clade. In particular, it would be just as parsimonious for the ancestor of Metazoa to have had a benthic stage, and for most ctenophores to have lost it. We therefore prefer to avoid any assumption of the ancestral life history strategy in referring to the clade.

## Parahoxozoans, sponges and placozoans

A clade grouping bilaterians, placozoans and cnidarians to the exclusion of sponges and ctenophores[54] has been recovered in multiple phylogenetic studies[8,10] and is now called Parahoxozoa on the basis of the shared presence of Hox/ParaHox-class genes[14]. Parahoxozoa is supported in our analysis by the disposition of the ancestral myriazoan linkage groups Ea and G, which are each partitioned across two chromosomes in non-metazoan outgroups and ctenophores. The pre-myriazoan partitions of Ea and G are fused in demosponges and parahoxozoans, but are mixed only in parahoxozoans, providing a candidate parahoxozoan synapomorphy. The most parsimonious interpretation is that fusions forming Ea and G occurred without mixing on the myriazoan stem, a state that is preserved in demosponges, but that mixing occurred on the parahoxozoan stem lineage so that the

mixed state is shared by all bilaterians, cnidarians and placozoans. However, a detailed understanding of the history of Ea and G linkages in sponges will require chromosome-scale genome sequences from other sponge classes beyond demosponges and lyssacinosid glass sponges (Supplementary Information 7.2.2, 7.2.5 and 7.2.6). If sponges are monophyletic (as supported by recent phylogenomic studies[8,10,55]), then the four fusions-with-mixing that are found in demosponges and parahoxozoans must be shared by all sponges. However, if one or more sponge classes branched before the split between the demosponge and parahoxozoan lineages, it is possible that the descendants of the early-branching sponges might not possess one or more of these myriazoan fusions-with-mixing.

Although the subchromosomal assemblies currently available for *Trichoplax* preclude its full integration into the present analysis, Fig. 3 shows that placozoans share the diagnostic myriazoan fusion-with-mixing characters related to ALG_C1 and the two bilaterian–cnidarian fusions-with-mixing related to ALGs A1a and G. The placozoa-sister-to-other-animals hypothesis[56] is rejected by the placement of placozoans within Myriazoa using synteny. It is therefore a strong prediction of our overall approach that chromosome-scale assemblies of placozoans will show that they share the fusions and mixing events that define Myriazoa. Furthermore, we previously showed that cnidarians and placozoans are united as sister lineages to the exclusion of bilaterians and sponges based on the mixing of genes from ALG_Ea and ALG_F found on cnidarian chromosomes and placozoan scaffolds[57], consistent with recent gene trees[55]. These characters do not appear in the present analysis owing to the stringent requirement that syntenies considered here are also preserved in outgroups to Metazoa. If placozoans are nested within Parahoxozoa, homologies between the mouth, gut and nervous systems of cnidarians and bilaterians imply that placozoans are secondarily flattened and have lost an ancestral nervous system, rather than representing the ancestral parahoxozoan state.

### Implications for early animal evolution

Finally, we consider implications of the ctenophore-sister hypothesis for early animal evolution[1,2]. Comparisons among diverse genomes have identified numerous genes that are present in myriazoans but are absent in ctenophores[1,5,58]. Under the ctenophore-sister scenario, these are most parsimoniously interpreted as arising on the myriazoan stem after the divergence of ctenophores[1,5,59], and include genes associated, in bilaterians and cnidarians, with neuronal function[1,5,59], development[58] and cell adhesion[60]. However, as gene loss is common throughout animal evolution[61], it is also possible that some of these genes were present in the ancestral metazoan but lost in ctenophores. Similarly, some genes are present in ctenophores and parahoxozoans but absent in sponges[58], and these must be interpreted as gene losses on the sponge lineage.

Perhaps the most intriguing suite of metazoan characters pertain to neuromuscular systems, which are present in varying complexity in ctenophores, bilaterians, and cnidarians but are absent in sponges[3,6,59]. In sponge-sister scenarios, these characters are interpreted as being primitively absent, arising after the divergence of sponges on the stem lineage leading to other animals. However, in the ctenophore-sister scenario supported here by deeply conserved syntenies, there are two possible alternatives explaining the evolution of neurons: either complex neural systems arose more than once[3,59,62] but were elaborated differently in ctenophores, cnidarians and bilaterians[3,4,59], or neuronal cell types were present in the metazoan ancestor but were lost in the sponge lineage[4,9,63].

Sponge-sister and ctenophore-sister hypotheses are sometimes erroneously interpreted as suggesting that the most recent common ancestor of animals was sponge-like or ctenophore-like. We must be mindful, however, that the living representatives of sponges, ctenophores, bilaterians and placozoans may be poor surrogates for the earliest members of each stem-lineage, as the crown group of each clade arose hundreds of millions of years after their divergence from each other, let alone from the common metazoan ancestor[2]. Although living sponges are often defined by the cellular, morphological and developmental characters that they lack relative to other animals, they are complex animals in their own right, successfully adapted to a unique benthic filter-feeding lifestyle[13]. Consistent with a neuron-bearing metazoan ancestor, sponges possess secretory cell types[15] and extensive molecular components associated with presynaptic function that could be derived from a primitive neurosecretory cell. Conversely, the elaborate and divergent nervous systems of living ctenophores, bilaterians, and cnidarians do not represent the stem ancestors of these groups, which would have had very different lifestyles in the Ediacaran. The nervous systems of living ctenophores, cnidarians, and bilaterians each have unique properties[6,19,59], and could represent divergent evolution from a simpler neuron-bearing common ancestor. With the ctenophore-sister topology in hand, reconstructing the characters of this metazoan ancestor will require an improved understanding of molecular, cellular and system homologies and specializations across the full range of animal diversity.

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

## Methods

A full description of the methods is provided in the Supplementary Information.

### Unicellular outgroup species genome scaffolding

Chromatin conformation capture (Hi-C) libraries were generated[64] from frozen cell cultures obtained directly from the American Type Culture Collection. The cultures used were of the species *C. owczarzaki* (ATCC, 30864), *C. fragrantissima* (ATCC, PRA-284) and *S. rosetta* (ATCC, PRA-366). The strains used were the same as those sequenced in the original genome assembly projects for each species[42–44]. The Hi-C libraries were sequenced at a depth of over 500× for each species on the Illumina NovaSeq 6000 system at MedGenome.

Previously published draft genome assemblies[42,44] were scaffolded to chromosome-scale using a combination of HiRise (v.Aug2019)[65] and SALSA2 (v.2.3)[66]. The genomes were manually curated using PretextView v.0.2.4 (https://github.com/wtsi-hpag/PretextView), HiGlass v.1.10.0104[67], Juicebox Assembly Tools (GitHub Commit 46c7ed1105)[68], the Juicebox visualization system (v.1.11.08106)[69] and artisanal (https://bitbucket.org/bredeson/artisanal/src). For *C. owczarzaki*, we used the most recent 'v4' assembly as input for scaffolding[70]. The Hi-C data were used as evidence to remove several megabases of the original *C. fragrantissima* assembly that, after further analysis, appeared to be fungal contaminants. We identified the general location of the centromeres in *C. fragrantissima* and *C. owczarzaki* using the Hi-C data as described in Supplementary Information 3.

### Sponge and ctenophore genome assembly

Samples of *B. microptera*[41] were collected in Monterey Bay, California (36.63° N, 121.90° W) from surface waters and were reared to an $F_3$ population at the Monterey Bay Aquarium, from which one adult was sequenced. One individual cladorhizid sponge[36] was collected off the coast of Big Sur, California (35.49° N, 124° W) from the seafloor at 3,975 m. One hexactinellid 'tulip' sponge (HEX) was collected near Southern California (34.57° N, 122.56° W) from the seafloor at 3,852 m. This species of ctenophore, and presumably these species of sponges, are hermaphroditic. Sponge and ctenophore samples were collected under the State of California Department of Fish and Wildlife collecting permits SC-2026 (*Bolinopsis*) and SC-4029 (sponges).

DNA and RNA were isolated from these species to generate Pacific Biosciences (PacBio) CLR WGS, HiFi WGS libraries or PacBio Iso-Seq libraries at the Brigham Young University DNA Sequencing Center. These libraries were sequenced on the PacBio Sequel II system. Illumina WGS libraries, Chicago libraries and Hi-C libraries were generated at UC Santa Cruz and sequenced at MedGenome on the Illumina HiSeq X system. PacBio WGS library coverage was over 70× for all three species, and Hi-C coverage was over 190× for all three species. Genome sizes were estimated using jellyfish (v.2.2.10)[71], then using the resulting spectrum in GenomeScope (v.2)[72].

The genome of *B. microptera* was assembled using wtdbg (v.2.4)[73], and the sponge genomes were assembled using hifiasm (v.0.16.1-r375)[74]. Hi-C reads were mapped using bwa mem (v.0.7.17)[75], processed using pairtools (v.0.3.0)[76], pairix (v.0.3.7; https://github.com/4dn-dcic/pairix) and Cooler (v.0.8.10)[77], and scaffolding was performed using HiRise (v.Aug2019)[65]. In *B. microptera*, gaps were closed using TGS-Gapcloser (v.1.1.1)[78], haplotigs were removed using Purge Haplotigs (v.1.0.4)[79] and the assembly was polished using Illumina WGS reads and pilon (v.1.23)[80]. In both the sponge and *B. microptera* genomes, bacterial scaffolds were removed using Diamond (v.0.9.24)[81] and Blobtools (v.1.0)[82]. The genomes were manually curated with Hi-C data as described above. The haplotypes of the hifiasm-based assemblies were compared to one another using D-Genies (v.1.4.0)[83].

### Genome annotations

The unicellular outgroup genome assemblies were annotated by mapping their transcripts from the original assemblies to the Hi-C scaffolded assemblies using minimap2 (v.2.23)[84]. To clarify demosponge macrosyntenic relationships, we produced putative *Ephydatia* protein coordinates in the cladorhizid sponge using tblastn (v.2.10.0+)[85]. To annotate the hexactinellid sponge genome, we mapped the proteins of closely-related hexactinellid species[86,87] using miniprot (v.0.2)[88] (Supplementary Information 2.1.5). The *Bolinopsis* genome was annotated using BRAKER (v.2.14)[89] supplied with evidence from RNA-seq reads mapped with STAR (v.2.7.1a)[90] and minimap2 (v.2.23)[84], Iso-Seq reads processed with lima (v.2.2.0; https://github.com/PacificBiosciences/barcoding) and isoseq3 (v.3.4.0; https://github.com/PacificBiosciences/IsoSeq) then mapped with minimap2 (v.2.23)[84], and protein orthology identified using ProtHint (v.2.6.0)[91] from ctenophore transcriptomes[92–94] assembled with Trinity (v.2.5.1)[95] and translated using TransDecoder (v.5.5; https://github.com/TransDecoder/TransDecoder). We assessed genome sequence and protein datasets using BUSCO (v.5)[96].

### Orthologue Inference

Orthologues were inferred between species by finding reciprocal-best BLASTp[97] hits between the proteins in the genomes, or with OrthoFinder (v.2.3.7)[98]. The reciprocal-best BLASTp hits were used to identify macrosyntenic chromosomes between species by performing Bonferroni-corrected one-sided Fisher's exact tests[57]. To determine the provenance of the ALG_H in sponges, the genomes of *Chondrosia* and *Petrosia*[39,99], *Oopsacas*[86], CLA and HEX were compared using the odp software suite.

Orthologues shared between three, four or more species were selected by finding groups of proteins that were *n*-way reciprocal best BLASTp hits. In this conservative method, each orthogroup has a single protein from each of the *n* species. We performed this analysis for three-way and four-way comparisons of combinations of the species CFR, COW, SRO, HCA, EMU, CLA, RES, BFL, NVE and *P. maximus*.

### Gene linkage group identification

Orthologues from three-way or four-way reciprocal-best BLASTp searches were grouped by the chromosomes on which the genes occurred in the *n* species. To identify which sets of orthologues were larger than expected by random chance, we shuffled the genome coordinates of the *n* species and measured the frequency of finding sets of orthologues of size *k* on the same chromosomes in the *n* species. By performing this for 10 million iterations, we calculated the false-discovery rate ($\alpha$) of finding an orthologue set of size *k* given the *n* input genomes.

### Combined unicellular outgroup analysis

Sets of orthologues with a false-discovery rate of less than 0.05 were retained from the four-way reciprocal best hit searches of COW–HCA–EMU–RES, CFR–HCA–EMU–RES and SRO–HCA–EMU–RES. The remaining orthogroups were joined based on gene identity in HCA–EMU–RES, such that each orthologue contained a protein from at least one of the unicellular outgroup species. This yielded 291 sets of orthologues.

### Identification of orthologues in other species

For each of the 291 orthologues, we aligned the proteins using MAFFT (v.7.310)[100], built a hidden Markov Model using hmmbuild in hmmer (v.3.3.2)[101], then found the best match using hmmsearch in the proteins of the genomes of other species, including the ctenophore *B. microptera*, the cladorhizid sponge, *T. adhaerens*[102], *H. vulgaris*[12], *N. vectensis*[103], *B. floridae*[57], *P. maximus*[45] and *E. muelleri*[37]. To test for Gene Ontology enrichment of the sets of orthogroups using PANTHER (v.17)[104], we also searched for the orthologues in *Homo sapiens*[105].

### Mixing analysis

To test whether the _x and _y gene sets present on single chromosomes were well-mixed, we used a metric that counts the number of transitions between a gene in _x to a gene in _y and vice versa. To provide an

objective measure of mixing, we computed the $\alpha$ value (false-discovery rate) that the two sets of genes are unmixed by building a distribution of mixing scores from randomly sorted groups of the same size of the _x and _y groups in question. We consider $\alpha < 0.05$ to be unmixed.

## Simulations testing the ctenophore-sister and sponge-sister hypotheses

We applied this methodology to test whether the findings supporting the ctenophore-sister hypothesis were due to the arrangement of any of the observed genomes, implemented as part of the odp software suite. For both the SRO–HCA–EMU–RES and COW–HCA–EMU–RES four-way reciprocal best hit results, we performed four analyses. One analysis shuffles the genome chromosome labels of one species 100 million times. Each time the genome chromosome labels are shuffled, we perform the gene linkage group identification analysis described above, and measure the quantity and size of gene linkage groups that support either the ctenophore-sister or sponge-sister hypothesis. The distribution of these results compared with the observed data of the real genomes is used to estimate the false-discovery rate of finding support for the ctenophore-sister hypothesis or sponge-sister hypothesis. We modelled fusion-with-mixing events in the animal genomes as state transitions, and used RevBayes (v.1.1.1)[106] and MrBayes (v.3.2.7a)[52] to estimate the likelihood of the ctenophore-sister hypothesis, and we used FigTree (v.1.4.4; https://github.com/rambaut/figtree) to visualize the trees.

## Software

We implemented a suite of tools for identifying orthologues, plotting syntenic relationships and performing synteny-based phylogenetic analyses using a tool called odp, implemented in snakemake (v.7)[107] for scalability. To confirm the validity of these methods, we applied them to several genome quartets and showed that odp recovers previously identified synapomorphic chromosomal fusion-with-mixing events[12] in bilaterians and cnidarians (Supplementary Information 6).

## Reporting summary

Further information on research design is available in the Nature Portfolio Reporting Summary linked to this article.

## Data availability

All data presented in this Article are available in public repositories. The sequencing reads are available in the NCBI database under BioProject accession numbers PRJNA818620, PRJNA818630, PRJNA903214 and PRJNA818537. The genomes for each species are available through the above BioProject accession codes, with the exception of the genomes of *C. fragrantissima*, *C. owczarzaki* and *S. rosetta*, which are available at Dryad (https://doi.org/10.5061/dryad.dncjsxm47). The results shown in the Supplementary Information, when not contained in figures, are also available in the Dryad repository (https://doi.org/10.5061/dryad.dncjsxm47). Publicly available sequencing data and genomes were downloaded from NCBI from BioProject accession numbers PRJNA168, PRJDB8655, PRJNA12874, PRJNA20249, PRJNA20341, PRJEB28334, PRJNA30931, PRJNA31257, PRJNA37927, PRJEB56075, PRJEB56892, PRJNA64405, PRJNA193541, PRJNA193613, PRJNA213480, PRJNA278284, PRJNA281977, PRJNA283290, PRJNA377365, PRJNA396415, PRJNA512552, PRJNA544471, PRJNA576068, PRJNA579531, PRJNA625562, PRJNA667495, PRJNA761294 and PRJNA814716. The *E. muelleri* genome was downloaded from https://spaces.facsci.ualberta.ca/ephybase/.

## Code availability

The scripts and software developed for this manuscript are available at Dryad (https://doi.org/10.5061/dryad.dncjsxm47) and Zenodo (https://doi.org/10.5281/zenodo.7857390). Long-term development of odp is available at GitHub (https://github.com/conchoecia/odp).

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

**Acknowledgements** We acknowledge the support of the David and Lucile Packard Foundation, the Monterey Bay Aquarium Research Institute and the United States National Science Foundation (NSF). D.T.S. was supported by the NSF GRFP DGE 1339067; S.H.D.H. by NSF DEB-1542679; D.T.S. and O.S. by the European Research Council's Horizon 2020: European Union Research and Innovation Programme, grant no. 945026; D.S.R. by internal funds of the OIST Molecular Genetics Unit, the Chan Zuckerberg Biohub and the Marthella Foskett Brown Chair in Biology. We thank W. Patry and the staff at Monterey Bay Aquarium for providing the *Bolinopsis* specimen; and C. Dunn, J. Eizenga, R. Revilla-i-Domingo and G. Genikhovich for discussions. The *Petrosia* and *Chondrosia* genomes from the Aquatic Symbiosis Genomics project of the Tree of Life Programme, Wellcome Sanger Institute, were funded by the Gordon and Betty Moore Foundation and the Wellcome Trust.

**Author contributions** D.T.S., O.S. and D.S.R. designed the scientific objectives of the study. D.T.S. planned and carried out experiments and analyses, wrote the code for the analyses and the odp package, and wrote the first draft of the manuscript. O.S. and D.T.S. performed the Bayesian analyses. D.T.S. and S.H.D.H. collected samples and created figures and tables. J.V.B. assisted in genome assemblies and identified putative centromeres. D.T.S. and D.S.R. wrote the first drafts of the manuscript with input from O.S. and S.H.D.H. All of the authors contributed to the interpretation, presentation and writing of the Article and the Supplementary Information.

**Competing interests** R.E.G. and D.S.R. are paid consultants and equity holders of Dovetail Genomics. D.T.S. is a shareholder of Pacific Biosciences of California. The other authors declare no competing interests.

**Additional information**
**Correspondence and requests for materials** should be addressed to Darrin T. Schultz, Oleg Simakov or Daniel S. Rokhsar.

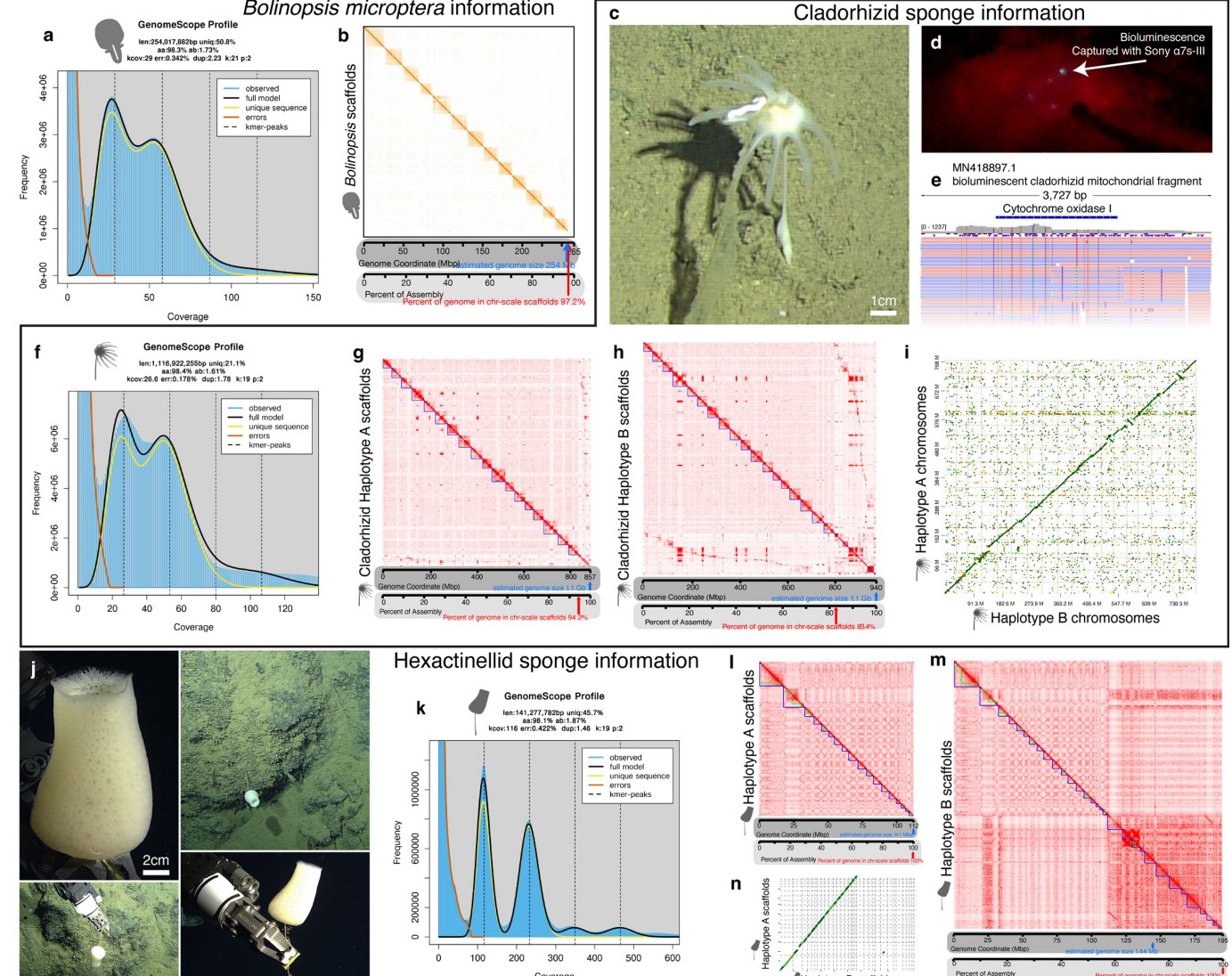

**Extended Data Fig. 1 | Genomes of one ctenophore and two sponges. a.** The k-mer spectrum of the *Bolinopsis* data suggests that the animal is diploid, and the 1n genome size is approximately 254 Mbp. **b.** The *Bolinopsis microptera* genome assembly contains 13 chromosome-scale scaffolds, which account for 97.23% of the total bases in the assembly. Panel shows the Hi-C contact map. **c.** The cladorhizid sponge individual used in the genome sequencing at its collection site. **d.** This sponge was bioluminescent when mechanically disturbed. **e.** Its mitochondrial sequence is 99.2% identical to the previously identified bioluminescent cladorhizid sponge[36]. **f.** The estimated genome size of this sponge is 1.11 Gb, and the spectrum is consistent with diploid organisms. **g.**,**h.** Each haplotype's genome assembly has 18 chromosome-scale scaffolds based on chromatin confirmation data as shown. In haplotype A 94.2% of bases are in the chromosome-scale scaffolds. **i.** A whole-genome alignment of haplotypes A and B showed a high degree of concordance. **j.** The hexactinellid sponge collected and sequenced for this study. **k.** The estimated genome size is 1n = 141 Mb. **l.** Haplotype A contains only one haplotype of chromosome-scale scaffolds orthologous with the scaffolds of the closely-related sponge *Oopsacas minuta*. Panel shows chromatin conformation capture contact map of haplotype A. **m.** In addition to the alternate haplotype of chromosome-scale scaffolds from haplotype A, the haplotype B assembly contains the large, gene-poor, unplaced scaffolds that lack detectable homology to other sponges. The Hi-C contact map for haplotype B shown. **n.** Whole-genome alignments of the two haplotypes show colinearity. Photograph credits: (**d.**) Darrin Schultz, (**c.**,**j.**) © 2021 MBARI.

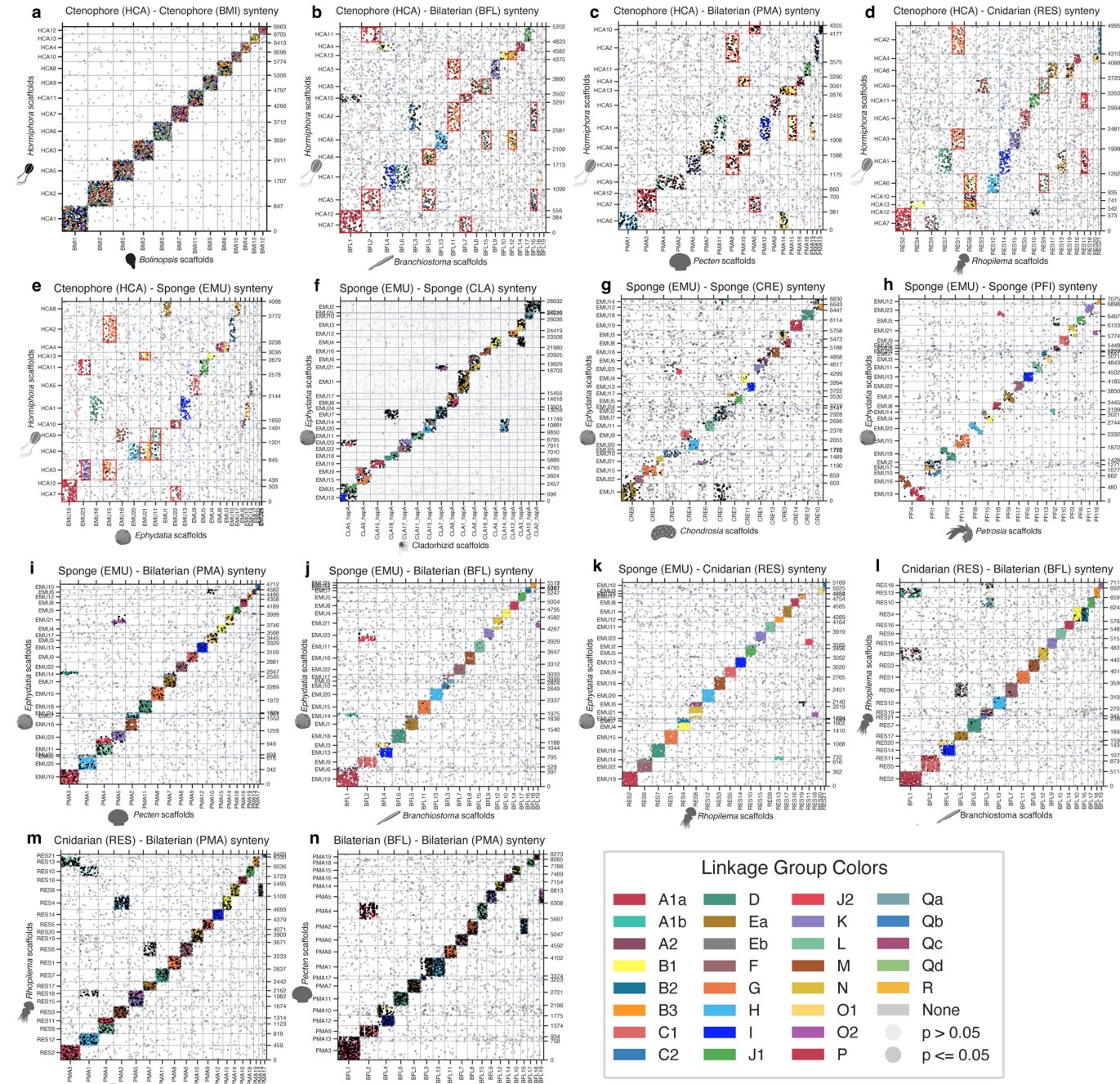

**Extended Data Fig. 2 | Chromosomes are largely conserved among metazoans.** The chromosome position of orthologous proteins plotted in panels **a-n** are coloured by orthologs in the previously identified ancestral bilaterian, cnidarian, and sponge linkage groups (BCnS-ALGs)[12]. Significant inter-species chromosome pairs (p ≤ 0.05, Bonferroni-corrected one-sided Fisher's exact test[57]) are opaque. **a.** The karyotype of the Pleurobrachiid and lobate ctenophores is conserved (1n = 13). **b.-e.** Ctenophore chromosomes share macrosynteny with BCnS-ALGs, but many BCnS ALGs are split onto several ctenophore chromosomes (red dotted boxes). There are many ctenophore-specific chromosome fusions. **f.-h.** Macrosynteny is highly conserved between distantly-related demosponges. The sponge lineages shown diverged an estimated 358 Mya - 500 Mya[38]. **f.-k.** Macrosynteny is also conserved between sponge, bilaterian, and cnidarian genomes. Many chromosomes in a species of one clade have a one-to-one homologous chromosome in the other clade. The genomes of species in these clades can be described by 29 constituent BCnS-ALGs[12].

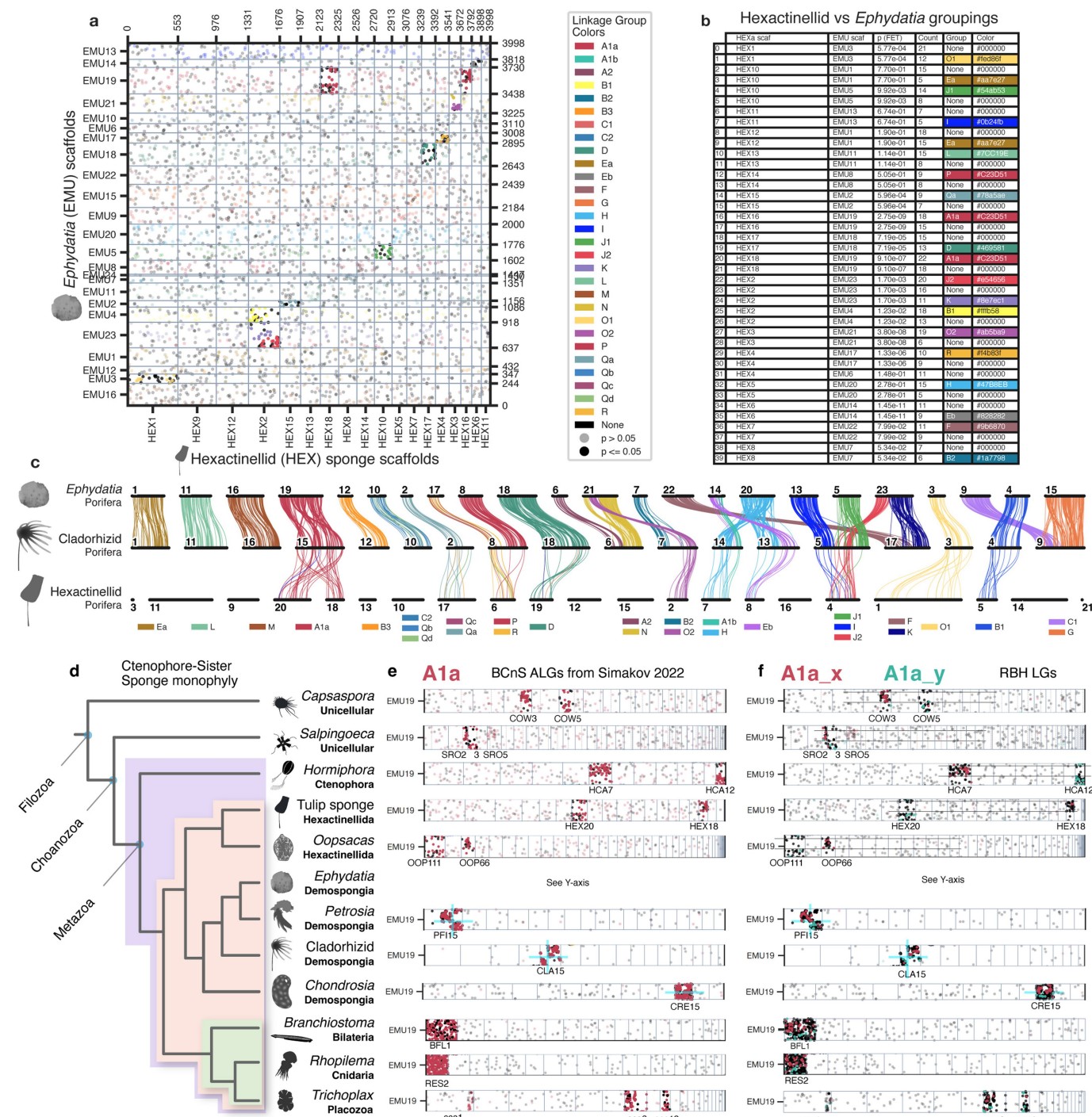

**Extended Data Fig. 3 | Sponge macrosynteny. a.-c.** There have been many genome rearrangements since the divergence of the demosponge *Ephydatia* and the tulip hexactinellid genome, and they share macrosynteny of only some BCnS linkage groups (p ≤ 0.05, Bonferroni-corrected one-sided Fisher's exact test[57], opaque dots in **a.**, rows in **b.**, interspecies lines in **c.**). **d.** The sponge cladogram is based on Schuster et al. 2018[38]. **e.** The orthologs in A1a_x and A1a_y are predominantly present on separate chromosomes in both the tulip hexactinellid and in *Oopsacas minuta*. **f.** A1a_x and A1a_y are on partly overlapping regions of single demosponge chromosomes, but are mixed on a single *Chondrosia* chromosome. However, the linkage groups A1a_x and A1a_y are on separate chromosomes in the ctenophores and the unicellular outgroup species. This evidence suggests that hexactinellid sponges retain the ancestral state of A1a_x and A1a_y being present on separate chromosomes. The possible evolutionary scenarios explaining this karyotype will require further chromosome-scale sequencing of sponge genomes.

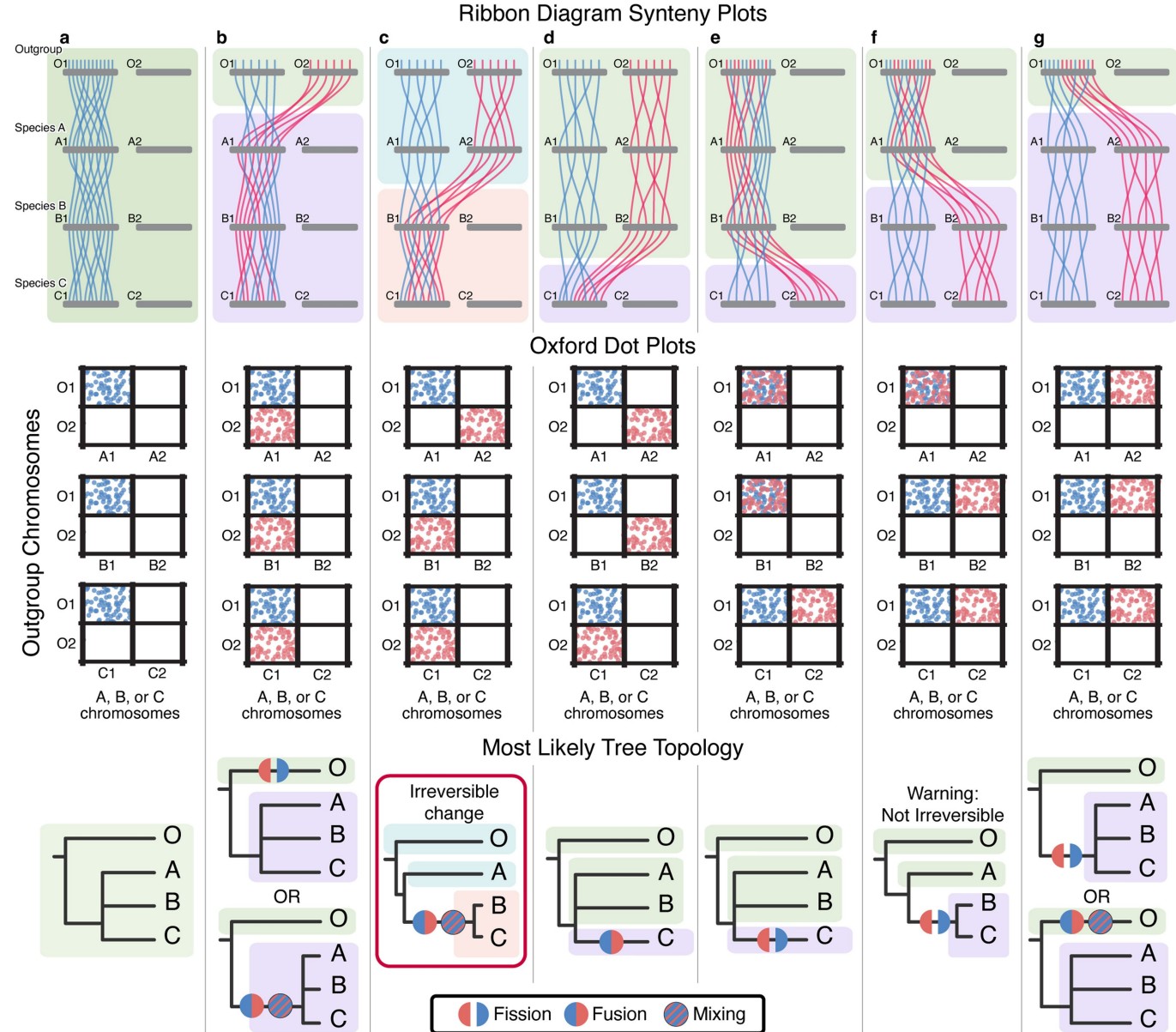

**Extended Data Fig. 4 | Seven basic ALG configurations in species quartets. a-g.** The seven configurations of ALGs found in four species highlight the evolutionary history of chromosomes. The cartoon ribbon plot in each panel shows chromosomes (horizontal bars) and the positions of genes in two ALGs along those chromosomes (vertical blue or red lines, respectively). The cartoon Oxford dot plot in each panel shows the same information as the ribbon plot, but only in the context of the outgroup genome. The most parsimonious tree topology based on the ALG evolutionary history is also pictured.

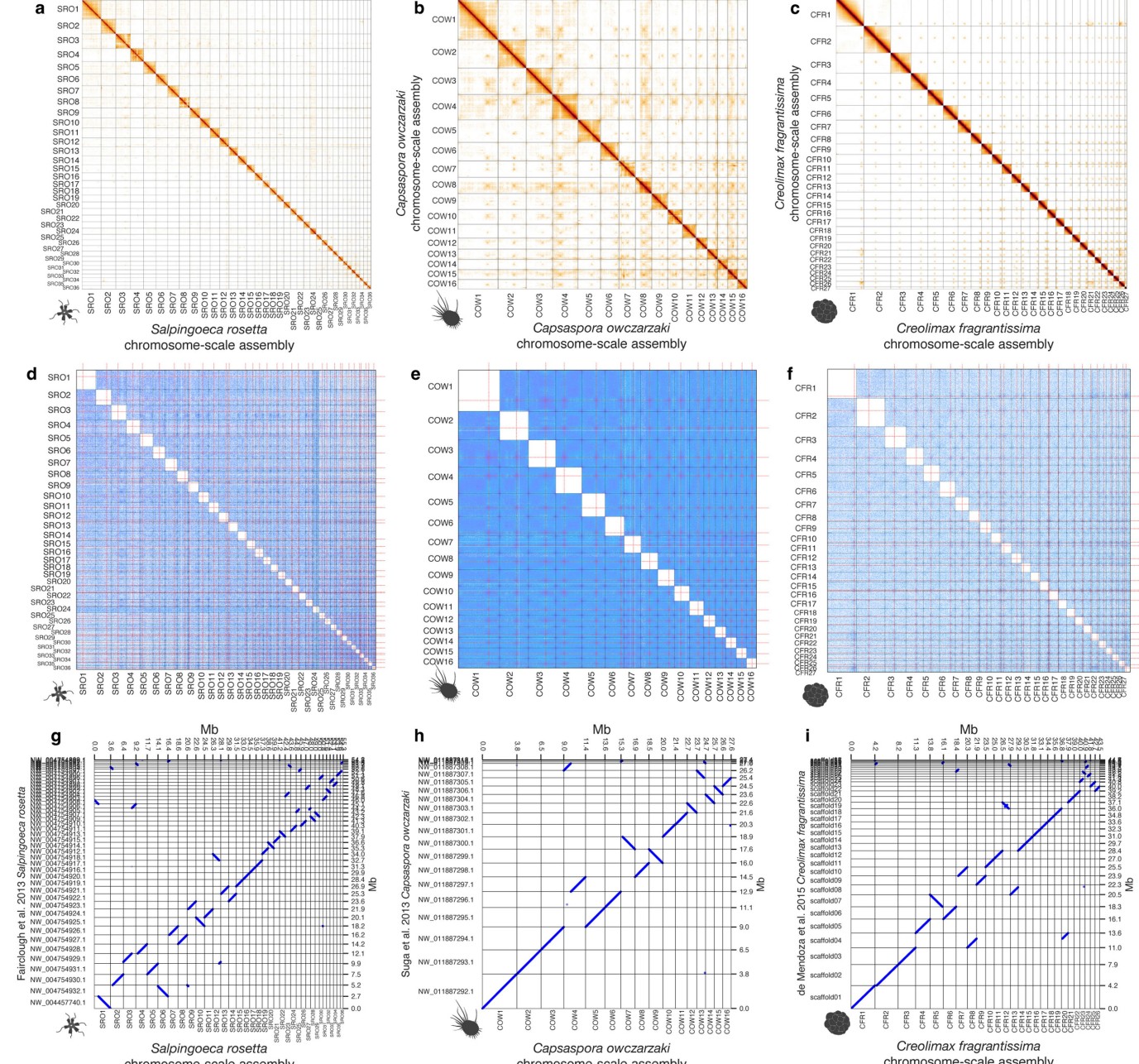

**Extended Data Fig. 5 | Unicellular species chromosome-scale genome assemblies.** Hi-C heatmaps of **a**. *Salpingoeca rosetta*, **b**. *Capsaspora owczarzaki*, and **c**. *Creolimax fragrantissima* show that the assemblies are consistent with chromosome-scale assemblies of other unicellular species. **d.-e**. Genome-wide ICE-normalized [108] observed count contact maps for (**d**.) *Salpingoeca*, (**e**.) *Capsaspora*, and (**f**.) *Creolimax* are shown at MapQ0 and 10 kb resolution. Chromosome boundaries are drawn as solid black lines. The intersections of horizontal and vertical red lines mark the Centurion-estimated centromere positions. The Hi-C heatmaps of *Capsaspora* and *Creolimax* both contain inter-chromosomal hotspots that are consistent with centromeres in other species. **g.-i**. Protein orthology plots (Oxford dot plots) of the chromosome-scale genome assemblies compared to the previously published assemblies. Despite the lack of Hi-C data, the original scaffold assemblies for all three species were nearly chromosome-scale.

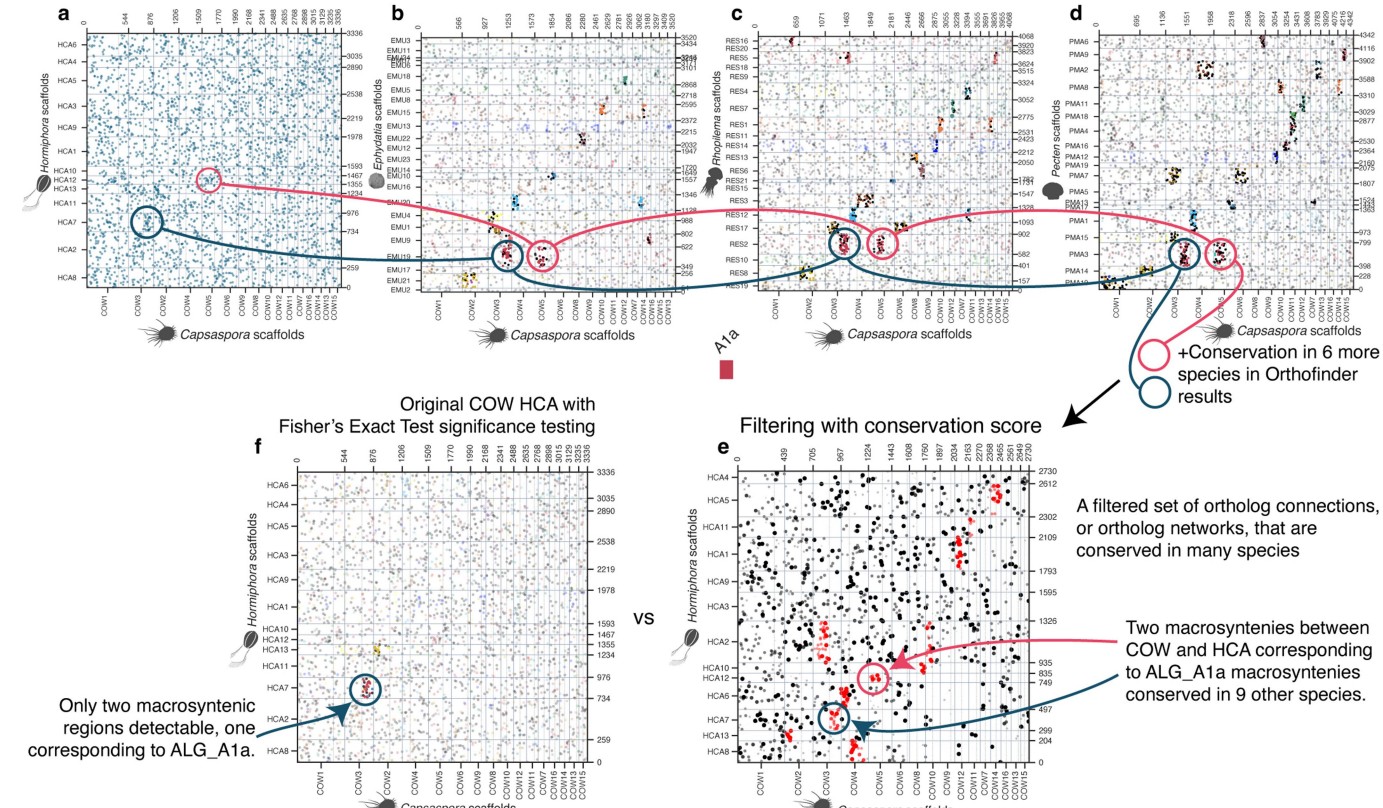

**Extended Data Fig. 6 | Visual representation of multi-species gene linkage conservation score. a.-d**. The dot plots of the *C. owczarzaki* genome show that there is conservation of the ALG_A1a linkage group in ctenophore, sponge, cnidarian, and bilaterian genomes. The conservation score can be calculated from shared gene linkages across many species. **f**. Due to the highly rearranged state of both the *Hormiphora* and *Capsaspora* genomes, a Bonferroni-corrected one-sided Fisher's exact test[57] only distinguishes three chromosome relationships as significant (p ≤ 0.05). **e**. Calculating the orthology conservation score for the relationships in these two genomes reveals more gene linkages that have been conserved across Filozoans. Red dots here are orthologs that are in significantly-conserved ortholog networks (α ≤ 0.05, permutation test). See complete results in Supplementary Information 11.

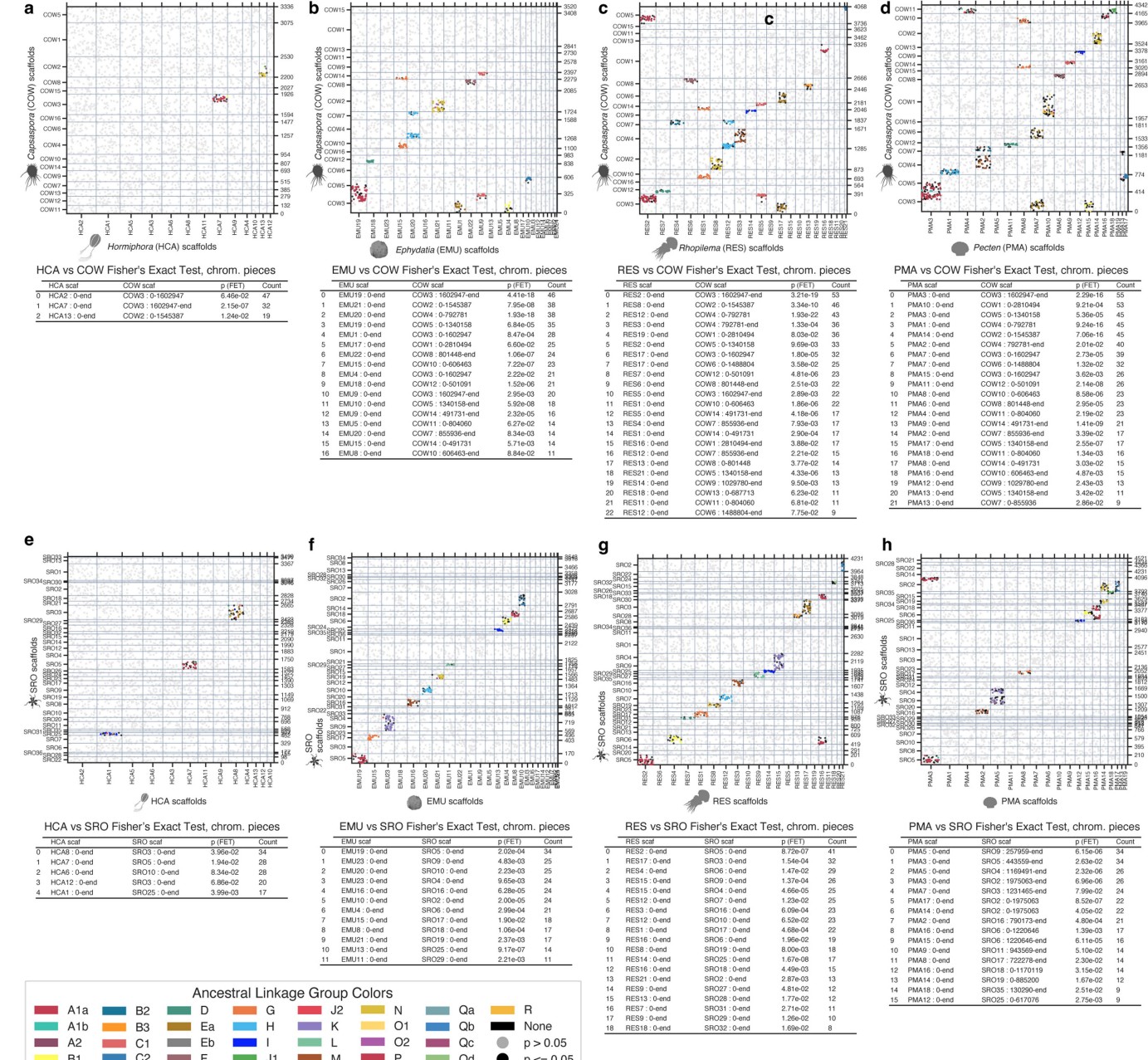

**Extended Data Fig. 7 | Filozoan and choanoflagellate genomes share macrosynteny with metazoans.** Two-way reciprocal best hits blast searches between the filasterean amoeba *Capsaspora* and animals (**a.-d**.), or between the choanoflagellate *Salpingoeca* and animals (**d.-g**.) show that the chromosomes of these unicellular species are rearranged relative to animal chromosomes, that some regions of synteny remain, and that some ALGs are split across multiple chromosomes of the unicellular species. Orthologs are coloured based on BCnS-ALGs from Simakov et al. 2022[12], and chromosome pairs with significantly-conserved macrosynteny (p ≤ 0.05, Bonferroni-corrected one-sided Fisher's exact test[57]) have opaque dots. Axis labels show cumulative number of orthologs. Putative centromeres are marked by dotted lines.

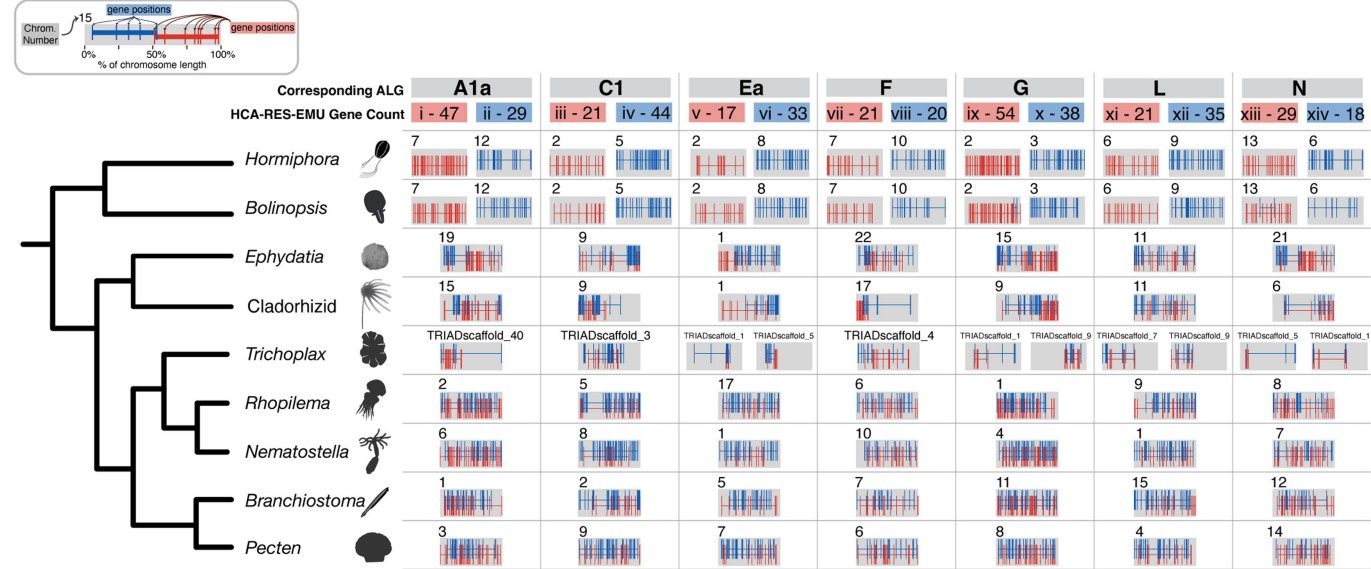

**Extended Data Fig. 8 | Mixing plots of HCA-EMU-RES reciprocal best blastp results.** This figure parallels Fig. 3 of the main text, but includes more genes by requiring orthology between metazoans without requiring orthologs in corresponding outgroups. Limiting the macrosynteny search to animals shows many genes participating in the extension of metazoan ALGs to the ctenophores. The _x and _y components of ALG_Ea and ALG_G are mixed and widely distributed across single sponge chromosomes, while the (COW/SRO)-HCA-EMU-RES results show no _x and _y overlap for ALG_Ea, and little overlap for ALG_G. We placed placozoans as the sister clade to cnidarians based on the findings of Simakov et al. 2022[12]. See also Supplementary Information 13.2.2.

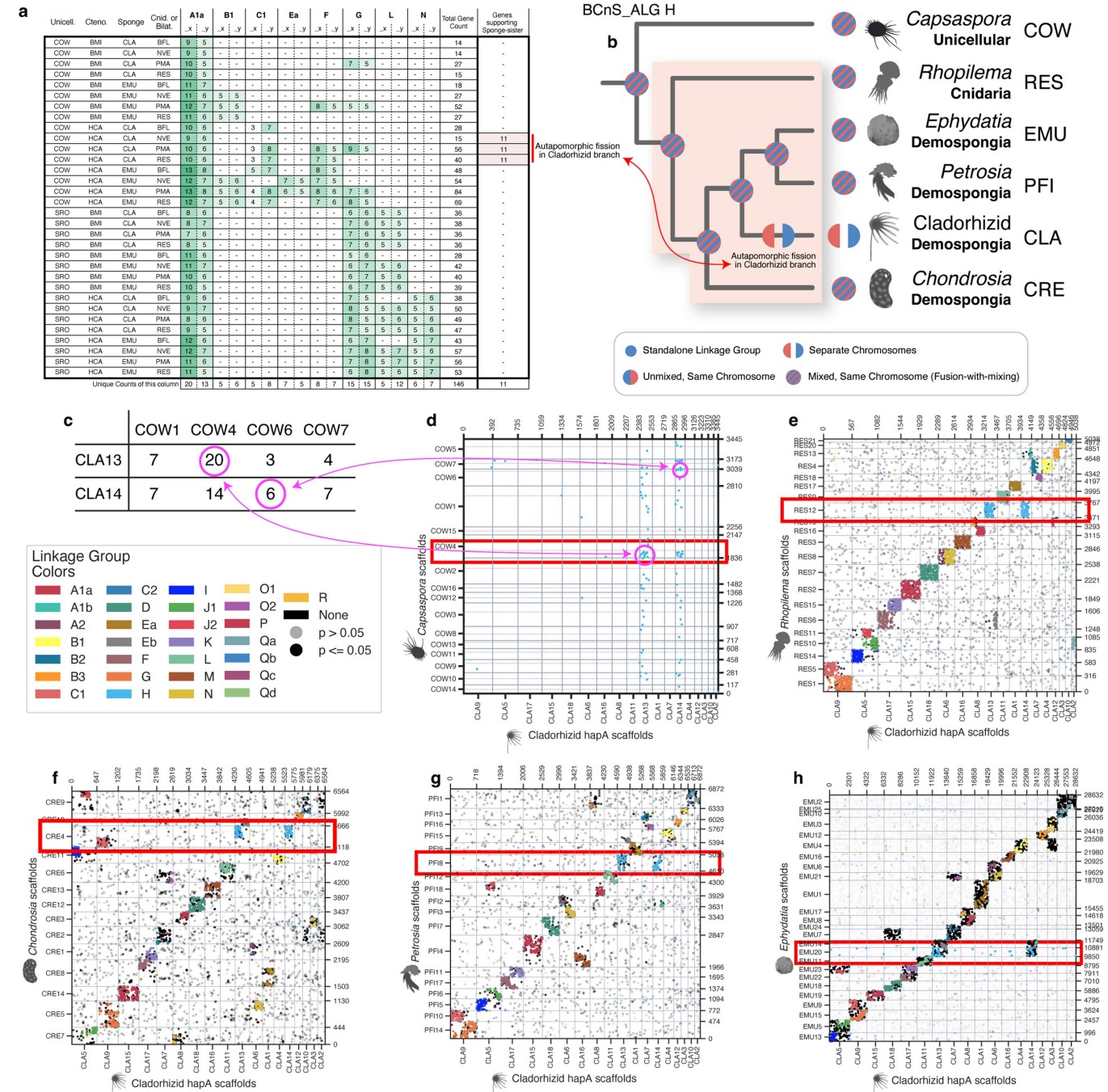

**Extended Data Fig. 9 | OrthoFinder results are consistent with the ctenophore-sister hypothesis. a.** Each green cell shows how many orthogroups support the ctenophore-sister hypothesis from each ALG in each species quartet. The Total Gene Count column is the total number of orthogroups supporting the ctenophore-sister hypothesis for that species quartet. The bottom row shows the number of unique orthogroups in each column. There are 146 orthogroups that support ctenophore-sister. **b.** The 11 orthologs that support CLA-sister in three analyses are due to a lineage-specific fission of ALG_H that is only found in the cladorhizid genome, but not in the genome of

other sponges. Tree topology based on previous studies[38,109]. **c.** The *Capsaspora*-cladorhizid chromosome pairs with the most genes from ALG_H (COW4-CLA13, COW4-CLA14) are not the chromosome pairs supporting sponge-sister (magenta circles, COW4-CLA13, COW6-CLA14). **d.-h.** The fission of ALG_H is specific to the cladorhizid sponge genome and is not found in the unicellular organism *Capsaspora* (COW), in other demosponges (EMU, CRE, PFI), in cnidarians (RES), or in bilaterians (not shown). Chromosome pairs that have significantly-conserved macrosynteny (p ≤ 0.05, Bonferroni-corrected one-sided Fisher's exact test[57]) have opaque dots.

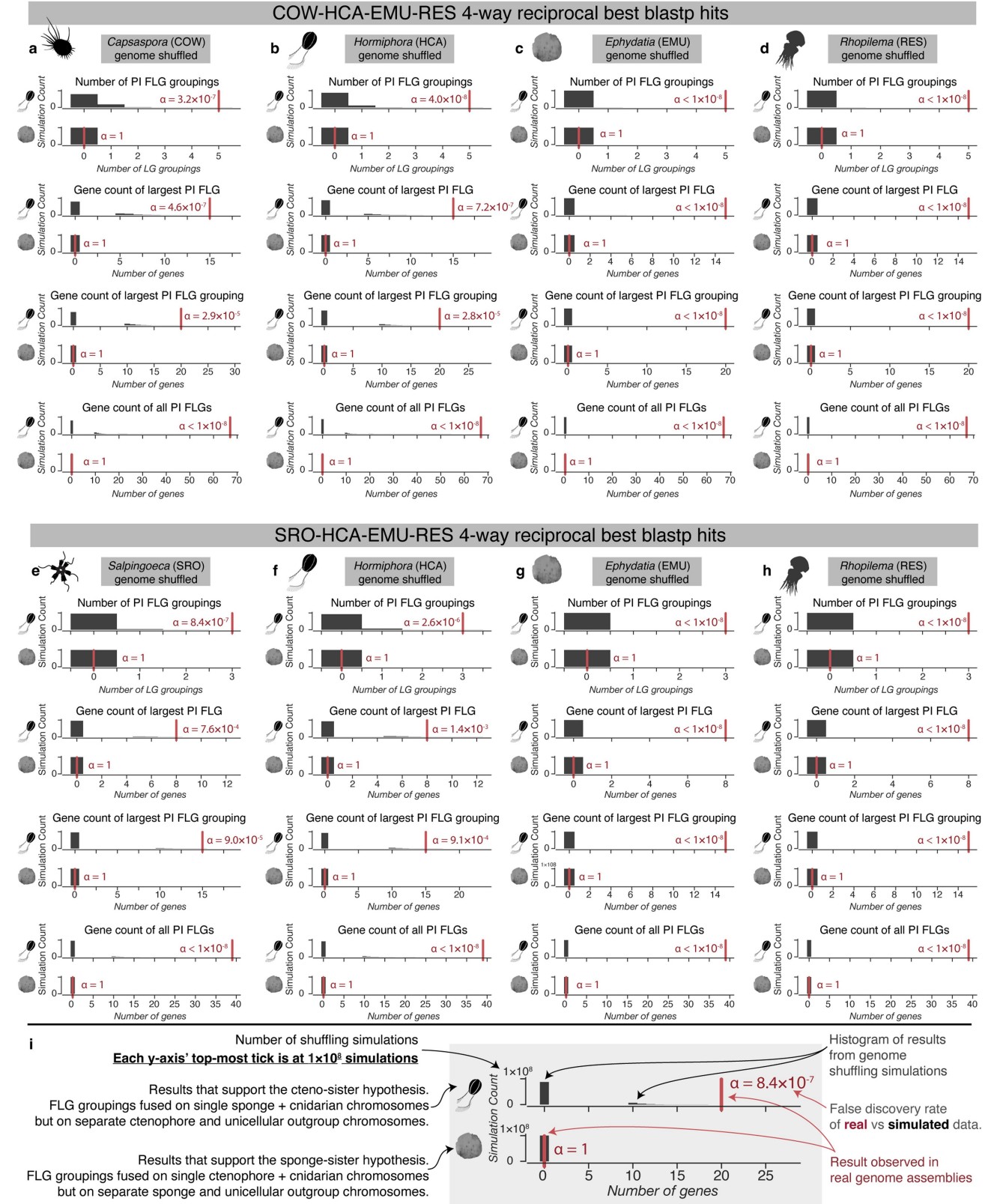

**Extended Data Fig. 10 | Results of genome shuffling simulations.**
**a.-d**. Shuffling one of the genomes before the COW-HCA-EMU-RES comparison shows that the rearranged state of the ctenophore genome, let alone the other species in the analysis, cannot explain the signal supporting the ctenophore-sister hypothesis (vertical red lines). **e.-h**. Shuffling simulations using SRO as the outgroup independently support the ctenophore-sister hypothesis. **i**. contains a legend to interpret panels a-h.

# Extended Data Table 1 | Linkage groups conserved in animals and unicellular outgroups

| | | | | Unicellular Outgroups | | | Ctenophora | | Porifera | | Placozoa | Cnidaria | | Bilateria | |
|---|---|---|---|---|---|---|---|---|---|---|---|---|---|---|---|
| Group Number | ALG | No. of Shared Orthologs | Creolimax | Capsaspora | Salpingoeca | Hormiphora | Bolinopsis | Ephydatia | cladorhizid | Trichoplax | Nematostella | Rhopilema | Branchiostoma | Pecten |
| **BCnS only** | | | | | | | | | | | | | | |
| - | A1b | 52 | N/A | N/A | N/A | N/A | N/A | EMU14R | CLA14 | complex | NVE2 | RES13 | BFL1 | PMA3 |
| - | Qabcd | - | N/A | N/A | N/A | N/A | N/A | complex | complex | complex | complex | complex | complex | complex |
| **Metazoan Only** | | | | | | | | | | | | | | |
| - | A2 | 10 | N/A | N/A | N/A | HCA10 | BMI10 | EMU6 | CLA6R | TRIADscaffold_5 | NVE7 | RES8 | BFL1 | PMA13 |
| - | O1 | 21 | N/A | N/A | N/A | HCA4 | BMI4 | EMU3 | CLA3 | complex | NVE5 | RES20 | BFL4L | PMA10 |
| - | R | 20 | N/A | N/A | N/A | HCA8 | BMI8 | EMU17 | CLA8R | TRIADscaffold_1 | NVE3 | RES19 | - | PMA10 |
| - | Eb | 12 | N/A | N/A | N/A | HCA5 | BMI5 | EMU14L | CLA13 | TRIADscaffold_4 | NVE10 | RES6 | BFL5 | PMA7 |
| - | O2 | 19 | N/A | N/A | N/A | | | EMU21mid | CLA7R | complex | NVE15 | RES18 | BFL19 | PMA5 |
| - | B2 | 14 | N/A | N/A | N/A | | | EMU7 | CLA7L | TRIADscaffold_15 / TRIADscaffold_2 | NVE2 | RES4 | BFL16 | PMA2 |
| - | B3 | 14 | N/A | N/A | N/A | HCA1 | BMI1 | EMU12 | CLA12 | TRIADscaffold_3 / TRIADscaffold_16 | | RES13 | BFL18 | PMA19 |
| **Single-Chromosome in Filozoa** | | | | | | | | | | | | | | |
| 1 | P | 6 | N/A | N/A | SRO6 | HCA4 | BMI4 | EMU8 | CLA8 | TRIADscaffold_6 | NVE3 | RES16 | BFL14 | PMA16 |
| 2 | C2 | 8 | N/A | COW5 | SRO2 | HCA2 | BMI2 | EMU10 | CLA10 | TRIADscaffold_5 | | RES21 | BFL3 | PMA17 |
| 3 | D | 15 | N/A | COW12 | SRO2 | HCA1 | BMI1 | EMU18 | CLA18 | TRIADscaffold_5 / TRIADscaffold_1 | NVE11 | RES7 | BFL6 | PMA11 |
| 4 | J1 | 5 | N/A | COW1 | N/A | HCA11 | BMI11 | EMU5 | CLA5 | TRIADscaffold_7 / TRIADscaffold_20 / TRIADscaffold_9 | NVE12 | RES10 | BFL17 | PMA18 |
| 5 | J2 | 7 | N/A | COW11 | N/A | | | EMU23 | | | NVE4 | RES11 | BFL2 | PMA4 |
| 6 | M | 10 | N/A | COW4 | SRO16 | HCA9 | BMI9 | EMU16 | CLA16 | TRIADscaffold_2 | NVE9 | RES3 | BFL8 | PMA2 |
| **Holozoan, Translocation-participating, not phylogenetically informative** | | | | | | | | | | | | | | |
| 7 | I_z1 | 11 | CFR1 CFR3 | COW4 COW9 COW13 | N/A | HCA1 | BMI1 | EMU13 | CLA5 | TRIADscaffold_8 / TRIADscaffold_18 / TRIADscaffold_23 / TRIADscaffold_27 | NVE5 | RES14 | BFL4 | PMA12 |
| 8 | I_z2 | 7 | CFR1 CFR3 | COW4 COW9 COW13 | SRO25 | | | | | | | | | |
| 9 | I_z3 | 6 | CFR1 CFR3 | COW4 COW9 COW13 | SRO3 | | | | | | | | | |
| 10 | K_z1 | 7 | N/A | COW4 | N/A | HCA3 | BMI3 | EMU23 | CLA17 | TRIADscaffold_4 / TRIADscaffold_17 | NVE14 | RES15 | BFL9 | PMA5 |
| 11 | K_z2 | 14 | CFR4 | COW6 | SRO9 | | | | | | | | | |
| 12 | B1_z1 | 6 | N/A | COW1 | SRO6 | HCA13 | BMI13 | EMU4 | CLA4 | TRIADscaffold_3 | NVE2 | RES4 | BFL10 | PMA15 |
| 13 | B1_z2 | 7 | N/A | COW3 | SRO6 | | | | | | | | | |
| 14 | C1_z1 | 6 | N/A | COW3 | N/A | HCA2 | BMI2 | EMU9 | CLA9 | | NVE8 | RES5 | BFL2 | PMA9 |
| 15 | C1_z2 | 8 | N/A | COW3 | (SRO1) | HCA5 | BMI5 | | | | | | | |
| 16 | H_z1 | 14 | CFR1 CFR3 | COW4 | SRO10 | HCA6 | BMI6 | EMU20 | CLA13 CLA14 | TRIADscaffold_2 | NVE13 | RES12 | BFL13 | PMA1 |
| 17 | H_z2 | 10 | CFR1 | COW6 | | | | | | | | | | |
| **Ctenophore and unicellular outgroups share ancestral state — with fusion on the BCnS stem** | | | | | | | | | | | | | | |
| 18 | A1a_x | 29 | CFR1 | COW3 | SRO2 SRO5 | HCA7 | BMI7 | EMU19 | CLA15 | TRIADscaffold_40 | NVE6 | RES2 | BFL1 | PMA3 |
| 19 | A1a_y | 11 | N/A | COW5 | SRO3 | HCA12 | BMI12 | | | | | | | |
| 20 | Ea_x | 8 | N/A | COW3 | N/A | HCA2 | BMI2 | EMU1 | CLA1 | N/A | NVE1 | RES17 | BFL5 | PMA7 |
| 21 | Ea_y | 5 | N/A | COW4 | N/A | HCA8 | BMI8 | | | N/A | | | | |
| 22 | G_x | 27 | CFR1 CFR2 | COW10 | SRO4 | HCA2 | BMI2 | EMU15 | CLA9 | TRIADscaffold_9 | NVE4 | RES1 | BFL11 | PMA8 |
| 23 | G_y | 10 | N/A | COW12 | SRO17 | HCA3 | BMI3 | | | | | | | |
| **Ctenophore and unicellular outgroups share ancestral state — with fusion and mixing on the BCnS stem** | | | | | | | | | | | | | | |
| 24 | C1_x | 5 | N/A | COW1 | N/A | HCA2 | BMI2 | EMU9 | CLA9 | TRIADscaffold_3 | NVE8 | RES5 | BFL2 | PMA9 |
| 25 | C1_y | 8 | N/A | COW14 | SRO1 | HCA5 | BMI5 | | | | | | | |
| 26 | F_x | 5 | N/A | COW1 | N/A | HCA7 | BMI7 | EMU22 | CLA17 | N/A | NVE10 | RES6 | BFL7 | PMA6 |
| 27 | F_y | 7 | N/A | COW8 | SRO1 | HCA10 | BMI10 | | | N/A | | | | |
| 28 | L_x | 5 | N/A | N/A | SRO9 | HCA6 | BMI6 | EMU11 | CLA11 | TRIADscaffold_7 / TRIADscaffold_9 | NVE1 | RES9 | BFL15 | PMA4 |
| 29 | L_y | 13 | N/A | COW1 | SRO27 | HCA9 | BMI9 | | | | | | | |
| 30 | N_x | 9 | N/A | COW2 | SRO2 | HCA13 | BMI13 | EMU21 | CLA6 | TRIADscaffold_4 / TRIADscaffold_1 / TRIADscaffold_11 | NVE7 | RES8 | BFL12 | PMA14 |
| 31 | N_y | 14 | N/A | COW2 | SRO19 | HCA6 | BMI6 | | | | | | | |

Row-group labels (left margin): The BCnS only and Metazoan Only rows are "From EMU-RES-BFL or HCA-RES-EMU searches." Rows 1–31 are "From (CFR, COW, SRO)-HCA-EMU-RES searches."

The gene linkage groups only found in BCnS or Metazoans (group number "-"), and the merged OG-metazoan four-way reciprocal best blastp results (Group Number 1 through 31). Fusions-with-mixing events in the ancestor of the Choanozoa, or the ancestor of the Metazoa, or the ancestor of the BCnS clade are represented by rows of different colours, joined by striped cells. There is evidence for four fusion-with-mixing events uniting sponges, cnidarians, placozoans, and bilaterians, to the exclusion of ctenophores and unicellular OGs.

**Extended Data Table 2 | Ancestral linkage groups found by four-way outgroup-animal comparisons**

**a**

| ALG | No. of Shared Orthologs | *C. owczarzaki* **Unicell** COW | *H. californensis* **Ctenophora** HCA | *E. muelleri* **Porifera** EMU | *R. esculentum* **Cnidaria** RES | False Discovery Rate |
|---|---|---|---|---|---|---|
| A1a_x | 15 | COW3 | HCA7 | EMU19 | RES2 | $\alpha < 1.0e{-}07$ |
| A1a_y | 5 | COW5 | HCA12 | EMU19 | RES2 | $\alpha = 3.2e{-}04$ |
| C1_x | 5 | COW1 | HCA2 | EMU9 | RES5 | $\alpha = 3.2e{-}04$ |
| C1_y | 5 | COW14 | HCA5 | EMU9 | RES5 | $\alpha = 3.2e{-}04$ |
| Ea_x | 8 | COW3 | HCA2 | EMU1 | RES17 | $\alpha < 1.0e{-}07$ |
| Ea_y | 5 | COW4 | HCA8 | EMU1 | RES17 | $\alpha = 3.2e{-}04$ |
| F_x | 5 | COW1 | HCA7 | EMU22 | RES6 | $\alpha = 3.2e{-}04$ |
| F_y | 5 | COW8 | HCA10 | EMU22 | RES6 | $\alpha = 3.2e{-}04$ |
| G_x | 8 | COW10 | HCA2 | EMU15 | RES1 | $\alpha < 1.0e{-}07$ |
| G_y | 6 | COW12 | HCA3 | EMU15 | RES1 | $\alpha = 7.1e{-}06$ |
| N_x | 7 | COW2 | HCA6 | EMU21 | RES8 | $\alpha = 1.0e{-}07$ |
| N_y | 7 | COW2 | HCA13 | EMU21 | RES8 | $\alpha = 1.0e{-}07$ |
| C1_z1 | 6 | COW3 | HCA2 | EMU9 | RES5 | $\alpha = 7.1e{-}06$ |
| C1_z2 | 8 | COW3 | HCA5 | EMU9 | RES5 | $\alpha < 1.0e{-}07$ |
| B1_z1 | 7 | COW3 | HCA13 | EMU4 | RES4 | $\alpha = 1.0e{-}07$ |
| B1_z2 | 6 | COW1 | HCA13 | EMU4 | RES4 | $\alpha = 7.1e{-}06$ |
| H_z1 | 10 | COW4 | HCA6 | EMU20 | RES12 | $\alpha < 1.0e{-}07$ |
| H_z2 | 5 | COW6 | HCA6 | EMU20 | RES12 | $\alpha = 3.2e{-}04$ |
| I_z1,2,3 | 5 | COW13 | HCA1 | EMU13 | RES14 | $\alpha = 3.2e{-}04$ |
| I_z1,2,3 | 6 | COW4 | HCA1 | EMU13 | RES14 | $\alpha = 7.1e{-}06$ |
| I_z1,2,3 | 6 | COW9 | HCA1 | EMU13 | RES14 | $\alpha = 7.1e{-}06$ |
| K_z1 | 7 | COW4 | HCA3 | EMU23 | RES15 | $\alpha = 1.0e{-}07$ |
| K_z2 | 8 | COW6 | HCA3 | EMU23 | RES15 | $\alpha < 1.0e{-}07$ |
| C2 | 5 | COW5 | HCA2 | EMU10 | RES21 | $\alpha = 3.2e{-}04$ |
| D | 11 | COW12 | HCA1 | EMU18 | RES7 | $\alpha < 1.0e{-}07$ |
| J1 | 5 | COW1 | HCA11 | EMU5 | RES10 | $\alpha = 3.2e{-}04$ |
| J2 | 7 | COW11 | HCA11 | EMU23 | RES11 | $\alpha = 1.0e{-}07$ |
| L_y | 8 | COW1 | HCA9 | EMU11 | RES9 | $\alpha < 1.0e{-}07$ |
| M | 6 | COW4 | HCA9 | EMU16 | RES3 | $\alpha = 7.1e{-}06$ |

**b**

| ALG | No. of Shared Orthologs | *S. rosetta* **Unicell** SRO | *H. californensis* **Ctenophora** HCA | *E. muelleri* **Porifera** EMU | *R. esculentum* **Cnidaria** RES | False Discovery Rate |
|---|---|---|---|---|---|---|
| A1a_x | 6 | SRO2 | HCA7 | EMU19 | RES2 | $\alpha = 2.0e{-}07$ |
| A1a_x | 11 | SRO5 | HCA7 | EMU19 | RES2 | $\alpha < 2.0e{-}07$ |
| A1a_y | 7 | SRO3 | HCA12 | EMU19 | RES2 | $\alpha < 2.0e{-}07$ |
| G_x | 8 | SRO4 | HCA2 | EMU15 | RES1 | $\alpha < 2.0e{-}07$ |
| G_y | 7 | SRO17 | HCA3 | EMU15 | RES1 | $\alpha < 2.0e{-}07$ |
| L_x | 5 | SRO9 | HCA6 | EMU11 | RES9 | $\alpha = 2.3e{-}05$ |
| L_y | 6 | SRO27 | HCA9 | EMU11 | RES9 | $\alpha = 2.0e{-}07$ |
| N_x | 7 | SRO19 | HCA6 | EMU21 | RES8 | $\alpha < 2.0e{-}07$ |
| N_y | 6 | SRO2 | HCA13 | EMU21 | RES8 | $\alpha = 2.0e{-}07$ |
| I_z2 | 6 | SRO3 | HCA1 | EMU13 | RES14 | $\alpha = 2.0e{-}07$ |
| I_z3 | 7 | SRO25 | HCA1 | EMU13 | RES14 | $\alpha < 2.0e{-}07$ |
| B1_z1,2 | 5 | SRO6 | HCA13 | EMU4 | RES4 | $\alpha = 2.3e{-}05$ |
| C1 | 6 | SRO1 | HCA5 | EMU9 | RES5 | $\alpha = 2.0e{-}07$ |
| C2 | 7 | SRO2 | HCA2 | EMU10 | RES21 | $\alpha < 2.0e{-}07$ |
| D | 10 | SRO2 | HCA1 | EMU18 | RES7 | $\alpha < 2.0e{-}07$ |
| F | 5 | SRO1 | HCA10 | EMU22 | RES6 | $\alpha = 2.3e{-}05$ |
| H | 9 | SRO10 | HCA6 | EMU20 | RES12 | $\alpha < 2.0e{-}07$ |
| K | 7 | SRO9 | HCA3 | EMU23 | RES15 | $\alpha < 2.0e{-}07$ |
| M | 6 | SRO16 | HCA9 | EMU16 | RES3 | $\alpha = 2.0e{-}07$ |
| P | 6 | SRO6 | HCA4 | EMU8 | RES16 | $\alpha = 2.0e{-}07$ |

**c**

| ALG | No. of Shared Orthologs | *C. fragran.* **Unicell** CFR | *H. californensis* **Ctenophora** HCA | *E. muelleri* **Porifera** EMU | *R. esculentum* **Cnidaria** RES | False Discovery Rate |
|---|---|---|---|---|---|---|
| G_x | 5 | CFR1 | HCA2 | EMU15 | RES1 | $\alpha = 8.0e{-}05$ |
| G_x | 8 | CFR2 | HCA2 | EMU15 | RES1 | $\alpha < 4.0e{-}08$ |
| H_z1,2 | 5 | CFR1 | HCA6 | EMU20 | RES12 | $\alpha = 8.0e{-}05$ |
| H_z1 | 5 | CFR3 | HCA6 | EMU20 | RES12 | $\alpha = 8.0e{-}05$ |
| I_z1,2,3 | 5 | CFR1 | HCA1 | EMU13 | RES14 | $\alpha = 8.0e{-}05$ |
| I_z1,2,3 | 7 | CFR3 | HCA1 | EMU13 | RES14 | $\alpha = 4.0e{-}08$ |
| K_z2 | 5 | CFR4 | HCA3 | EMU23 | RES15 | $\alpha = 8.0e{-}05$ |
| A1a_x | 7 | CFR1 | HCA7 | EMU19 | RES2 | $\alpha = 4.0e{-}08$ |

Each row represents a conserved syntenic group, with false discovery rate (**a**,**b**. $n = 1 \times 10^7$, **c**. $n = 1 \times 10^8$) estimated by comparison with genome shuffling simulations (Suppl. Information 8). We use three-letter code for species in each column. **a.** The COW-HCA-EMU-RES search yielded 10 gene groups where *Hormiphora* shares an ancestral partitioned state with *Capsaspora*, but are fused onto 5 chromosomes in the sponge and cnidarian. Part of ALG C1 appears to have a derived split in ctenophores. Linkage groups corresponding to ALGs B1, H, I, and K appear to have each become established on single chromosomes by time of the common ancestor of metazoans. In this comparison, there is no evidence that ALGs C2, D, J1, J2, and M do not participate in any clade-specific fusions that are informative to cteno- vs sponge-sister. **b.** The SRO-HCA-EMU-RES search yielded nine gene groups where *Hormiphora* shares an ancestral partitioned state with *Salpingoeca*, but are fused onto four chromosomes in the sponge and cnidarian. There is no evidence for ctenophore-derived splits in this search. Linkage groups corresponding to ALG_I appear to have merged onto single chromosomes by the common ancestor of metazoans. This table suggests that ALGs B1, C1, C2, D, F, H, K, M and P do not participate in any clade-specific fusions that are informative to cteno- vs sponge-sister. **c.** A comparison of *Creolimax fragrantissima* to HCA-EMU-RES shows limited conservation of gene linkages between animals and *Creolimax*, an ichthyosporean.

Oleg Simakov
Daniel S. Rokhsar

# Reporting Summary

## Statistics

For all statistical analyses, confirm that the following items are present in the figure legend, table legend, main text, or Methods section.

| n/a | Confirmed | |
|---|---|---|
| ☐ | ☒ | The exact sample size (*n*) for each experimental group/condition, given as a discrete number and unit of measurement |
| ☐ | ☒ | A statement on whether measurements were taken from distinct samples or whether the same sample was measured repeatedly |
| ☐ | ☒ | The statistical test(s) used AND whether they are one- or two-sided<br>*Only common tests should be described solely by name; describe more complex techniques in the Methods section.* |
| ☐ | ☒ | A description of all covariates tested |
| ☐ | ☒ | A description of any assumptions or corrections, such as tests of normality and adjustment for multiple comparisons |
| ☐ | ☒ | A full description of the statistical parameters including central tendency (e.g. means) or other basic estimates (e.g. regression coefficient) AND variation (e.g. standard deviation) or associated estimates of uncertainty (e.g. confidence intervals) |
| ☐ | ☒ | For null hypothesis testing, the test statistic (e.g. *F*, *t*, *r*) with confidence intervals, effect sizes, degrees of freedom and *P* value noted<br>*Give P values as exact values whenever suitable.* |
| ☐ | ☒ | For Bayesian analysis, information on the choice of priors and Markov chain Monte Carlo settings |
| ☒ | ☐ | For hierarchical and complex designs, identification of the appropriate level for tests and full reporting of outcomes |
| ☒ | ☐ | Estimates of effect sizes (e.g. Cohen's *d*, Pearson's *r*), indicating how they were calculated |

*Our web collection on statistics for biologists contains articles on many of the points above.*

## Software and code

Policy information about availability of computer code

| Data collection | The software developed for this manuscript is available at https://github.com/conchoecia/odp and at https://doi.org/10.5061/dryad.dncjsxm47. |
|---|---|
| Data analysis | The software packages used to analyze data in this manuscript were: odp v0.2.0 and v0.3.0, HiRise vAug2019, SALSA2 v2.3, wtdbg v2.4, hifiasm v0.16.1-r375, TGS-Gapcloser v1.1.1, Purge Haplotigs v1.0.4, pilon v1.23, Diamond v0.9.24, Blobtools v1.0, minimap2 v2.17 and v2.23, tblastn v2.10.0+, miniprot v0.2, BRAKER v2.14, , STAR v2.7.1a, ProtHint v2.6.0, Trinity v2.5.1, OrthoFinder v2.3.7 , hmmer v3.3.2, PANTHER v17, snakemake v7, BUSCO v5, lima v2.2.0, isoseq3 v3.4.0, jellyfish v2.2.10, GenomeScope 2, bwa mem v0.7.17, PretextView v0.2.4, HiGlass v1.10.0104, Juicebox Assembly Tools github commit 46c7ed1105, Juicebox visualization system v1.11.08106, TransDecoder v5.5, D-genies v1.4.0, pairtools v0.3.0 pairix v0.3.7, Cooler v0.8.10, MAFFT v7.310, RevBayes version 1.1.1, MrBayes version 3.2.7a, and FigTree v1.4.4. |

For manuscripts utilizing custom algorithms or software that are central to the research but not yet described in published literature, software must be made available to editors and reviewers. We strongly encourage code deposition in a community repository (e.g. GitHub). See the Nature Portfolio guidelines for submitting code & software for further information.

## Data

The authors confirm that all data presented in this manuscript are available in public repositories. The sequencing reads are available in the NCBI database under BioProject accession numbers PRJNA818620, PRJNA818630, PRJNA903214, and PRJNA818537. The genomes for each species are available through the above BioProjects, with the exception that the genomes of C. fragrantissima, C. owczarzaki, and S. rosetta, which are available on Dryad: https://doi.org/10.5061/dryad.dncjsxm47. The scripts and results of the supplementary information, when not contained in figures, are also available in the aforementioned Dryad repository. Publicly available sequencing data and genomes were downloaded from NCBI from BioProject accession numbers PRJNA168, PRJDB8655, PRJNA12874, PRJNA20249, PRJNA20341, PRJEB28334, PRJNA30931, PRJNA31257, PRJNA37927, PRJEB56075, PRJEB56892, PRJNA64405, PRJNA193541, PRJNA193613, PRJNA213480, PRJNA278284, PRJNA281977, PRJNA377365, PRJNA396415, PRJNA512552, PRJNA544471, PRJNA576068, PRJNA579531, PRJNA625562, PRJNA667495, PRJNA761294, and PRJNA814716. The Ephydatia muelleri genome was downloaded from https://spaces.facsci.ualberta.ca/ephybase/.  All of the above information is also included in the main text of the manuscript.

## Human research participants

| | |
|---|---|
| Reporting on sex and gender | N/A |
| Population characteristics | N/A |
| Recruitment | N/A |
| Ethics oversight | N/A |

Note that full information on the approval of the study protocol must also be provided in the manuscript.

# Field-specific reporting

Please select the one below that is the best fit for your research. If you are not sure, read the appropriate sections before making your selection.

☒ Life sciences          ☐ Behavioural & social sciences          ☐ Ecological, evolutionary & environmental sciences

For a reference copy of the document with all sections, see nature.com/documents/nr-reporting-summary-flat.pdf

# Life sciences study design

All studies must disclose on these points even when the disclosure is negative.

| | |
|---|---|
| Sample size | These categories are not applicable to this study, which involves analysis of genome sequences and chromosome architecture. Sample sizes of permutation tests and the resulting statistics are included in the manuscript. |
| Data exclusions | DNA sequences were filtered (for quality) during the assembly process as described in the methods. |
| Replication | Our findings were replicated by (1) performing many analyses using combinations of different species, and (2) performing the same range of analyses using multiple orthology inference techniques. Otherwise, because this was a comparative genomics study using replicates of the same was not relevant. |
| Randomization | Randomization is not relevant to this study, as the genomes used in comparisons are selected at the point of experimental design. It is not possible to randomize these selections as we are dependent on the phylogenetic relationships between the species in the comparisons to draw conclusions. |
| Blinding | As above, the conclusions of this comparative genomics study were dependent on selecting groups of species with particular phylogenetic relationships. Therefore, blinding was not relevant to this study. |

# Reporting for specific materials, systems and methods

We require information from authors about some types of materials, experimental systems and methods used in many studies. Here, indicate whether each material, system or method listed is relevant to your study. If you are not sure if a list item applies to your research, read the appropriate section before selecting a response.

## Materials & experimental systems

| n/a | Involved in the study |
|-----|----------------------|
| ☒ | ☐ Antibodies |
| ☒ | ☐ Eukaryotic cell lines |
| ☒ | ☐ Palaeontology and archaeology |
| ☐ | ☒ Animals and other organisms |
| ☒ | ☐ Clinical data |
| ☒ | ☐ Dual use research of concern |

## Methods

| n/a | Involved in the study |
|-----|----------------------|
| ☒ | ☐ ChIP-seq |
| ☒ | ☐ Flow cytometry |
| ☒ | ☐ MRI-based neuroimaging |

## Animals and other research organisms

Policy information about studies involving animals; ARRIVE guidelines recommended for reporting animal research, and Sex and Gender in Research

**Laboratory animals**
No laboratory animals were used in this study.

**Wild animals**
Samples of adult Bolinopsis microptera were collected on May 24th, 2015 in the Monterey Bay, California (36.63°N, 121.90°W) with jars from the surface waters, with permission under the State of California Department of Fish and Wildlife collecting permit SC-2026 to the Monterey Bay Aquarium. These samples were transported to the Monterey Bay Aquarium for rearing.

One individual of an undescribed hexactinellid sponge was collected on June 1st, 2021, in the Monterey Bay, California (34.57°N, 122.56°W) from the seafloor at 3,852 meters depth using the MBARI ROV Doc Ricketts aboard the R/V Western Flyer. On the following day one individual of an undescribed bioluminescent cladorhizid sponge was collected from a nearby site (35.49°N, 124°W) from the seafloor at 3,975 meters depth. The collection temperature of both samples was 1.5°C. The cladorhizid sample was consistent in morphology and locale with previously reported bioluminescent, carnivorous, cladorhizid sponges (1). Upon retrieval from the ROV, the samples were washed gently with 1°C filtered seawater to remove debris and maintained in the dark at 1°C for no more than 30 minutes after being collected from the ROV. Then, both samples were flash-frozen in liquid nitrogen. The samples were collected with the State of California Department of Fish and Wildlife collecting permit SC-4029 granted to the Haddock Laboratory at the Monterey Bay Aquarium Research Institute.

(1) Martini, S., Schultz, D. T., Lundsten, L. & Haddock, S. H. D. Bioluminescence in an Undescribed Species of Carnivorous Sponge (Cladorhizidae) From the Deep Sea. Front. Mar. Sci. 7, 576476 (2020).

**Reporting on sex**
N/A. The organisms sequenced in this study are hermaphrodites, or the biology of the species' sex is unknown.

**Field-collected samples**
A community culture was founded with 20 Bolinopsis microptera individuals in pseudokreisel tanks and diffusion tubes in 12°C seawater at the Monterey Bay Aquarium in Monterey, California. The culture was reared according to the published protocol (1) for three generations, and an F3 adult, called Bmic1, was selected and flash-frozen in liquid nitrogen on November 18th, 2019 for DNA sequencing for genome assembly and annotation. Four other F3 adults were placed into a spawning tank and spawned according to the published protocol(1). Fertilized eggs were collected 18 hours post-spawning for RNA sequencing.

(1) Patry, W. L., Bubel, M., Hansen, C. & Knowles, T. Diffusion tubes: a method for the mass culture of ctenophores and other pelagic marine invertebrates. PeerJ 8, e8938 (2020).

**Ethics oversight**
No vertebrates or cephalopods were used in the study, and the organisms included in this study are unrestricted other than the collecting permits listed above.

Note that full information on the approval of the study protocol must also be provided in the manuscript.

