## [Peer Review File · Nature]

Manuscript Title: Ancient gene linkages support ctenophores as sister to other animals

Reviewer Comments & Author Rebuttals

Reviewer Reports on the Initial Version:

Referee expertise:

Referee #1: evolutionary genetics

Referee #2: evolutionary genetics, phylogenomics

Referee #3: phylogenomics

Referees' comments:

Referee #1 (Remarks to the Author):

I enjoyed reading the manuscript titled “Ancient gene linkages show that ctenophores are sister to other animals” by Schultz and collaborators. The manuscript is a data and analytical tour de force, producing novel chromosome-level genomes for early diverging animals and their closest relatives, which are analysed together with other previously sequenced high-quality genomes. For the first time in deep metazoan phylogeny, the manuscript exploits the shared genome-wide chromosomal location of genes (synteny but without collinearity) as a phylogenetic marker. The analyses aim to resolve the contentious issue of the evolutionary position of sponges vs ctenophores as first-splitting animals.

The manuscript claims that syntenic regions support the ctenophores-first hypothesis, based on two required conditions: 1) syntenic blocks shared by bilaterians + cnidarians + sponges (BCnS) and 2) synteny shared by ctenophores and the single-celled outgroups. The analyses are original and sound (except for the homology identification approach, more on that later), and of the quality usually produced by the labs involved in the manuscript. The text is clear and well-written, although there are substantial mismatches between the figure panels invoked in the text and the actual figures, making an already complex manuscript harder to follow. I want to congratulate the authors for their efforts in developing a new approach to a question that has been (over)analysed with classic phylogenomic methods — without being resolved — for almost 15 years.

However, while the use of synteny as a phylogenetic marker is truly novel and very interesting, I am afraid its application to this particular phylogenetic question is not conclusive. I believe that the results (or their interpretation) do not support the main claim of the manuscript and its title.

First, while the text shows compelling proof of the first condition (shared synteny between BCnS), it does not show a strong signal supporting the second one (shared synteny between ctenophores and the metazoan outgroups). Both conditions need to be fulfilled to support ctenophores-first, otherwise, syntenically apomorphic ctenophores could still be nested within a BCnS with sponges-

first. The low number of events in Figure 3 is hard to assess from an evolutionary point of view, and might be biased by methodological issues outlined below. Supp Figures 4.2 to 4.8, which include the comparisons of the ctenophores against the outgroups, don't provide evidence of shared synteny between ctenophores and the outgroup. I suspect the manuscript is self-aware of that because the main text goes a great length to show various pairwise dot plots between members of the BCnS or between the metazoan outgroups and the BCnS, but it never shows any between the ctenophore and the outgroups which are presented in the Supp Data. Given that the sharing of syntenic blocks between ctenophores and the outgroups is one of the main conditions to support ctenophores-first, one would expect these analyses to be featured front and centre in the main text rather than a comparison between two ctenophores or two sponges.

Second, the manuscript argues that the formation of these synteny blocks is the result of "rare, irreversible chromosome fusion-and-mixing events". However, it is not clear if these blocks can be lost. Or even if they may remerge by convergent evolution: if evolution keeps these genes generally together (synteny without collinearity) it means that there are selective pressures to keep genes together may be due to similar biological functions or because of structural constraints at the chromosome level. If evolutionary forces keep these genes together, why can't these events happen more than once? Even more considering the loose way in which synteny blocks are defined in this study, with no collinearity. To sum up, loss of synteny or maybe even convergent evolution are processes not well-understood in these characters. This, together with the first point (lack of a strong signal grouping ctenophores with the outgroups) means that one cannot rule out that ctenophores could be nested within BCnS with sponges as a first-splitting metazoan phylum. On a related note, the fact that we don't have an "evolutionary model" behind these patterns makes it very hard to assess the evolutionary significance of the fusion events, which are present in very low numbers.

Third, methodology-wise, the approach to identify gene homology for the synteny analyses is reciprocal best blast hit (RBH). RBH fails to recognise homologs, due to biases in gene length and gene duplications, when compared to modern methods like OrthoFinder (Emms and Kelly 2015, *Genome Biology*). This is especially problematic with fast-evolving taxa like ctenophores, as these taxa add additional difficulties to homology identification (Weisman et al 2020, *PLOS Biology*; Natsidis et al 2021, *iScience*). These issues combined may produce a failure to detect shared homologs, especially exacerbated when the pairwise comparison involves the fast-evolving ctenophores, which might artifactually erase the signal of shared synteny blocks. Evidence of this potential issue is found in Supp Data Figures (Oxford plots including a ctenophore in Supp Figure 4.2 and following figures) which fail to recognise any synteny between ctenophores and any animal outgroup, and there is only a bit of signal when compared against sponges or cnidarians. These problems may also cause failure to recognise homologs between sponges and the outgroups. Finally, all the analyses in this manuscript are based on pairwise comparisons rather than using comparative phylogenetic methods. Shared genomic patterns observed in pairwise comparisons are not proof of the general processes explaining their emergence and are prone to statistical artifacts (Dunn et al 2018, *PNAS*). Comparative genomics should not be focused on the tips of the tree using pairwise comparisons but should implement inclusive phylogenetic comparative methods. The analyses come close to that in Supp Data (comparison using the metazoan ALG, Supp Figures 4.2i, 4.2j, 4.3e, 4.4i, 4.4j, 4.5e, 4.6, etc.), but then these plots show no syntenic signal. A holistic

phylogeny-based approach could be more elaborated, applied to different nodes of the tree, and expanded in the main text.

Overall, I think the synteny analyses are very interesting and original, showing great potential as phylogenetic markers, but not for the case of the ctenophores presented here. It is exciting to develop/discover new phylogenetic markers, but we must be cautious (Hillis 1999, PNAS). I think the same analyses and datasets could be used to elaborate further on the functional or structural significance of these blocks, their gene content (especially in blocks shared with the metazoan outgroups), their physical distribution in the chromosomes (is the conservation due to chromosome structure?) or their biological functions (is conservation due to similar expression/functional patterns?). At the moment, I think the genome biology and evolution of these syntenic blocks are more reliable and fundamental. This would advance our understanding of these syntenic blocks, which would help us to comprehend how they evolved and how they can be applied to phylogenetic questions.

Referee #2 (Remarks to the Author):

Schultz et al.

This manuscript provided some stunning and important data that directly bears on a major hotly-debated phylogenetic question. Are ctenophores or sponges the sister taxon to all other extant animals? The manuscript provides compelling evidence that the ctenophore-sister hypothesis is correct. This is a strong reason to publish this manuscript in a top-tier journal.

Although there is much to like with the manuscript there are several items that can be improved. Despite all these comments, the manuscript is clearly a valuable contribution, and the work is highly important. All the comments are largely issues with presentation that can be easily addressed.

Major items that really must be fixed prior to publication:

1) The connection and interdependence between this manuscript and Simakov et al. 2022 must be further explained in the manuscript. This is important because of the Placozoan issue.

2) Evidence for Placozoan-Cnidarian clade is lacking - The manuscript contends that placozoans and cnidarians are sister taxa, but in reality almost no data is provided to support this conclusion and yet this conclusion is a major part of the Results and Discussion despite being hardly raised earlier in the manuscript. Much of the discussion is about placozoan being part of the BCnS clade and not about placozoan and cnidarians being sister taxa. The statement is made, "The sub-chromosomal genome assembly of the placozoan *Trichoplax* shares several BCnS-ALG syntenic synapomorphies with cnidarians²⁷." In Simakov et al. 2022, the paper cited, the comment is made that only a single synapomorphy supports this conclusion. (On page 9 of that manuscript, "Thus, ALG_Eb ⊗ ALG_F is a synapomorphy (shared derived character) that unites placozoans and cnidarians as sister groups, consistent with some gene-based phylogenies (51)."). Please note gene trees pretty much supported

all possible relationships of extant basal animal lineages.

So, based on this one synapomorphy, the authors propose a new name “Placodarians.” This is not appropriate as they do not give sufficient evidence for the proposed clade. Also, I personally find this choice of name confusing and unfortunate. Why would you choose to emphasize the Placozoa (arguably only 1-3 morphotypes and about 25-30 lineages) over the rich diversity of Cnidaria? Further why call it Placodaria? A term that will lead many less familiar with animal diversity and phylogeny to think of placoderms! Given the evidence, just drop the naming for this group.

This issue also bleeds over into Figure 4. The tree is showing synapomorphies that support the conclusions of the manuscript, but no synapomorphy is given for uniting placozoans and cnidarians. What is the source of this topology? Where did it come from? Please note the topology used in Supplementary Fig 7 showing ambiguity is more appropriate.

The issue here is not if evidence for a cnidarian/placozoan clade is credible.... It just has not been presented. There is insufficient data for conclusions made or how those conclusions are emphasized.

Note in the Placodarian section, Citation #46 is given for Simakov et al 2022 – the number should be 27.

3) Where’s the gene info? This paper (and related ones) talks about the synteny groups, but no information is given on what genes or families of genes are being held together. This was a complete tease.... Like eating something that tastes great but is completely unsatisfying and one is hungry again 5 minutes later. If these gene clusters have been maintained over ½ a billion years, they must be pretty darn important. So, give the reader some idea of what they are. Given the opportunities over millions if not billions of generations to undergo fusion and fission, there must be a reasons genes stay on the same chromosome. Information which genes stay together seems critical.

Some suggestions that might help greatly improve the manuscript.

1) The manuscript writing need to improve given the journal being considered

a. The writing seems rushed and not well edited. For example, there is “Results & Discussion” and then another “Discussion” section. The flow of topics in Discussion can be better developed and do not flow direction from previous text – see below.

b. The manuscript is very redundant to read. For example, the time one gets to the discussion those points in the initial paragraph have been made repeatedly.

c. Much of the front half of the manuscript is written for too broad of an audience. At times it read more like a Scientific American manuscript meaning broad overarching ideas were given and the reader was asked to take the authors’ word for it. Add details to the mix to convince the reader. Most who would read this, know how phylogenetic arguments work. Too much time is spent in the text on this. Please note that Fig 2 does an excellent job of presenting the alternative hypotheses. So instead of repeating this in the text add some information about the synteny groups (which genes? how big are they?).

2) The discussion of morphological features at the end of the manuscript is simplistic. As others have

done the manuscript mistakenly talks about the origins neuromuscular systems relative to sponge vs ctenophore debate. That is not the issue as we know neuromuscular systems have evolved repeatedly within animals. The issue is evolution of neurons (and independent muscle cells).

3) Myriazoan – I do not think a name is need for this node, but if you really want to name it come up with a name that is at least partially informative and makes sense. OK so the last common ancestor of sponges, placozoan, cnidarians, and bilaterians is “Myriazoa” meaning “countless animals.” So when ctenophores are included is this “more than countless animals?” Is this like infinity plus one? Animals can be counted – it will just take a lot of work.

4) Models – The fact that modeling was employed to look at how synteny may evolve over time is a big plus for the manuscript. How the models were not well explained or detailed. The details are pages 26 and 27 in methods are not helpful. For example, when shuffling, what size regions were being shuffled? Was is always a 1:1 swap or were repeats and inversions allowed? On simulation was for 100 million times. While this sounds like a lot it is maybe a tenth of what happened in nature. These organisms have been separated for likely 600MY with rearrangements happening on 2 lineages. While it is unlikely that the results will change with more generations, it is incumbent on the authors to clear articulate the assumptions of the models and what potential limitations are. For example, there is no mention of the probably of shuffling along a chromosome relative to shuffling between chromosomes.

Comparing distributions between simulated and real data is a big positive. This should be represented in the main figures. It is more critical to the manuscript’s arguments than Fig 5 which is pretty-straight forward and obvious from the text and other figures.

5) Assumptions of ancestral ctenophore synteny and karyotype are stated too dogmatically. There may be two disparate ctenophores with the same pattern but it is big leap to assume that is correct for all ctenophores. That assumption is also dependent on our knowledge of ctenophore phylogeny which is questionable at best. It would be best to tone those statements down.

6) Chromosome numbers for the non-animals are presented. Where are these numbers derived from? Karyotype staining or assumptions based on genome assembly? It is fine either way be the source of the numbers used should be provide.

7) Page 16 – “A Parahoxozoa clade grouping bilaterians together with placozoans and cnidarians to the exclusion of sponges and ctenophores was originally proposed based on the shared presence of Hox/ParaHox-class genes⁵.” This should be reworded. The name Parahoxozoa may have been proposed in Ryan et al. but the clade/grouping of bilaterians, cnidarians and placozoans has been long hypothesized. The wording makes it sound like this idea came from the citation number 5.

Referee #3 (Remarks to the Author):

The manuscript by Schultz and colleagues provides chromosome-scale-level assemblies for a ctenophore, a sponge, and representatives of three unicellular protist lineages; they then use these assemblies to identify ancestral linkage groups (ALGs) and use them as characters to resolve relationships at the base of the metazoan phylogeny. The key finding is that there are seven such ALGs that are shared by all metazoans except ctenophores (they are also not conserved in protist outgroups). In contrast, there are no ALGs shared by all metazoans except sponges, providing support for the ctenophore-sister hypothesis. I found the manuscript to be well written and the work is generally well done and insightful. While the sponge/ctenophore debate may continue, this work definitely moves the needle and adds valuable new data to the debate.

I have a few comments that i would like the authors to address:

1. Maybe i missed it, but a lot of the ALG analysis rests on genome quality. Have the authors validated the accuracy of the new assemblies, especially for the seven ALGs? It would be good to examine at least a few of them and definitely show that they sit on separate linkage groups in ctenophores and protists and on single linkage groups in the rest of animals.
2. I would have liked to see a bit more analysis on the genes that make up these ALGs. Are they comprised of duplicated genes (like the Hox / ParaHox clusters)? Do they contain genes involved in the same function (like the metabolic gene clusters observed in fungi)? It is likely that these ALGs have been conserved through negative selection so adding some information about the functions of these 7 ALGs would be very interesting.
3. I did not find the naming of "Myriazoa" and "Placodaria" necessary or useful - given that these relationships have been controversial for some time now, I feel that naming additional clades ends up further contributing to the confusion (and makes reading of these studies even more difficult for outsiders) than helping. The discussion is a bit on the long side so i feel parts of these two sections could be deleted without much loss of information.
4. lines 387-403: thank you for stating this - it is amazing how often both science journalists and professional biologists conflate sponges and ctenophores with the likely animal ancestor!
5. Figure 1: Please also show a ctenophore-bilaterian synteny plot. This plot could be added or could replace the sponge-ctenophore plot.
6. Figure 2: can you also provide the numbers of events observed in your analysis that you attribute to the patterns of synteny shown in panels a through f? Also, not sure how informative or necessary the dot plots are (especially at this level of resolution)
7. Figure 3: either in the figure legend or (preferably) in the main text, it would be important to explain why the overlap in ALGs found between the analysis shown in panels a-e and the analysis

shown in panels f-j is only 2 ALGs.

Minor comments:

line 88: "such characters are easily lost" - has this been demonstrated? If not, please change "are easily lost" to "can be lost"

line 129: "(animal synapomorphies)" appears twice

lines 143-144: conservation of synteny without conservation of gene order has been called mesosynteny in fungi (see <https://genomebiology.biomedcentral.com/articles/10.1186/gb-2011-12-5-r45>). Would be good to cite this paper and add brief mention to this.

line 147: the BCnS acronym was defined in line 118 already - delete?

line 155: what does "residual colinearity" mean? please clarify.

line 172: "earliest branching" -> "sister"

line 280: "Trichoplax" should be in italics

Author Rebuttals to Initial Comments:

“Ancient gene linkages support ctenophores...”

Schultz et al. - Dec. 2022

Formatting Guide:

The comments from referees are in black and left-margin-aligned.

Responses from authors are in blue and are indented.

Referee #1 (Remarks to the Author):

I enjoyed reading the manuscript titled “Ancient gene linkages show that ctenophores are sister to other animals” by Schultz and collaborators. The manuscript is a data and analytical tour de force, producing novel chromosome-level genomes for early diverging animals and their closest relatives, which are analyzed together with other previously sequenced high-quality genomes. For the first time in deep metazoan phylogeny, the manuscript exploits the shared genome-wide chromosomal location of genes (synteny but without collinearity) as a phylogenetic marker. The analyses aim to resolve the contentious issue of the evolutionary position of sponges vs ctenophores as first-splitting animals.

Thanks!

The manuscript claims that syntenic regions support the ctenophores-first hypothesis, based on two required conditions: 1) syntenic blocks shared by bilaterians + cnidarians + sponges (BCnS) and 2) synteny shared by ctenophores and the single-celled outgroups. The analyses are original and sound (except for the homology identification approach, more on that later), and of the quality usually produced by the labs involved in the manuscript. The text is clear and well-written, although there are substantial mismatches between the figure panels invoked in the text and the actual figures, making an already complex manuscript harder to follow. I want to congratulate the authors for their efforts in developing a new approach to a question that has been (over)analyzed with classic phylogenomic methods — without being resolved — for almost 15 years.

Thanks. The gene homology identification issue is addressed below. We apologize for any confusion in the original manuscript arising from mismatches between figure panels and their citation in the text. We have extensively revised the figures and text in the resubmitted manuscript and corrected these issues.

Please note that conditions (1) and (2) summarized above omit an essential third condition of our analysis. Crucially, we also show that (3) the syntenies shared by BCnS are combinations (fusions) of the groups shared by ctenophores and single-celled outgroups, i.e., they are shared derived changes that unite bilaterians, cnidarians, and sponges to the exclusion of ctenophores. The alternative sponge-sister hypothesis requires improbable convergence and/or reversion (now more fully discussed and shown in **Fig. 4**). We have attempted to clarify this throughout the text, and in the responses below. ,

However, while the use of synteny as a phylogenetic marker is truly novel and very interesting, I am afraid its application to this particular phylogenetic question is not conclusive. I believe that the results (or their interpretation) do not support the main claim of the manuscript and its title.

We disagree, and have revised the manuscript to clarify potentially confusing points. We address your specific comments further below. Most notably, the original manuscript emphasized the parsimoniousness of the ctenophore-sister hypothesis with respect to syntenic characters. In the revised manuscript we have defined these syntenic characters more formally and provide a Bayesian phylogenetic analysis that reaches the same conclusion. We now also discuss the highly improbable changes in synteny that would be needed to explain the data in terms of the alternative sponge-sister hypothesis.

First, while the text shows compelling proof of the first condition (shared synteny between BCnS), it does not show a strong signal supporting the second one (shared synteny between ctenophores and the metazoan outgroups). Both conditions need to be fulfilled to support ctenophores-first, otherwise, syntenically apomorphic ctenophores could still be nested within a BCnS with sponges-first. The low number of events in Figure 3 is hard to assess from an evolutionary point of view, and might be biased by methodological issues outlined below. Supp Figures 4.2 to 4.8, which include the comparisons of the ctenophores against the outgroups, don't provide evidence of shared synteny between ctenophores and the outgroup. I suspect the manuscript is self-aware of that because the main text goes a great length to show various pairwise dot plots between members of the BCnS or between the metazoan outgroups and the BCnS, but it never shows any between the ctenophore and the outgroups which are presented in the Supp Data. Given that the sharing of syntenic blocks between ctenophores and the outgroups is one of the main conditions to support ctenophores-first, one would expect these analyses to be featured front and centre in the main text rather than a comparison between two ctenophores or two sponges.

Taking the last point first, we started the main text with a comparison of the two ctenophores and the two sponges to establish that within these two groups, extant ctenophore and sponge chromosomes are stable with respect to gene content but not gene order. This is a prerequisite for all that follows – for example, if extant ctenophore karyotypes differed wildly, then there would be no reason to expect that ancient syntenies are preserved. To document the extensive chromosome/chromosome-arm scale conservation within ctenophores and demosponges, we used pairwise dotplots, which are a useful guide to the trained eye. Following your suggestion, in the revision we have streamlined this preliminary material and simplified **Figure 1**, using a ribbon diagram that more clearly shows the broad conservation across metazoans, and the discontinuity between BCnS and ctenophores. The relevant dotplots showing conserved synteny among metazoans can now be found in **Extended Data Figure 2**.

Regarding the key issue of the demonstration of shared synteny between ctenophores, BCnS, and the unicellular outgroups to metazoa, this is now more clearly shown in the

revised **Figure 2a, b**. Notably, we found significant multi-species associations between ctenophore chromosomes and both metazoan and unicell chromosomes (false discovery rate $\alpha < 0.05$ under permutations of gene positions – see **Online Methods, Supplementary Information 8-10**, and **Extended Data Figure 6**). The requested ctenophore-outgroup and other dotplots are provided in **Extended Data Figures 7-8**, which also show significant pairwise associations (by Fisher’s exact test, or the gene conservation score in **Supplementary Information 11**). Not surprisingly, conserved syntenies between the unicellular outgroups *Capsaspora* and *Salpingoeca* and metazoa are less visually apparent from pairwise dotplots than the intra-metazoan dotplots due to the accumulation of small-scale translocations over a much longer period of time. Nevertheless they are statistically significant.

Second, the manuscript argues that the formation of these synten blocks is the result of “rare, irreversible chromosome fusion-and-mixing events”. However, it is not clear if these blocks can be lost. Or even if they may remerge by convergent evolution: if evolution keeps these genes generally together (synteny without colinearity) it means that there are selective pressures to keep genes together may be due to similar biological functions or because of structural constraints at the chromosome level. If evolutionary forces keep these genes together, why can’t these events happen more than once? Even more considering the loose way in which synteny blocks are defined in this study, with no colinearity. To sum up, loss of synteny or maybe even convergent evolution are processes not well-understood in these characters. This, together with the first point (lack of a strong signal grouping ctenophores with the outgroups) means that one cannot rule out that ctenophores could be nested within BCnS with sponges as a first-splitting metazoan phylum. On a related note, the fact that we don’t have an “evolutionary model” behind these patterns makes it very hard to assess the evolutionary significance of the fusion events, which are present in very low numbers.

Thanks for these comments. We attempt to clarify several key points.

1. The meaning of the comment “it is not clear if these blocks can be lost” is not clear to us. We traced the chromosomal distribution of genes across both in- and outgroup genomes. Since we focused on conserved genes, the genes themselves have not been lost; since we focused on conserved chromosomal linkages (synteny as defined originally Renwick 1973 in the sense of “same thread”), these specific linkages are also not lost. While there could, of course, be groups of genes that were linked in the ancestor but are no longer detectable, or that are conserved within one clade but cannot be traced in others, we do not consider them since they are phylogenetically uninformative.
2. The statement “if evolution keeps these genes generally together (synteny without colinearity) it mean that there are selective pressures to keep genes together,” and comments below, seems to suggest that the selective pressure is somehow related to the biological function of the genes involved. There is, however, a classical explanation for conserved synteny that relies instead on constraints on chromosome changes imposed by meiosis. The first evolutionary step in disrupting

chromosome-scale synteny is a partial arm translocation between non-homologous chromosomes. As first noted by HJ Muller (Muller, 1940), partial chromosome-arm translocations are deleterious because translocation heterozygotes typically have reduced fertility due to the production of unbalanced gametes. (Whole arm (Robertsonian) translocations are a special case.) It follows that there is purifying selection against changes in synteny *purely on the basis of selection against unbalanced gametes*, as further elaborated by Sewall Wright (Wright 1941) and modeled more recently by Lv et al (Lv et al. 2011). As noted by Wright (and further elaborated by R Lande (Lande 1985)), this purifying selection against meiotically disruptive rearrangements can be overcome by drift only in small or subdivided populations. Conversely, species with consistently large effective population sizes will tend to retain chromosome-scale linkages that erode over time by small-scale translocations.

3. Regarding “loss of synteny or maybe even convergent evolution are processes not well-understood in these characters,” we have clarified this further below and in the revised text. “Loss of synteny” is addressed above in point 1. Regarding possible convergence, to challenge our synteny-based phylogenetic argument and instead place sponges as sister to other animals would require seven convergent events that mimic the seven BCnS synapomorphies. In the revised manuscript we have discussed this theoretical possibility more clearly (see **Figure 4c,d**). Given our current understanding of chromosome evolution as dominated by fusions (with and without mixing) as documented in Simakov et al. 2022, each of the required convergences shown in **Figure 4c,d** are highly improbable (i.e., have never been observed in prior studies of chromosome evolution). We hope that by acknowledging the possibility of convergence (present in any phylogenetic analysis) we have more clearly stated our findings.

- Lande, R. (1985). The fixation of chromosomal rearrangements in a subdivided population with local extinction and colonization. *Heredity*, 54(3), 323-332.
- Muller, H. J. (1940). Bearing of the Drosophila work on systematics. *The new systematics*, 185-268.
- Wright, S. (1941). On the probability of fixation of reciprocal translocations. *The American Naturalist*, 75(761), 513-522.
- Lv, J., Havlak, P., & Putnam, N. H. (2011). Constraints on genes shape long-term conservation of macro-synteny in metazoan genomes. *BMC bioinformatics*, 12(9), 1-12

Third, methodology-wise, the approach to identify gene homology for the synteny analyses is reciprocal best blast hit (RBH). RBH fails to recognise homologs, due to biases in gene length and gene duplications, when compared to modern methods like OrthoFinder (Emms and Kelly 2015, *Genome Biology*). This is especially problematic with fast-evolving taxa like ctenophores, as these taxa add additional difficulties to homology identification (Weisman et al 2020, *PLOS Biology*; Natsidis et al 2021, *iScience*). These issues combined may produce a failure to detect shared homologs, especially exacerbated when the pairwise comparison involves the fast-evolving ctenophores, which might artifactually erase the signal of shared synteny blocks. Evidence of this potential issue is found in Supp Data Figures (Oxford plots including a ctenophore in Supp Figure 4.2 and following figures) which fail to recognise any synteny between ctenophores and any animal outgroup, and there is only a bit of signal when compared against sponges or cnidarians. These problems may also cause failure to recognise homologs between sponges and the outgroups.

Our preference in the original submission was to use “mutual best hits” as the simplest and most direct method for identifying putative orthologs. As noted by the reviewer comment above this a conservative approach that may not detect some orthologs (as, e.g., in the cited papers by Weisman et al. 2020 or Natsidis et al. 2021) that a more sensitive/complex orthology-finding algorithm might find. Nevertheless, the numbers of genes we identified with a simple approach were more than enough to reach statistically well-supported conclusions, and we find significant conservation of linkage supporting ctenophores are retaining ancestral linkages that are fused and mixed in bilaterians, cnidarians, and sponges.

To test the robustness of our approach, as suggested we repeated our calculations using OrthoFinder (Emms and Kelly 2015). As anticipated this approach recovers the same conclusions as found using mutual-best-hits. This alternative OrthoFinder analysis is described in **Supplementary Information 10** and **Extended Data Figure 10**.

Finally, all the analyses in this manuscript are based on pairwise comparisons rather than using comparative phylogenetic methods. Shared genomic patterns observed in pairwise comparisons are not proof of the general processes explaining their emergence and are prone to statistical artifacts (Dunn et al 2018, *PNAS*). Comparative genomics should not be focused on the tips of the tree using pairwise comparisons but should implement inclusive phylogenetic comparative methods. The analyses come close to that in Supp Data (comparison using the metazoan ALG, Supp Figures 4.2i, 4.2j, 4.3e, 4.4i, 4.4j, 4.5e, 4.6, etc.), but then these plots show no syntenic signal. A holistic phylogeny-based approach could be more elaborated, applied to different nodes of the tree, and expanded in the main text.

Thanks for raising this point. Our arguments are in fact fully comparative and are **not** simply based on pairwise comparisons. The pairwise dot plots in the original manuscript were meant to frame the question and may have given a mistaken impression that they were the basis of our analysis. To emphasize the inherently comparative nature of our

study we revised the main figures and text. Specifically, in the revision we have (1) more clearly described the character states that our analysis is based on, (2) provided a formal matrix of these characters, and (3) applied Bayesian phylogenetic analyses of this matrix (**Supplementary Information 14**) in addition to the earlier parsimony-based analyses.

In our manuscript we identify synteny-based characters (chromosomal linkages of sets of orthologous genes) based on multi-species comparisons, and analyze them using cladistic methods. As in any phylogenetic analysis, we seek characters that unite sub-clades to the exclusion of others, and use outgroups to polarize changes. The synteny character states that carry phylogenetic information relevant to ctenophore- vs. sponge-sister debate are shown in **Figures 2** and **3**. Support for the ctenophore-sister hypothesis is evident in our data and is most easily appreciated in a simple parsimony framework once the irreversibility of fusion-with-mixing is recognized. Extensive convergence would be needed to accommodate a sponge-sister phylogeny.

Overall, I think the synteny analyses are very interesting and original, showing great potential as phylogenetic markers, but not for the case of the ctenophores presented here. It is exciting to develop/discover new phylogenetic markers, but we must be cautious (Hillis 1999, PNAS). I think the same analyses and datasets could be used to elaborate further on the functional or structural significance of these blocks, their gene content (especially in blocks shared with the metazoan outgroups), their physical distribution in the chromosomes (is the conservation due to chromosome structure?) or their biological functions (is conservation due to similar expression/functional patterns?). At the moment, I think the genome biology and evolution of these syntenic blocks are more reliable and fundamental. This would advance our understanding of these syntenic blocks, which would help us to comprehend how they evolved and how they can be applied to phylogenetic questions.

We thank the reviewer for their thoughtful comments, but stand by our conclusion that synteny-based characters strongly support ctenophores as sister to all other animals. There are several points raised here, some of which we have addressed throughout this response.

As suggested, we have tried to be “cautious” in the revised manuscript by more clearly stating our assumptions and by detailing the improbable set of character changes that would be required to make our data consistent with the alternative sponge-sister hypothesis (**Figure 4**). We cite Hillis PNAS 1999 (in both the original and revised manuscripts) for the key idea that rare and irreversible changes are attractive phylogenetic characters. We have presented fusions of ancestral synteny groups as rare changes that, when accompanied by mixing, are effectively irreversible, in the same sense that the spontaneous reversal of chemical mixing is irreversible. We find four such fusions-with-mixing that support ctenophore-sister (and three without mixing).

As noted above, while there may be “functional or structural significance” to conserved synteny, we did not find any functional correlations among anciently linked genes. We agree that this is an interesting question and the data is made available for others to explore. In the absence of any obvious functional significance, we favor the “null hypothesis” in which changes in synteny are rare because of the generic selection against translocation heterozygotes, as articulated by HJ Muller, S. Wright, and others. Since this selection can be weak against small-scale translocations (as argued by Lv et al. 2011), it appears that conservation of synteny typically decays by the steady accumulation of such small translocations over time (quantified in Simakov et al. 2022 among BCnS). As we now point out more clearly in the revised manuscript, this slow rate of small-scale translocations is also consistent with the residual conservation of synteny between metazoans and their unicellular relatives.

We propose that this residual conserved synteny represents a useful new class of phylogenetic character. Rare changes in these characters are dominated by fusion of ancestral synteny groups, as previously documented among BCnS (see Simakov et al 2022 for discussion) and extended here to include ctenophores and non-metazoan outgroups. We hope that the revised manuscript more clearly describes these characters and how they can be used to differentiate between the ctenophore-sister and sponge-sister hypotheses that have been so challenging to resolve with traditional sequence-based methods.

Referee #2, evolutionary genetics, phylogenomics (Remarks to the Author):

This manuscript provided some stunning and important data that directly bears on a major hotly-debated phylogenetic question. Are ctenophores or sponges the sister taxon to all other extant animals? The manuscript provides compelling evidence that the ctenophore-sister hypotheses is correct. This is a strong reason to publish this manuscript in a top-tier journal.

Although there is much to like with the manuscript there are several items that can be improved. Despite all these comments, the manuscript is clearly a valuable contribution, and the work is highly important. All the comments are largely issues with presentation that can be easily addressed.

Thanks!

Major items that really must be fixed prior to publication:

1) The connection and interdependence between this manuscript and Simakov et al. 2022 must be further explained in the manuscript. This is important because of the Placozoan issue.

Thanks. We have made efforts to clarify the connection between this manuscript and Simakov et al 2022 (which focuses on bilaterians, cnidarians, and sponge), and have cited this earlier study where appropriate. Specifically regarding the “Placozoan issue” we have revised the text to emphasize that the current analysis only groups placozoans with bilaterians and cnidarians, as discussed further below. The data supporting this grouping can be found in **Figure 3**, which shows that the fusions leading to A1a, C1, and G are shared by *Trichoplax*, bilaterians, cnidarians, and sponges to the exclusion of ctenophores. We clarify in the legend of Figure 3 that our display of placozoans as the sister clade of cnidarians is based on the results of Simakov et al. 2022, and not a novel finding from this study.

2) Evidence for Placozoan-Cnidarian clade is lacking - The manuscript contends that placozoans and cnidarians are sister taxa, but in reality almost no data is provided to support this conclusion and yet this conclusion is a major part of the Results and Discussion despite being hardly raised earlier in the manuscript. Much of the discussion is about placozoan being part of the BCnS clade and not about placozoan and cnidarians being sister taxa. The statement is made, “The sub-chromosomal genome assembly of the placozoan *Trichoplax* shares several BCnS-ALG syntenic synapomorphies with cnidarians²⁷.” In Simakov et al. 2022, the paper cited, the comment is made that only a single synapomorphy supports this conclusion. (On page 9 of that manuscript, “Thus, ALG_Eb[⊗]ALG_F is a synapomorphy (shared derived character) that unites placozoans and cnidarians as sister groups, consistent with some gene-based phylogenies (51).”). Please note gene trees pretty much supported all possible relationships of extant basal animal lineages.

So, based on this one synapomorphy, the authors propose a new name “Placodarians.” This is not appropriate as they do not give sufficient evidence for the proposed clade. Also, I personally find this choice of name confusing and unfortunate. Why would you choose to emphasize the Placozoa (arguably only 1-3 morphotypes and about 25-30 lineages) over the rich diversity of Cnidaria? Further why call it Placodaria? A term that will lead many less familiar with animal diversity and phylogeny to think of placoderms! Given the evidence, just drop the naming for this group.

This issue also bleeds over into Figure 4. The tree is showing synapomorphies that support the conclusions of the manuscript, but no synapomorphy is given for uniting placozoans and cnidarians. What is the source of this topology? Where did it come from? Please note the topology used in Supplementary Fig 7 showing ambiguity is more appropriate.

The issue here is not if evidence for a cnidarian/placozoan clade is credible.... It just has not been presented. There is insufficient data for conclusions made or how those conclusions are emphasized.

Note in the Placodarian section, Citation #46 is given for Simakov et al 2022 – the number should be 27.

Thanks for raising these points. We have revised the presentation to streamline the discussion of the evidence for the relationship of placozoa to other animals and emphasize results from the present analysis..

As noted above, in the present manuscript we find syntenic characters that group placozoans (*Trichoplax*) with bilaterians, cnidarians, and sponges to the exclusion of ctenophores. These are shown in **Fig. 3** and include fusions leading to A1a, C1, G, and L. (Note that the non-chromosomal nature of the *Trichoplax* genome assemblies mean that we cannot reliably characterize the states of Ea and F, which is why those cells are blank in **Fig. 3**; N is present but the fragmented nature of the assembly means we cannot conclude from this data that *Trichoplax* shares this fusion.) These observations rule out placozoans as candidates for the sister group to all other animals, a finding that is consistent with our main ctenophore-sister result.

In **Figures 3b** and **4g**, where placozoans are shown in a phylogeny, we placed placozoans as sister to cnidarians and cited Simakov et al. 2022 for the ALG_Eb[⊗]ALG_F syntenic synapomorphy that unites these two lineages. Since we now cite this as a prior result we hope that this clarifies the rationale for this topology. We have also modified **Extended Data Figure 3** and **Extended Data Figure 9** accordingly. The reason that this particular fusion-with-mixing does not appear in the present manuscript is that here we only consider syntenic characters that can be traced back to the outgroups, and the present scaffolded assemblies of *Trichoplax* do not allow BCnS ALG's Eb and F to be

analyzed in this way. Regarding gene trees supporting all possible relationships, as true as that is, it still seems appropriate to cite prior work with the same conclusion.

Taking the discussion above into consideration, we have revised the text by simply stating our observations and removing the proposed “Placodarian” name. The syntenic relationships between placozoans and other BCnS members will be better addressed with a chromosome-scale placozoan genome.

3) Where’s the gene info? This paper (and related ones) talks about the synteny groups, but no information is given on what genes or families of genes are being held together. This was a complete tease.... Like eating something that tastes great but is completely unsatisfying and one is hungry again 5 minutes later. If these gene clusters have been maintained over ½ a billion years, they must be pretty darn important. So, give the reader some idea of what they are. Given the opportunities over millions if not billions of generations to undergo fusion and fission, there must be a reasons genes stay on the same chromosome. Information which genes stay together seems critical.

Thanks, and apologies if it seemed like a tease! We have rewritten parts of the text to respond to this comment (and similar questions from other reviewers). “If the gene clusters have been maintained over ½ a billion years, they must be pretty darn important” suggests that the syntenic clusters are held together by some selective pressure depending on the biological functions of the genes involved. While this is true for some famous clusters (e.g., Hox), we looked for but did not find any gene-functional explanation for the maintenance of chromosome-scale linkage, using methods like GO-term enrichment as now noted more clearly in the text (and discussed further in **Supplementary Information 12**). While previously we provided a list of gene IDs belonging to each synteny group (**Supplementary Data 2**, Tab 8) we have now added a table with their annotations (**Supplementary Data 3**) to allow others to more easily search for possible functional relevance of chromosome-scale linkage using other methods.

Notably, there is a classical and less dramatic explanation for conserved synteny that relies on constraints on chromosome changes imposed by meiosis rather than on details of gene function. The first evolutionary step in disrupting chromosome-scale synteny is a partial arm translocation between non-homologous chromosomes. As first noted by HJ Muller (Muller, 1940), partial chromosome-arm translocations are deleterious because translocation heterozygotes typically have reduced fertility due to the production of unbalanced gametes. (Translocations involving whole arms (i.e., Robertsonian translocations) are a special case.) It follows that there is purifying selection against changes in synteny *purely on the basis of selection against unbalanced gametes*, as further elaborated by Sewall Wright (Wright 1941). As noted by Wright and further analyzed by R Lande (Lande 1985), the underdominance of meiotically disruptive rearrangements can be overcome by drift only in small or subdivided populations.

We therefore expect that lineages that have maintained consistently large effective population sizes will tend to retain chromosome-scale linkages. These will erode gradually over time by the accumulation of many small-scale translocations, each of which transfers a few genes between chromosomes (Lv et al. 2011). We now note in the text that extrapolating the rate of these small scale changes (as estimated in Simakov et al. 2022) are consistent with the retention of limited conserved synteny between some metazoans and non-metazoan outgroups.

This is a simple picture that explains widespread conservation of linkage in (largely marine) invertebrates with large population sizes. It connects to the meiotic function of chromosomes but does not depend on the individual genes involved. Presumably the relatively few changes that have occurred in a large number of metazoan lineages (as discussed here and in Simakov et al. 2022) arise as discussed by Wright and Lande. As noted in Simakov et al. 2022 (Science Advances), however, exceptions to this rule include flies (which have conserved synteny among themselves, as first emphasized by Muller (1940), but which have only limited conserved synteny with other invertebrates), and a sea squirt (*Ciona intestinalis*) that is well-known to be a “long branch” with respect to genome organization (see, e.g., Holland and Gibson Brown, “The *Ciona intestinalis* genome: when the constraints are off” <https://pubmed.ncbi.nlm.nih.gov/12766941/>) that is also rearranged relative to other deuterostomes like amphioxus and sea urchins (Simakov et al. 2022).

The meiotic perspective flips the question around: what is it about some lineages that allow their genomes to become so rearranged? This is a fascinating topic for further discussion, but the main point for our analysis is that two of the outgroups we consider (a choanoflagellate and a filasterian amoeba), along with ctenophores, demosponges, and representative bilaterians and cnidarians preserve ancient synteny that we can use as phylogenetic characters. Notably, the lack of gene-functional coherence of the ancient synteny groups makes them more suitable for use as phylogenomic characters, since they are less likely to arise by convergent fusion driven by gene function.

- Holland, L. Z., & Gibson-Brown, J. J. (2003). The *Ciona intestinalis* genome: when the constraints are off. *Bioessays*, 25(6), 529-532.
- Muller, H. J. (1940). Bearing of the *Drosophila* work on systematics. *The new systematics*, 185-268.
- Wright, S. (1941). On the probability of fixation of reciprocal translocations. *The American Naturalist*, 75(761), 513-522.
- Lv, J., Havlak, P., & Putnam, N. H. (2011). Constraints on genes shape long-term conservation of macro-synteny in metazoan genomes. *BMC bioinformatics*, 12(9), 1-12

Some suggestions that might help greatly improve the manuscript.

1) The manuscript writing need to improve given the journal being considered

1. The writing seems rushed and not well edited. For example, there is “Results & Discussion” and then another “Discussion” section. The flow of topics in Discussion can be better developed and do not flow direction from previous text – see below.

Thanks. We have revised most of the text, corrected typos and attempted to ensure that the text flows better.

2. The manuscript is very redundant to read. For example, the time one gets to the discussion those points in the initial paragraph have been made repeatedly.

Thanks. We have tried to streamline the text throughout to minimize redundancy.

3. Much of the front half of the manuscript is written for too broad of an audience. At times it read more like a Scientific American manuscript meaning broad overarching ideas were given and the reader was asked to take the authors’ word for it. Add details to the mix to convince the reader. Most who would read this, know how phylogenetic arguments work. Too much time is spent in the text on this. Please note that Fig 2 does an excellent job of presenting the alternative hypotheses. So instead of repeating this in the text add some information about the synteny groups (which genes? how big are they?).

Thanks for this suggestion. We have streamlined the introduction accordingly. We have also provided the requested information about the number and names of genes in each synteny group as **Supplementary Data 2 and 3**.

2) The discussion of morphological features at the end of the manuscript is simplistic. As others have done the manuscript mistakenly talks about the origins neuromuscular systems relative to sponge vs ctenophore debate. That is not the issue as we know neuromuscular systems have evolved repeatedly within animals. The issue is evolution of neurons (and independent muscle cells).

We have revised this brief paragraph. We think it is important to at least allude to the implications of our ctenophore-sister conclusion, lest readers think it is a sterile question of purely academic interest.

3) Myriazoan – I do not think a name is needed for this node, but if you really want to name it come up with a name that is at least partially informative and makes sense. OK so the last common ancestor of sponges, placozoan, cnidarians, and bilaterians is “Myriazoa” meaning “countless animals.” So when ctenophores are included is this “more than countless animals?” Is this like infinity plus one? Animals can be counted – it will just take a lot of work.

In discussion with systematist and phylogeneticist colleagues (notably, Casey Dunn) we were encouraged to propose a name for this clade. We tried many other names and this one seemed to us (and colleagues we surveyed) to be the best one, reflecting (as noted) the diversity of this animal group. While we provided one translation of the Greek “myrios” it also just means “many” and perhaps this is a better translation than “countless.” There is precedent for using myrios in clade names (e.g., “myriapods”). As noted, we think “Benthozoa” is a bad choice because it incorrectly implies that the ancestor of sponges, bilaterians, cnidarians, and placozoans were benthic. Another names we rejected were Ctenadelphozoa (sister to ctenophores) which has the disadvantage of defining the group relative to what it isn’t. Myriazoa seemed the best choice.

4) Models – The fact that modeling was employed to look at how synteny may evolve over time is a big plus for the manuscript. How the models were not well explained or detailed. The details are pages 26 and 27 in methods are not helpful. For example, when shuffling, what size regions were being shuffled? Was is always a 1:1 swap or were repeats and inversions allowed? On simulation was for 100 million times. While this sounds like a lot it is maybe a tenth of what happened in nature. These organisms have been separated for likely 600MY with rearrangements happening on 2 lineages. While it is unlikely that the results will change with more generations, it is incumbent on the authors to clear articulate the assumptions of the models and what potential limitations are. For example, there is no mention of the probably of shuffling along a chromosome relative to shuffling between chromosomes.

Thanks. We have attempted to clarify our models as requested in both the Online Methods and in the Supplementary Information. In the original submission there were two types of calculations. In this revision we now explicitly use fusions of conserved synteny groups as characters for Bayesian phylogenetic analysis.

1. First, regarding the number of permutations sampled in genome shuffling, our goal was to develop a false discovery rate alpha-value (or bounds on an alpha-value) under a specific model of random linkage. Since our various statistics depend on the number of chromosomes and genes/chromosome in each taxon, a permutation test was the best choice. We simply shuffled the chromosome labels for the different proteins in the genome to simulate a genome with identical gene and chromosome content, but unrelated gene composition on each chromosome. To be clear, these are not simulations of individual chromosomal translocations or inversions – each simulation was a complete randomization of gene labels. We have tried to clarify this in the revised text. After sampling one hundred million permutations, we did not find a single simulated genome that matched the degree of evidence for ctenophore-sister as the real genomes. The distribution of results suggests that sampling additional permutations would only drive our bound on the false discovery rate down further. We mention these findings in the results section and have tried to

match the language of the results and Online Methods to make the connection more clear for our readers.

2. Second, we developed a simple simulation of mixing after chromosome fusion. In this test we seek to determine whether the genes in the *_x* and *_y* gene sets are mixed, or unmixed. The measure for mixing that we have devised for this study is scaled between 0 and 1, wherein 0 means that the two sets of genes in *_x* and *_y* are not interlaced, and 1, wherein the genes from *_x* and *_y* alternate perfectly. Given size discrepancies between the number of genes in *_x* and *_y* the “most mixed” state may be less than one. For this reason we build a distribution of mixing values for randomly sorted groups of the same sizes as *_x* and *_y*. The real mixing value is measured against the distribution to acquire an alpha value that the true genome is unmixed. As above, we have edited the Online Methods and to be more clear for a quick readthrough.
3. We have also included evolutionary models with asymmetric rates as implemented in RevBayes and MrBayes, based on character matrices of presence and absence of particular fusion-with-mixing events across species. We model the (very low) probability of (1) unmixing, and that of (2) convergent fusion (see **Supplementary Information 14**). In all analyses, ctenophores branched as earliest diverging animals.

Comparing distributions between simulated and real data is a big positive. This should be represented in the main figures. It is more critical to the manuscript's arguments than Fig 5 which is pretty-straight forward and obvious from the text and other figures.

Thanks. As suggested we have eliminated **Figure 5** of the original submission, although a summary is now provided as the final panel of **Figure 4**. We compare distributions between simulated and real data in **Figure 4f**. The revised manuscript also includes Bayesian analysis and shows posterior probabilities supporting the ctenophore positioning. The simulation shows that the patterns of synteny we observe are exceedingly unlikely to arise by chance; the Bayesian analysis shows that they overwhelmingly support a ctenophore-sister scenario.

We further support our scenario by now directly showing the most parsimonious explanation for the observed patterns of synteny (**Fig 4b**), and show the unlikely convergent changes in synteny that would be required if the alternative sponge-sister hypothesis were true (**Figure 4c,d**).

5) Assumptions of ancestral ctenophore synteny and karyotype are stated too dogmatically. There may be two disparate ctenophores with the same pattern but it is big leap to assume that is correct for all ctenophores. That assumption is also dependent on our knowledge of ctenophore phylogeny which is questionable at best. It would be best to tone those statements down.

We note that (1) the n=13 chromosomes of two deeply diverged ctenophores are in clear 1:1 correspondence (dot plot), and (2) most ctenophores have n=13 karyotypes (our unpublished observations). Our finding strongly implies that the last common ancestor of these two ctenophores had n=13 chromosomes, each carrying the same genes as our two exemplars, and that other ctenophores with same karyotype are likely to have homologous chromosomes. To make this clearer we now note that we are talking about the ancestral karyotype of the clade containing *Hormiphora* and *Bolinopsis*.

6) Chromosome numbers for the non-animals are presented. Where are these numbers derived from? Karyotype staining or assumptions based on genome assembly? It is fine either way be the source of the numbers used should be provided.

Thanks. We have noted more clearly in the main text that these numbers are based on our genome assemblies and Hi-C data. There are very few karyotype studies outside of plants, fungi, and animals, so this information is generally not readily available.

7) Page 16 – “A Parahoxozoa clade grouping bilaterians together with placozoans and cnidarians to the exclusion of sponges and ctenophores was originally proposed based on the shared presence of Hox/ParaHox-class genes⁵.” This should be reworded. The name Parahoxozoa may have been proposed in Ryan et al. but the clade/grouping of bilaterians, cnidarians and placozoans has been long hypothesized. The wording makes it sound like this idea came from the citation number 5.

Thanks. We have rewritten this sentence section to cite the relevant original proposals for a clade of bilaterians, cnidarians, and placozoans. We now cite Wainright et al 1993 for the earliest proposal of this clade, Collins 1998 for an extensive investigation into this topology, and Ryan et al 2010 for the first use of the term Parahoxozoa.

- Wainright, P. O., Hinkle, G., Sogin, M. L., & Stickel, S. K. (1993). Monophyletic origins of the metazoa: an evolutionary link with fungi. *Science*, 260(5106), 340-342.
- Collins, A. G. (1998). Evaluating multiple alternative hypotheses for the origin of Bilateria: an analysis of 18S rRNA molecular evidence. *Proceedings of the National Academy of Sciences*, 95(26), 15458-15463.
- Ryan, J. F., Pang, K., Mullikin, J. C., Martindale, M. Q., & Baxevanis, A. D. (2010). The homeodomain complement of the ctenophore *Mnemiopsis leidyi* suggests that Ctenophora and Porifera diverged prior to the ParaHoxozoa. *Evodevo*, 1(1), 1-18.

Referee #3 (Remarks to the Author):

The manuscript by Schultz and colleagues provides chromosome-scale-level assemblies for a ctenophore, a sponge, and representatives of three unicellular protist lineages; they then use these assemblies to identify ancestral linkage groups (ALGs) and use them as characters to resolve relationships at the base of the metazoan phylogeny. The key finding is that there are seven such ALGs that are shared by all metazoans except ctenophores (they are also not conserved in protist outgroups). In contrast, there are no ALGs shared by all metazoans except sponges, providing support for the ctenophore-sister hypothesis. I found the manuscript to be well written and the work is generally well done and insightful. While the sponge/ctenophore debate may continue, this work definitely moves the needle and adds valuable new data to the debate.

Thanks.

I have a few comments that i would like the authors to address:

1. Maybe i missed it, but a lot of the ALG analysis rests on genome quality. Have the authors validated the accuracy of the new assemblies, especially for the seven ALGs? It would be good to examine at least a few of them and definitely show that they sit on separate linkage groups in ctenophores and protists and on single linkage groups in the rest of animals.

All of our results are based on genome assemblies, and we tried to make this clearer in the revised manuscript. The use of high-throughput chromatin confirmation capture (HiC) for chromosome-scale genome assembly is now well-established and has been validated for many other genome sequences (e.g., our amphioxus assembly Simakov Nature Communications 2020 was validated by a genetic linkage map) and by many others. The method is now quite well developed and has been used by many other groups in numerous other works, and all of our new genome assemblies were manually curated following best practices. The consistent conserved synteny observed between (1) two independently assembled demosponge genomes, and (2) the two independently assembled ctenophore genomes, as well as (3) the consistent metazoan synteny observed among metazoan groups, and (4) with two independently assembled unicellular outgroups makes it implausible that a series of coordinated assembly errors could have led to our findings. These findings do not support a scenario explained by many independent misassemblies across these independent *de novo* genome assembly projects that all converge on placing the same genes, wrongly, together on chromosomes.

2. I would have liked to see a bit more analysis on the genes that make up these ALGs. Are they comprised of duplicated genes (like the Hox / ParaHox clusters)? Do they contain genes involved in the same function (like the metabolic gene clusters observed in fungi)? It is likely that these ALGs have been conserved through negative selection so adding some information about the functions of these 7 ALGs would be very interesting.

Thanks. As now noted more clearly in the text we looked for but did not find any gene-functional explanation for the maintenance of chromosome-scale linkage, using methods like GO-term enrichment (**Supplementary Information 12**). The genes involved are paralogs. While previously we provided a list of gene IDs belonging to each synteny group (**Supplementary Data 2**, Tab 8) we have now added a table with their annotations (**Supplementary Data 3**) to allow others to more easily search for possible functional relevance of chromosome-scale linkage using other methods.

We have rewritten parts of the text to respond to this comment (and similar questions from other reviewers). The comment “It is likely that these ALGs have been conserved through negative selection” assumes that the syntenic groups are held together at distant regions of the same chromosome by some selective pressure depending on the biological functions of the genes involved.

There is, however, a classical explanation for conserved synteny that relies instead on constraints on chromosome changes imposed by meiosis. The first evolutionary step in disrupting chromosome-scale synteny is a partial arm translocation between non-homologous chromosomes. As first noted by HJ Muller (Muller, 1940), partial chromosome-arm translocations are deleterious because translocation heterozygotes typically have reduced fertility due to the production of unbalanced gametes. (Robertsonian translocations involving whole arms are a special case.) It follows that there is purifying selection against changes in synteny *purely on the basis of selection against unbalanced gametes*, as further elaborated by Sewall Wright (Wright 1941). As noted by Wright and further analyzed by R Lande (Lande 1985), the underdominance of meiotically disruptive rearrangements can only be overcome by drift in small or subdivided populations.

Therefore we expect that lineages that have maintained consistently large effective population sizes will tend to retain chromosome-scale linkages. These will erode gradually over time by the accumulation of many small-scale translocations, each of which transfers a few genes between chromosomes (Lv et al. 2011). We now note in the text that extrapolating the rate of these small scale changes (as estimated in Simakov et al. 2022) are consistent with the retention of limited conserved syntenies between some metazoans and non-metazoan outgroups.

This is a simple picture that explains widespread conservation of linkage in (largely marine) invertebrates with large populations sizes. It connects to the meiotic function of

chromosomes but does not depend on the individual genes involved. Presumably the relatively few changes that have occurred in a large number of metazoan lineages (as discussed here and in Simakov et al. 2022) arise as discussed by Wright and Lande (1941). As noted in Simakov et al. 2022, however, exceptions to this rule include flies (which have conserved synteny among themselves, as first emphasized by Muller (1940), but which have only limited conserved syntenies with other invertebrates), and a sea squirt (*Ciona intestinalis*) that is well-known to be a “long branch” with respect to genome organization (see, e.g., Holland and Gibson Brown, “The *Ciona intestinalis* genome: when the constraints are off” <https://pubmed.ncbi.nlm.nih.gov/12766941/>) that is also rearranged relative to other deuterostomes like amphioxus and sea urchins (Simakov et al. 2022).

- Muller, H. J. (1940). Bearing of the Drosophila work on systematics. The new systematics, 185-268.
- Wright, S. (1941). On the probability of fixation of reciprocal translocations. The American Naturalist, 75(761), 513-522.
- Lande, R. (1985). The fixation of chromosomal rearrangements in a subdivided population with local extinction and colonization. Heredity, 54(3), 323-332.
- Lv, J., Havlak, P., & Putnam, N. H. (2011). Constraints on genes shape long-term conservation of macro-synteny in metazoan genomes. BMC bioinformatics, 12(9), 1-12
- Simakov, O., Bredeson, J., Berkoff, K., Marletaz, F., Mitros, T., Schultz, D. T., ... & Rokhsar, D. S. (2022). Deeply conserved synteny and the evolution of metazoan chromosomes. Science advances, 8(5), eabi5884.

3. I did not find the naming of "Myriazoa" and "Placodaria" necessary or useful - given that these relationships have been controversial for some time now, I feel that naming additional clades ends up further contributing to the confusion (and makes reading of these studies even more difficult for outsiders) than helping. The discussion is a bit on the long side so I feel parts of these two sections could be deleted without much loss of information.

As suggested we have simplified the discussion and removed the proposed name Placodaria. In discussion with systematist and phylogeneticist colleagues (notably, Casey Dunn) we were encouraged to propose a name for the clade comprising bilaterians, cnidarians, sponges, and placozoans. We considered many other names and this one seemed to us (and colleagues we surveyed) to be the best one, reflecting (as noted) the diversity of this animal group. There is precedent for using myrios in clade names (e.g., “myriapods”). As noted, we think “Benthozoa” is a bad choice because it incorrectly implies that the ancestor of sponges, bilaterians, cnidarians, and placozoans were benthic. Another names we rejected were Ctenadelphozoa (sister to ctenophores) which has the disadvantage of defining the group relative to what it isn’t. Myriazoa seemed the best choice.

4. lines 387-403: thank you for stating this - it is amazing how often both science journalists and professional biologists conflate sponges and ctenophores with the likely animal ancestor!

Thanks, we agree that this is important to emphasize.

5. Figure 1: Please also show a ctenophore-bilaterian synteny plot. This plot could be added or could replace the sponge-ctenophore plot.

We have completely reworked Figure 1, which now includes a ribbon diagram that includes ctenophore macrosynteny with all other lineages, including bilaterians. This diagram has the advantage that it shows conservation across multiple species, compared with a pairwise dotplot. The requested ctenophore-bilaterian dotplot can be found in **Extended Data Figure 2** and is discussed in **Supplementary Information 7**.

6. Figure 2: can you also provide the numbers of events observed in your analysis that you attribute to the patterns of synteny shown in panels a through f? Also, not sure how informative or necessary the dot plots are (especially at this level of resolution)

We agree that the dot plots are not particularly useful in such complex cases and we have revised **Figure 1** and **2** to provide more informative figures. The dotplots are relegated to **Extended Data Figure 2**. Following your suggestion, we have added numbers of occurrences of different types of chromosome tectonic events as appropriate to **Figure 2**.

7. Figure 3: either in the figure legend or (preferably) in the main text, it would be important to explain why the overlap in ALGs found between the analysis shown in panels a-e and the analysis shown in panels f-j is only 2 ALGs.

In the revised **Fig. 2**, the panels in question are now **2h** and **2i**, and we have noted this in the corresponding legend. In these panels we focus on ALGs that are phylogenetically informative for testing the sponge-sister vs. ctenophore-sister hypotheses. Each outgroup establishes a distinct set of ancestral metazoan synteny. Between the two outgroups we find 7 distinct synteny groups that serve to polarize changes in metazoans. Notably for the two that are found to be statistically significant for both outgroups (A1a and G), the character states determined by two outgroups agree. This is also shown in **Fig. 3**, where blank cells indicate synteny not found. From a cladistic perspective, however, we only need to show that a metazoan synteny group is conserved in a single outgroup to infer that it is ancestral.

Minor comments:

line 88: "such characters are easily lost" - has this been demonstrated? If not, please change "are easily lost" to "can be lost"

Thanks, we have made this change as suggested.

line 129: "(animal synapomorphies)" appears twice

Thanks for catching this.

lines 143-144: conservation of synteny without conservation of gene order has been called mesosynteny in fungi (see <https://genomebiology.biomedcentral.com/articles/10.1186/gb-2011-12-5-r45>). Would be good to cite this paper and add brief mention to this.

We added a citation to this paper but have not mentioned the term “mesosynteny”, since we think it could be confusing. The term ‘synteny’ has a standard and established definition due to Renwick 1973, where it was introduced to refer to physical linkage on the same chromosome without regard to gene order (in contrast to genetic linkage a la Sturtevant, which may not exist, for example, between loci at opposite ends of a large metacentric chromosome).

line 147: the BCnS acronym was defined in line 118 already - delete?

Thanks, we removed this redundancy.

line 155: what does "residual colinearity" mean? please clarify.

In the revision, we removed the term “residual colinearity” from the main text. This phrase is now mentioned (as “residual gene order colinearity”) only in **Supplementary Information section 7.2.1**. Briefly, “colinearity” refers to the same ordering of orthologs along the chromosomes of two or more species. With incomplete gene order shuffling between more closely related species, small colinear blocks of conserved gene order can persist (see, e.g., Fig. 3 of <https://www.nature.com/articles/nature06341>, which shows the disruption of colinearity with the preservation of chromosome-arm-scale synteny among drosophilids). By “residual gene order colinearity” we mean local regions of retained colinearity between two species. As such, some colinear runs of orthologous genes could be observed between two sponges (**Supplementary Information 7.2.2**, e.g., **Extended Data Figure 2h**). We have relegated this minor point/observation to the Supplement since it does not bear on our considerations of synteny in a deep phylogenetic context.

line 172: "earliest branching" -> "sister"

Thanks.

line 280: "Trichoplax" should be in italics

Thanks.

Reviewer Reports on the First Revision:

Referees' comments:

Referee #1 (Remarks to the Author):

I wanted to truly congratulate Schultz and their collaborators for their efforts in improving the manuscript. I acknowledge that they have implemented all my methodological suggestions, some of which were quite demanding (e.g., reassessing all the orthology assignments with better methods).

I still disagree with some of the conclusions or arguments made in the rebuttal letter. Point 1 about the loss of linkage, that linkage groups analyses include genes shared by all the species does not really address the concern about ancestral linkage groups being lost in evolution; in a sponge-first scenario, how can we be sure that linkage groups present in the LCA of animals were not lost in later branching ctenophores? The only way to polarise this tree is to show a shared linkage between ctenophores and the outgroups, otherwise, ctenophores could be apomorphic and be placed anywhere in the animal tree of life. I am still not convinced that there is no strong linkage signal in the Oxford plots of ctenophores vs outgroups. This is my main criticism, and I don't think it has been addressed (more on that below).

Point 2, constraints in chromosomal changes imposed by meiosis or other structural parameters are still selective pressures, and the rebuttal acknowledges that in the same paragraph; even if they are arguably not based on biological function (e.g., preserving correct meiosis is still a biological function), the fact that there are selective pressures indicates that convergent evolution could influence the evolution of linkage groups. So, I think the argument for convergent evolution has been reinforced by the rebuttal answer.

Finally, the criticism in my review about pairwise comparisons has been used to move the Oxford plots to Extended Data; instead, only the linkage plots are used now in the main text. I think this is a big mistake.

First, I do not think the linkage plots really overcome the pairwise criticism, as they are an array of pairwise comparisons aligned with each other. Second, linkage plots do not allow the quantitative assessment of the linkage conservation as well as the Oxford plots do.

I have only one major suggestion related to the latter, for the sake of transparency and to improve the debate on these results. In the previous version, I mentioned that Oxford plots for BCnS in the main figures show a strong linkage signal, while the Oxford plots for ctenophores vs outgroups (in the Extended Data) do not, thus not providing support for the second condition (i.e., ctenophores sharing conserved gene order with the outgroups). Back in the previous revision, I suggested moving

the ctenophore-outgroups Oxford plots to the main text so readers can make up their own minds. However, this new version buries even deeper ALL the Oxford plots (including the BCnS) to the very last figures in the Extended Data, leaving only the heavy-hypnotic-but-hard-to-quantify linkage figures in the main text.

If the paper wants to propose an open clear debate about this highly promising approach, my main suggestion is that the Oxford plots must be in the main text. Especially Extended Figure panels 2b to 2e (ctenophores vs other animals) compared against Extended Figures 7 and 8 (the panels comparing ctenophores vs outgroups), ideally combined with some sponge vs cnidarians/bilaterian. If this looks like a humongous and complex figure, pick a few and honest examples for each comparison and leave the rest for the Extended Data; all the BCnS plots are good and any should do, but the interesting ones for ctenophores-outgroups are Extended Figure panels 7a, 7f, 8a, and 8e.

Finally, I have a minor suggestion. Many times, the text and figures refer to synteny plots and analyses, when they should really refer to linkage groups. Synteny has a stronger evolutionary connotation than the linkage groups used here. I think the text should introduce better the difference between synteny and linkage at the beginning and correct the mentions of synteny across the manuscript.

Referee #2 (Remarks to the Author):

The manuscript is much improved, and the authors made an effort to address the comments of all the reviewers. As mentioned before, this will be an excellent addition to the literature and will advance science in this area.

Just a few minor thoughts. In the response to reviewers, the authors favor the idea of meiotic dynamics, over selection, accounting for the preservation of ALGs for 100s of millions of years. That may well be, but in their explanations, I do not understand why they assume that these lineages have maintained large populations sizes. Is there any evidence for this other than the circular argument of the presence of ALGs? Just because the lineages gave rise to major groups does not mean that the populations were not experiencing severe bottlenecks that cumulatively accounted for 10 of millions of years...or more.

Thank you for removing Placodarian, but the discussion on the name "Myriazoa" is still sorta weak. If the authors want to keep it, so be it. However, the justification for doing so (namely, "because Casey Dunn said do it") was weak and made me chuckle. There should be a more compelling reason.

The figures are much improved. There are a lot of trees, but I think they help with the points and summary.

The discussion on page 16 about modern taxa not representing ancestral forms (with reference to

sponges and ctenophores) was discussed in Halanych 2015 (ref #18). Maybe site that here.

The methods are sound and the addition of the Orthofinder analyses was a plus.

Again the manuscript was not carefully read before submitting.

1st paragraph of R7D on p.6 – missing a closing parenthesis.

P8 – change “uniformative” to “uninformative”.

P8 – bottom – “from from” and “totalling” errors

P9- the font jumps to Courier for a line or two

P13 – change “fusion” to “fusions”

Referee #3 (Remarks to the Author):

The authors have satisfactorily addressed my comments and requests. I particularly appreciate the revision of the figures - much more clear and informative.

Author Rebuttals to First Revision:

Response to Referees

Article: *Ancient gene linkages support ctenophores as sister to other animals*

Decision Date: February 2nd, 2023

Resubmission Date: February 27th, 2023

Authors: Darrin T. Schultz, Steven H.D. Haddock, Jessen V. Bredeson, Richard E. Green, Oleg Simakov, Daniel S. Rokhsar

Referee's text is in black, Authors' responses are in blue.

Referee #1 (Remarks to the Author):

I wanted to truly congratulate Schultz and their collaborators for their efforts in improving the manuscript. I acknowledge that they have implemented all my methodological suggestions, some of which were quite demanding (e.g., reassessing all the orthology assignments with better methods).

Thanks for your helpful comments, criticisms, and suggestions – they definitely spurred us to improve the manuscript!

I still disagree with some of the conclusions or arguments made in the rebuttal letter. Point 1 about the loss of linkage, that linkage groups analyses include genes shared by all the species does not really address the concern about ancestral linkage groups being lost in evolution; in a sponge-first scenario, how can we be sure that linkage groups present in the LCA of animals were not lost in later branching ctenophores? The only way to polarise this tree is to show a shared linkage between ctenophores and the outgroups, otherwise, ctenophores could be apomorphic and be placed anywhere in the animal tree of life. I am still not convinced that there is no strong linkage signal in the Oxford plots of ctenophores vs outgroups. This is my main criticism, and I don't think it has been addressed (more on that below).

Our analysis is focused on groups of genes that are consistently linked in all major animal lineages, including both sponges and ctenophores, as well as outgroups to animals. These are the "linkage groups" that we refer to in the manuscript and are the only ones that can be informative about the relative position of sponges and ctenophores. As described in the text, multiple pairs of linkage groups are found on separate chromosomes in ctenophores and outgroups are found to be fused and mixed together in all bilaterian, cnidarian, and sponges. These fusions-with-mixing unite bilaterians, cnidarians, and sponges to the exclusion of ctenophores.

To follow your example, if a linkage group were lost in ctenophores (i.e., genes are linked in sponges, bilaterians, cnidarians, and outgroups, but dispersed in ctenophores), we would not be able to use it in our cladistic analysis. In this case, homologous linkages detected in sponges and other animals would not be used to determine the phylogenetic placement of ctenophores, since such loss would be an autapomorphy.

Linkage loss is nevertheless an interesting scenario to consider in the broader context of ALG evolution. We partially touched on it in Simakov et al, 2022 (ALG_R - the only detectably 'lost' linkage, absent in chordates), however, more needs to be understood about the underlying potential technical reasoning and evolutionary mechanism (such as fast evolutionary rates of genes making orthology prediction more difficult or, rather, enhanced rates of chromosomal rearrangements/translocations, etc) that contribute to it.

The apparent lack of pairwise linkage signal in ctenophore vs outgroup Oxford plots is discussed further below.

Point 2, constraints in chromosomal changes imposed by meiosis or other structural parameters are still selective pressures, and the rebuttal acknowledges that in the same paragraph; even if they are arguably not based on biological function (e.g., preserving correct meiosis is still a biological function), the fact that there are selective pressures indicates that convergent evolution could influence the evolution of linkage groups. So, I think the argument for convergent evolution has been reinforced by the rebuttal answer.

In the previously revised manuscript we discuss in several places the classical idea that selection against translocation heterozygotes can constrain chromosome evolution, so we feel this point has been clearly made. This argument for stability based on constraints from meiosis (a classical one due to Muller, Sewall Wright, and others) does not imply any functional relationship *among the genes* involved in the linkage groups themselves. We note in the main text that we do not find any functional relationships among anciently linked genes using GO.

Finally, the criticism in my review about pairwise comparisons has been used to move the Oxford plots to Extended Data; instead, only the linkage plots are used now in the main text. I think this is a big mistake.

We address this point in two parts below.

First, I do not think the linkage plots really overcome the pairwise criticism, as they are an array of pairwise comparisons aligned with each other. Second, linkage plots do not allow the quantitative assessment of the linkage conservation as well as the Oxford plots do.

The plots in main text are not merely “an array of pairwise analyses aligned with each other.” This is a subtle but important point. Each vertical colored ribbon represents a gene that is found in all species, and each color represents a group of linked genes that is conserved in all species, based on our statistically supported findings using multi-species comparisons as described in Supplementary Information 4.4. None of our results depend on the order in which the species are arranged in these figures.

In contrast, Oxford plots (1) only show two-species comparisons, and importantly (2) do not allow readers to follow the chromosomal position of a gene, or group of genes, across more than two species, which is critical to our approach. The dotplots are for data visualization, rather than analysis.

I have only one major suggestion related to the latter, for the sake of transparency and to improve the debate on these results. In the previous version, I mentioned that Oxford plots for BCnS in the main figures show a strong linkage signal, while the Oxford plots for ctenophores vs outgroups (in the Extended Data) do not, thus not providing support for

the second condition (i.e., ctenophores sharing conserved gene order with the outgroups). Back in the previous revision, I suggested moving the ctenophore-outgroups Oxford plots to the main text so readers can make up their own minds. However, this new version buries even deeper ALL the Oxford plots (including the BCnS) to the very last figures in the Extended Data, leaving only the heavy-hypnotic-but-hard-to-quantify linkage figures in the main text.

If the paper wants to propose an open clear debate about this highly promising approach, my main suggestion is that the Oxford plots must be in the main text. Especially Extended Figure panels 2b to 2e (ctenophores vs other animals) compared against Extended Figures 7 and 8 (the panels comparing ctenophores vs outgroups), ideally combined with some sponge vs cnidarians/bilaterian. If this looks like a humongous and complex figure, pick a few and honest examples for each comparison and leave the rest for the Extended Data; all the BCnS plots are good and any should do, but the interesting ones for ctenophores-outgroups are Extended Figure panels 7a, 7f, 8a, and 8e.

Thanks for this suggestion. We definitely don't want to hide (or be accused of hiding!) these plots, but we feel that emphasizing pairwise comparisons in a main figure could give the mistaken impression that these are the key data analyses, rather than just a format for data visualization. As noted above, and in the text, all of our statistically significant claims are based on multi-species analyses (see e.g. Supplementary Information 4.4). Much of the streamlining

and clarification of our methods and findings appreciated by all the Reviewers comes from now using 'ribbon plots' as a standard representation throughout. Furthermore, this collection of pairwise Oxford plots would take up a large fraction of a journal page and still be quite small.

We have discussed this issue with the Editor and have come up with the following compromise in the interest of transparency and promoting "an open clear debate about this highly promising approach":

- We will leave a complete set of the detailed 'Oxford plots' in the Extended Data Figures as in the previous submission. As described in Nature documentation, Extended Data Figures are integral parts of the paper; they are included online within the full-text HTML and integrated in the downloadable (and zoomable) PDF, so they are in no way hidden.
- We have added a note to the main text clearly stating that the syntenic signal we use is generally not visually evident in simple pairwise analyses involving ctenophores and outgroups (pointing the reader directly to the EDFs) but making clear that our approach uses multi-species comparisons to extract statistically significant signals as shown in the main figures:

"In contrast to the readily detected conserved syntenies among sponges, cnidarians, and bilaterians, conserved syntenies involving ctenophores and non-animal outgroups are not visually evident in pairwise comparisons with other animals (**Extended Data Figures 2, 7**) but are statistically supported in multi-species comparisons (**Methods, Supplementary Information 4.**)"

We hope that this alternative solution satisfies the spirit of your suggested change!

Finally, I have a minor suggestion. Many times, the text and figures refer to synteny plots and analyses, when they should really refer to linkage groups. Synteny has a stronger evolutionary connotation than the linkage groups used here. I think the text should introduce better the difference between synteny and linkage at the beginning and correct the mentions of synteny across the manuscript.

Thanks. "Synteny" is a well-defined classical concept in genetics, introduced by J.H. Renwick, in 1971 ("The mapping of human chromosomes. *Annu. Rev. Genet.* 5: 81–120, as cited). It simply means *physical linkage without regard to gene order* ("same thread"). It was originally introduced by Renwick to refer to genes on the same chromosome that may not show direct genetic linkage (e.g., are separated by large distances in centimorgans in meiotic maps) "*Conserved synteny*" means groups of genes whose orthologs are linked together in two or more species. It is unfortunate that some authors confuse "conserved synteny" with "conserved *colinearity*," which is an even stronger statement. We have defined synteny properly in the main text and emphasize that it is synonymous with physical linkage, which should be sufficient.

Referee #2 (Remarks to the Author):

The manuscript is much improved, and the authors made an effort to address the comments of all the reviewers. As mentioned before, this will be an excellent addition to the literature and will advance science in this area.

Thanks for your helpful comments and suggestions – they definitely spurred us to improve the manuscript.

Just a few minor thoughts. In the response to reviewers, the authors favor the idea of meiotic dynamics, over selection, accounting for the preservation of ALGs for 100s of millions of years. That may well be, but in their explanations, I do not understand why they assume that these lineages have maintained large population sizes. Is there any evidence for this other than the circular argument of the presence of ALGs? Just because the lineages gave rise to major groups does not mean that the populations were not experiencing severe bottlenecks that cumulatively accounted for 10 of millions of years...or more.

Thanks. This is of course true, and we did not mean to imply that major groups could not experience bottlenecks. Our goal here, however, is not to provide a definite answer for why ALGs exist, but to briefly describe a null-hypothesis that is based on 'simple' and (already) well-understood meiotic processes. Whether and how functional constraints played a role in ALG evolution is an exciting future topic to test given the fundamental observations presented here and in previous work.

Thank you for removing Placodarian, but the discussion on the name "Myriazoa" is still sorta weak. If the authors want to keep it, so be it. However, the justification for doing so (namely, "because Casey Dunn said do it") was weak and made me chuckle. There should be a more compelling reason.

Our appeal to Casey Dunn was meant to show that we weren't going off completely on our own, but had consulted systematists who had thought about these issues! Since a unique name facilitates precise future discussions of relationships, there is a long history of naming monophyletic clades that are revealed through phylogenetic analyses (Medusozoa, Parahoxozoa, Lophotrochozoa, Cyclostomata, etc), including proposed clades that are controversial or deprecated (e.g., Coelenterata). In our manuscript, we used the acronym BCnS to refer to sponges, cnidarians, and bilaterians (and by extension, Placozoa), but that is not suitable for use as the name of a lineage. Since no accepted name exists, and since no morphological characters clearly distinguish the clade of non-ctenophore animals, we propose the name Myriazoa.

We are open to other suggestions but have not been able to come up with something that is more compelling or defensible than Myriazoa.

The figures are much improved. There are a lot of trees, but I think they help with the points and summary.

Thanks. Although there are “a lot of trees” we hope that they clarify the logic of our approach.

The discussion on page 16 about modern taxa not representing ancestral forms (with reference to sponges and ctenophores) was discussed in Halanych 2015 (ref #18). Maybe site that here.

This is a good idea, we have added this citation as suggested.

The methods are sound and the addition of the Orthofinder analyses was a plus.

Thanks.

Again the manuscript was not carefully read before submitting.

Thanks for catching these few typos that slipped through our proofreading! The font change appears to have been a google docs conversion issue; “totalling” with two “l” is the accepted British spelling.

1st paragraph of R7D on p.6 – missing a closing parenthesis.

P8 – change “uniformative” to “uninformative”.

P8 – bottom – “from from” and “totalling” errors

P9- the font jumps to Courier for a line or two

P13 – change “fusion” to “fusions”

Referee #3 (Remarks to the Author):

The authors have satisfactorily addressed my comments and requests. I particularly appreciate the revision of the figures - much more clear and informative.

Thanks for your helpful comments and suggestions – they definitely spurred us to improve the manuscript!